# (S)GD over Diagonal Linear Networks:
# Implicit Bias, Large Stepsizes and Edge of Stability

**Mathieu Even**[*]
Inria - ENS Paris

**Scott Pesme**[*]
EPFL

**Suriya Gunasekar**
Microsoft Research

**Nicolas Flammarion**
EPFL

## Abstract

In this paper, we investigate the impact of stochasticity and large stepsizes on the implicit regularisation of gradient descent (GD) and stochastic gradient descent (SGD) over 2-layer diagonal linear networks. We prove the convergence of GD and SGD with macroscopic stepsizes in an overparametrised regression setting and provide a characterisation of their solution through an implicit regularisation problem. Our characterisation provides insights on how the choice of minibatch sizes and stepsizes lead to qualitatively distinct behaviors in the solutions. Specifically, we show that for sparse regression learned with 2-layer diagonal linear networks, large stepsizes consistently benefit SGD, whereas they can hinder the recovery of sparse solutions for GD. These effects are amplified for stepsizes in a tight window just below the divergence threshold, known as the "edge of stability" regime.

## 1 Introduction

The stochastic gradient descent algorithm (SGD) [51] is the foundational algorithm for almost all neural network training. Though a remarkably simple algorithm, it has led to many impressive empirical results and is a key driver of deep learning. However the performances of SGD are quite puzzling from a theoretical point of view as (1) its convergence is highly non-trivial and (2) there exist many global minimums for the training objective which generalise very poorly [66].

To explain this second point, the concept of implicit regularisation has emerged: if overfitting is harmless in many real-world prediction tasks, it must be because the optimisation process is *implicitly favoring* solutions that have good generalisation properties for the task. The canonical example is overparametrised linear regression with more trainable parameters than number of samples: although there are infinitely many solutions that fit the samples, GD and SGD explore only a small subspace of all the possible parameters. As a result, it can be shown that they implicitly converge to the closest solution in terms of the $\ell_2$ distance, and this without explicit regularisation [66, 24].

Currently, most theoretical works on implicit regularisation have primarily focused on continuous time approximations of (S)GD where the impact of crucial hyperparameters such as the stepsize and the minibatch size are ignored. One such common simplification is to analyse gradient flow, which is a continuous time limit of GD and minibatch SGD with an infinitesimal stepsize. By definition, this analysis does not capture the effect of stepsize or stochasticity. Another approach is to approximate SGD by a stochastic gradient flow [60, 48], which tries to capture the noise and the stepsize using an appropriate stochastic differential equation. However, there are no theoretical guarantees that these results can be transferred to minibatch SGD as used in practice. This is a limitation in our understanding since the performances of most deep learning models are often sensitive to the choice of stepsize and minibatch size. The importance of stepsize and SGD minibatch size is common knowledge in practice and has also been systematically established in controlled experiments [36, 42, 20].

---

[*]Denotes equal contribution

37th Conference on Neural Information Processing Systems (NeurIPS 2023).

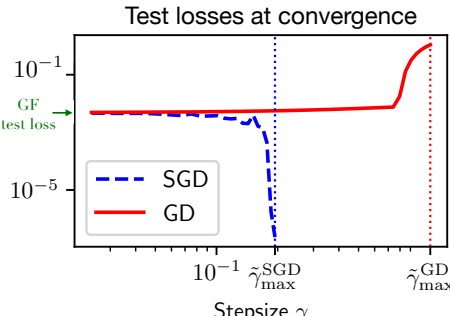

Figure 1: Noiseless sparse regression with a diagonal linear network using SGD and GD, with parameters initialized at the scale of $\alpha = 0.1$ (Section 2). The test losses at convergence for various stepsizes are plotted for GD and SGD. Small stepsizes correspond to gradient flow (GF) performance. We see that increasing the stepsize improves the generalisation properties of SGD, but deteriorates that of GD. The dashed vertical lines at stepsizes $\tilde{\gamma}_{\max}^{\mathrm{SGD}}$ and $\tilde{\gamma}_{\max}^{\mathrm{GD}}$ denote the largest stepsizes for which SGD and GD, respectively, converge. See Section 2 for the precise experimental setting.

In this work, we aim to expand our understanding of the impact of stochasticity and stepsizes by analysing the (S)GD trajectory in 2-layer diagonal networks (DLNs). In Fig. 1, we show that even in our simple network, there are significant differences between the nature of the solutions recovered by SGD and GD at macroscopic stepsizes. We discuss this behavior further in the later sections.

The 2-layer diagonal linear network which we consider is a simplified neural network that has received significant attention lately [61, 57, 26, 50]. Despite its simplicity, it surprisingly reveals training characteristics which are observed in much more complex architectures, such as the role of the initialisation [61], the role of noise [48, 50], or the emergence of saddle-to-saddle dynamics [6, 49]. It therefore serves as an ideal proxy model for gaining a deeper understanding of complex phenomenons such as the roles of stepsizes and of stochasticity as highlighted in this paper. We also point out that implicit bias and convergence for more complex architectures such as 2-layer ReLU networks, matrix multiplication are not yet fully understood, even for the simple gradient flow. Therefore studying the subtler effects of large stepsizes and stochasticity in these settings is currently out of reach.

## 1.1 Main results and paper organisation

The overparametrised regression setting and diagonal linear networks are introduced in Section 2. We formulate our theoretical results (Theorems 1 and 2) in Section 3: we prove that for **macroscopic stepsizes**, gradient descent and stochastic gradient descent over 2-layer diagonal linear networks converge to a zero-training loss solution $\beta_\infty^\star$. We further provide a refined characterization of $\beta_\infty^\star$ through a trajectory-dependent implicit regularisation problem, that captures the effects of hyperparameters of the algorithm, such as stepsizes and batchsizes, in useful and analysable ways. In Section 4 we then leverage this crisp characterisation to explain the influence of crucial parameters such as the stepsize and batch-size on the recovered solution. Importantly **our analysis shows a stark difference between the generalisation performances of GD and SGD for large stepsizes**, hence explaining the numerical results seen in Fig. 1 for the sparse regression setting. Finally, in Section 5, we use our results to shed new light on the *Edge of Stability* (*EoS*) phenomenon [14].

## 1.2 Related works

**Implicit bias.** The concept of implicit bias from optimization algorithm in neural networks has been studied extensively in the past few years, starting with early works of Telgarsky [55], Neyshabur et al. [45], Keskar et al. [36], Soudry et al. [53]. The theoretical results on implicit regularisation have been extended to multiplicative parametrisations [23, 25], linear networks [34], and homogeneous networks [40, 35, 13]. For regression loss on diagonal linear networks studied in this work, Woodworth et al. [61] demonstrate that the scale of the initialisation determines the type of solution obtained, with large initialisations yielding minimum $\ell_2$ norm solutions—the neural tangent kernel regime [30] and small initialisation resulting in minimum $\ell_1$ norm solutions—the *rich regime* [13]. The analysis relies on the link between gradient descent and mirror descent established by Ghai et al. [21] and further explored by Vaskevicius et al. [56], Wu and Rebeschini [62]. These works focus on full batch gradient, and often in the infitesimal stepsize limit (gradient flow), leading to general insights and results that do not take into account the effects of stochasticity and large stepsizes.

**The effect of stochasticity in SGD on generalisation.** The relationship between stochasticity in SGD and generalisation has been studied in various works [41, 29, 11, 38, 64]. Empirically, models generated by SGD exhibit better generalisation performance than those generated by GD [37, 31, 27].

Explanations related to the flatness of the minima picked by SGD have been proposed [28]. Label noise has been shown to influence the implicit bias of SGD [26, 8, 15, 50] by implicitly regularising the sharp minimisers. Recently, studying a *stochastic gradient flow* that models the noise of SGD in continuous time with Brownian diffusion, Pesme et al. [48] characterised for diagonal linear networks the limit of their stochastic process as the solution of an implicit regularisation problem. However similar explicit characterisation of the implicit bias remains unclear for SGD with large stepsizes.

**The effect of stepsizes in GD and SGD.** Recent efforts to understand how the choice of stepsizes affects the learning process and the properties of the recovered solution suggest that larger stepsizes lead to the minimisation of some notion of flatness of the loss function [52, 37, 44, 33, 64, 43], backed by empirical evidences or stability analyses. Larger stepsizes have also been proven to be beneficial for specific architectures or problems: two-layer network [39], regression [63], kernel regression [7] or matrix factorisation [59]. For large stepsizes, it has been observed that GD enters an *Edge of Stability (EoS)* regime [32, 14], in which the iterates and the train loss oscillate before converging to a zero-training error solution; this phenomenon has then been studied on simple toy models [1, 67, 12, 16] for GD. Recently, [2] presented empirical evidence that large stepsizes can lead to loss stabilisation and towards simpler predictors.

## 2   Setup and preliminaries

**Overparametrised linear regression.**   We consider a linear regression over inputs $X = (x_1, \dots, x_n) \in (\mathbb{R}^d)^n$ and outputs $y = (y_1, \dots, y_n) \in \mathbb{R}^n$. We consider *overparametrised* problems where input dimension $d$ is (much) larger than the number of samples $n$. In this case, there exists infinitely many linear predictors $\beta^\star \in \mathbb{R}^d$ which perfectly fit the training set, *i.e.*, $y_i = \langle \beta^\star, x_i \rangle$ for all $1 \leqslant i \leqslant n$. We call such vectors *interpolating predictors* or *interpolators* and we denote by $\mathcal{S}$ the set of all interpolators $\mathcal{S} = \{\beta^\star \in \mathbb{R}^d \text{ s.t. } \langle \beta^\star, x_i \rangle = y_i, \forall i \in [n]\}$. Note that $\mathcal{S}$ is an affine space of dimension greater than $d - n$ and equal to $\beta^\star + \mathrm{span}(x_1, \dots, x_n)^\perp$ for any $\beta^\star \in \mathcal{S}$. We consider the following quadratic loss: $\mathcal{L}(\beta) = \frac{1}{2n} \sum_{i=1}^n (\langle \beta, x_i \rangle - y_i)^2$, for $\beta \in \mathbb{R}^d$.

**2-layer linear diagonal network.** We parametrise regression vectors $\beta$ as functions $\beta_w$ of trainable parameters $w \in \mathbb{R}^p$. Although the final prediction function $x \mapsto \langle \beta_w, x \rangle$ is linear in the input $x$, the choice of the parametrisation drastically changes the solution recovered by the optimisation algorithm [25]. In the case of the linear parametrisation $\beta_w = w$ many first-order methods (SGD, GD, with or without momentum) converge towards the same solution and the choice of stepsize does not impact the recovered solution beyond convergence. In an effort to better understand the effects of stochasticity and large stepsize, we consider the next simple parametrisation, that of a 2-layer diagonal linear neural network given by:

$$\beta_w = u \odot v \text{ where } w = (u, v) \in \mathbb{R}^{2d}. \tag{1}$$

This parametrisation can be viewed as a simple neural network $x \mapsto \langle u, \sigma(\mathrm{diag}(v)x) \rangle$ where the output weights are represented by $u$, the inner weights is the diagonal matrix $\mathrm{diag}(v)$, and the activation $\sigma$ is the identity function. In this spirit, we refer to the entries of $w = (u, v) \in \mathbb{R}^{2d}$ as the *weights* and to $\beta := u \odot v \in \mathbb{R}^d$ as the *prediction parameter*. Despite the simplicity of the parametrisation (1), the loss function $F$ over parameters $w = (u, v) \in \mathbb{R}^{2d}$ is **non-convex** (and thus the corresponding optimization problem is challenging to analyse), and is given by:

$$F(w) := \mathcal{L}(u \odot v) = \frac{1}{2n} \sum_{i=1}^n (y_i - \langle u \odot v, x_i \rangle)^2. \tag{2}$$

**Mini-batch SGD.** We minimise $F$ using mini-batch SGD: let $w_0 = (u_0, v_0)$ and for $k \geqslant 0$,

$$w_{k+1} = w_k - \gamma_k \nabla F_{\mathcal{B}_k}(w_k), \quad \text{where} \quad F_{\mathcal{B}_k}(w) := \frac{1}{2b} \sum_{i \in \mathcal{B}_k} (y_i - \langle u \odot v, x_i \rangle)^2, \tag{3}$$

where $\gamma_k$ are stepsizes, $\mathcal{B}_k \subset [n]$ are mini-batches of $b \in [n]$ distinct samples sampled uniformly and independently, and $\nabla F_{\mathcal{B}_k}(w_k)$ are minibatch gradients of partial loss over $\mathcal{B}_k$, $F_{\mathcal{B}_k}(w) := \mathcal{L}_{\mathcal{B}_k}(u \odot v)$ defined above. Classical SGD and full-batch GD are special cases with $b = 1$ and $b = n$, respectively. For $k \geqslant 0$, we consider the successive prediction parameters $\beta_k := u_k \odot v_k$ built from the weights

$w_k = (u_k, v_k)$. We analyse SGD initialised at $u_0 = \sqrt{2}\boldsymbol{\alpha} \in \mathbb{R}^d_{>0}$ and $v_0 = \mathbf{0} \in \mathbb{R}^d$, resulting in $\beta_0 = \mathbf{0} \in \mathbb{R}^d$ independently of the chosen weight initialisation $\boldsymbol{\alpha}$[2].

**Experimental details.** We consider the noiseless sparse regression setting where $(x_i)_{i \in [n]} \sim \mathcal{N}(0, I_d)$ and $y_i = \langle \beta^\star_{\ell_1}, x_i \rangle$ for some $s$-sparse vector $\beta^\star_{\ell_1}$. We perform (S)GD over the DLN with a uniform initialisation $\boldsymbol{\alpha} = \alpha \mathbf{1} \in \mathbb{R}^d$ where $\alpha > 0$. Fig. 1 and Fig. 2 (left) correspond to the setup $(n, d, s, \alpha) = (20, 30, 3, 0.1)$, Fig. 2 (right) to $(n, d, s, \alpha) = (50, 100, 4, 0.1)$ and Fig. 3 to $(n, d, s, \alpha) = (50, 100, 2, 0.1)$.

**Notations.** Let $H := \nabla^2 \mathcal{L} = \frac{1}{n} \sum_i x_i x_i^\top$ denote the Hessian of $\mathcal{L}$, and for a batch $\mathcal{B} \subset [n]$ let $H_{\mathcal{B}} := \nabla^2 \mathcal{L}_{\mathcal{B}} = \frac{1}{|\mathcal{B}|} \sum_{i \in \mathcal{B}} x_i x_i^\top$ denote the Hessian of the partial loss over the batch $\mathcal{B}$. Let $L$ denote the "smoothness" such that $\forall \beta, \|H_{\mathcal{B}}\beta\|_2 \leqslant L\|\beta\|_2, \|H_{\mathcal{B}}\beta\|_\infty \leqslant L\|\beta\|_\infty$ for all batches $\mathcal{B} \subset [n]$ of size $b$. A real function (e.g, $\log, \exp$) applied to a vector must be understood as element-wise application, and for vectors $u, v \in \mathbb{R}^d$, $u^2 = (u_i^2)_{i \in [d]}$, $u \odot v = (u_i v_i)_{i \in [d]}$ and $u/v = (u_i/v_i)_{i \in [d]}$. We write $\mathbf{1}, \mathbf{0}$ for the constant vectors with coordinates 1 and 0 respectively. The Bregman divergence [9] of a differentiable convex function $h : \mathbb{R}^d \to \mathbb{R}$ is defined as $D_h(\beta_1, \beta_2) = h(\beta_1) - (h(\beta_2) + \langle \nabla h(\beta_2), \beta_1 - \beta_2 \rangle)$.

# 3 Implicit bias of SGD and GD

We start by recalling some known results on the implicit bias of gradient flow on diagonal linear networks before presenting our main theorems on characterising the (stochastic) gradient descent solutions (Theorem 1) as well as proving the convergence of the iterates (Theorem 2).

## 3.1 Warmup: gradient flow

We first review prior findings on gradient flow on diagonal linear neural networks. Woodworth et al. [61] show that the limit $\beta^*_{\boldsymbol{\alpha}}$ of the *gradient flow* $\mathrm{d}w_t = -\nabla F(w_t)\mathrm{d}t$ initialised at $(u_0, v_0) = (\sqrt{2}\boldsymbol{\alpha}, \mathbf{0})$ is the solution of the minimal interpolation problem:

$$\beta^*_{\boldsymbol{\alpha}} = \underset{\beta^\star \in \mathcal{S}}{\operatorname{argmin}} \ \psi_{\boldsymbol{\alpha}}(\beta^\star), \quad \text{where} \quad \psi_{\boldsymbol{\alpha}}(\beta) = \frac{1}{2} \sum_{i=1}^d \left( \beta_i \operatorname{arcsinh}(\frac{\beta_i}{\alpha_i^2}) - \sqrt{\beta_i^2 + \alpha_i^4} + \alpha_i^2 \right). \quad (4)$$

The convex potential $\psi_{\boldsymbol{\alpha}}$ is the **hyperbolic entropy function** (or **hypentropy**) [21]. Depending on the structure of the vector $\boldsymbol{\alpha}$, the generalisation properties of $\beta^\star_{\boldsymbol{\alpha}}$ highly vary. We point out the two main characteristics of $\boldsymbol{\alpha}$ that affect the behaviour of $\psi_{\boldsymbol{\alpha}}$ and therefore also the solution $\beta^\star_{\boldsymbol{\alpha}}$.

**1. The Scale of $\boldsymbol{\alpha}$.** For an initialisation vector $\boldsymbol{\alpha}$ we call the $\ell_1$-norm $\|\boldsymbol{\alpha}\|_1$ the **scale** of the initialisation. It is an important quantity affecting the properties of the recovered solution $\beta^\star_{\boldsymbol{\alpha}}$. To see this let us consider a uniform initialisation of the form $\boldsymbol{\alpha} = \alpha \mathbf{1}$ for a scalar value $\alpha > 0$. In this case the potential $\psi_{\boldsymbol{\alpha}}$ has the property of resembling the $\ell_1$-norm as the scale $\alpha$ vanishes: $\psi_{\boldsymbol{\alpha}} \sim \ln(1/\alpha)\|.\|_1$ as $\alpha \to 0$. Hence, a small initialisation results in a low $\ell_1$-norm solution which is known to induce sparse recovery guarantees [10]. This setting is often referred to as the "rich" regime [61]. In contrast, using a large initialisation scale leads to solutions with low $\ell_2$-norm: $\psi_{\boldsymbol{\alpha}} \sim \|.\|_2^2/(2\alpha^2)$ as $\alpha \to \infty$, a setting known as the "kernel" or "lazy" regime. Overall, to retrieve the minimum $\ell_1$-norm solution, one should use a uniform initialisation with small scale $\alpha$, see Fig. 7 in Appendix D for an illustration and [61, Theorem 2] for a precise characterisation.

**2. The Shape of $\boldsymbol{\alpha}$.** In addition to the scale of the initialisation $\boldsymbol{\alpha}$, a lesser studied aspect is its "shape", which is a term we use to refer to the relative distribution of $\{\alpha_i\}_i$ along the $d$ coordinates [3]. It is a crucial property because having $\boldsymbol{\alpha} \to \mathbf{0}$ **does not** necessarily lead to the potential $\psi_{\boldsymbol{\alpha}}$ being close to the $\ell_1$-norm. Indeed, we have that $\psi_{\boldsymbol{\alpha}}(\beta) \overset{\boldsymbol{\alpha} \to \mathbf{0}}{\sim} \sum_{i=1}^d \ln(\frac{1}{\alpha_i})|\beta_i|$ (see Appendix D), therefore if the vector $\ln(1/\boldsymbol{\alpha})$ has entries changing at different rates, then $\psi_{\boldsymbol{\alpha}}(\beta)$ is a **weighted** $\ell_1$-norm. In words, if the entries of $\boldsymbol{\alpha}$ *do not go to zero "uniformly"*, then the resulting implicit bias minimizes a

---

[2]In Appendix C, we show that the (S)GD trajectory with this initialisation exactly matches that of another common parametrisation $\beta_w = w_+^2 - w_-^2$ with initialisation $w_{+,0} = w_{-,0} = \boldsymbol{\alpha}$. The second layer of our diagonal linear network is set to 0 in order to obtain results that are easier to interpret. However, our proof techniques can be applied directly to a general initialisation, at the cost of additional notations in our Theorems.

weighed $\ell_1$-norm. This phenomenon can lead to solutions with vastly different sparsity structure than the minimum $\ell_1$-norm interpolator. See Fig. 7 and Example 1 in Appendix D.

### 3.2 Implicit bias of (stochastic) gradient descent

In Theorem 1, we prove that for an initialisation $\sqrt{2}\boldsymbol{\alpha} \in \mathbb{R}^d$ and for **arbitrary** stepsize sequences $(\gamma_k)_{k\geqslant 0}$ **if the iterates converge to an interpolator**, then this interpolator is the solution of a constrained minimisation problem which involves the hyperbolic entropy $\psi_{\boldsymbol{\alpha}_\infty}$ defined in (4), where $\boldsymbol{\alpha}_\infty \in \mathbb{R}^d$ is an effective initialisation which depends on the trajectory and on the stepsize sequence. Later, **we prove the convergence of iterates for macroscopic step sizes** in Theorem 2.

**Theorem 1** (Implicit bias of (S)GD). *Let $(u_k, v_k)_{k\geqslant 0}$ follow the mini-batch SGD recursion* (3) *initialised at $(u_0, v_0) = (\sqrt{2}\boldsymbol{\alpha}, \mathbf{0})$ and with stepsizes $(\gamma_k)_{k\geqslant 0}$. Let $(\beta_k)_{k\geqslant 0} = (u_k \odot v_k)_{k\geqslant 0}$ and assume that they converge to some interpolator $\beta_\infty^\star \in \mathcal{S}$. Then, $\beta_\infty^\star$ satisfies:*

$$\beta_\infty^\star = \underset{\beta^\star \in \mathcal{S}}{\operatorname{argmin}} \, D_{\psi_{\boldsymbol{\alpha}_\infty}}(\beta^\star, \tilde{\beta}_0), \tag{5}$$

*where $D_{\psi_{\boldsymbol{\alpha}_\infty}}$ is the Bregman divergence with hyperentropy potential $\psi_{\boldsymbol{\alpha}_\infty}$ of the **effective initialisation** $\boldsymbol{\alpha}_\infty$, and $\tilde{\beta}_0$ is a small **perturbation term**. The **effective initialisation** $\boldsymbol{\alpha}_\infty$ is given by,*

$$\boldsymbol{\alpha}_\infty^2 = \boldsymbol{\alpha}^2 \odot \exp\left(-\sum_{k=0}^\infty q(\gamma_k \nabla \mathcal{L}_{\mathcal{B}_k}(\beta_k))\right), \tag{6}$$

*where $q(x) = -\frac{1}{2}\ln((1-x^2)^2)$ satisfies $q(x) \geqslant 0$ for $|x| \leqslant \sqrt{2}$, with the convention $q(1) = +\infty$.*

*The **perturbation term** $\tilde{\beta}_0 \in \mathbb{R}^d$ is explicitly given by $\tilde{\beta}_0 = \frac{1}{2}(\boldsymbol{\alpha}_+^2 - \boldsymbol{\alpha}_-^2)$, where $q_\pm(x) = \mp 2x - \ln((1 \mp x)^2)$, and $\boldsymbol{\alpha}_\pm^2 = \boldsymbol{\alpha}^2 \odot \exp\left(-\sum_{k=0}^\infty q_\pm(\gamma_k \nabla \mathcal{L}_{\mathcal{B}_k}(\beta_k))\right)$.*

**Trajectory-dependent characterisation.** The characterisation of $\beta_\infty^\star$ in Theorem 1 holds for any stepsize schedule such that the iterates converge and goes beyond the continuous-time frameworks previously studied [61, 48]. The result even holds for adaptive stepsize schedules which keep the stepsize scalar such as AdaDelta [65]. An important aspect of our result is that $\boldsymbol{\alpha}_\infty$ and $\tilde{\beta}_0$ depend on the iterates' trajectory. Nevertheless, we argue that our formulation provides useful ingredients for understanding the implicit regularisation effects of (S)GD for this problem compared to trivial characterisations (such as *e.g.*, $\min_\beta \|\beta - \beta_\infty^\star\|$). Importantly, **the key parameters $\boldsymbol{\alpha}_\infty, \tilde{\beta}_0$ depend on crucial parameters such as the stepsize and noise in a useful and analysable manner**: understanding how they affect $\boldsymbol{\alpha}_\infty$ and $\tilde{\beta}_0$ coincides with understanding how they affect the recovered solution $\beta_\infty^\star$ and its generalisation properties. This is precisely the object of Sections 4 and 5 where we discuss the qualitative and quantitative insights from Theorem 1 in greater detail.

**The perturbation $\tilde{\beta}_0$ can be ignored.** We show in Proposition 16, under reasonable assumptions on the stepsizes, that $|\tilde{\beta}_0| \leqslant \boldsymbol{\alpha}^2$ and $\boldsymbol{\alpha}_\infty \leqslant \boldsymbol{\alpha}$ (component-wise). The magnitude of $\tilde{\beta}_0$ is therefore negligible in front of the magnitudes of $\beta^\star \in S$ and one can roughly ignore the term $\tilde{\beta}_0$. Hence, the implicit regularisation Eq. (5) can be thought of as $\beta_\infty^\star \approx \operatorname{argmin}_{\beta^\star \in S} D_{\psi_{\boldsymbol{\alpha}_\infty}}(\beta^\star, 0) = \psi_{\boldsymbol{\alpha}_\infty}(\beta^\star)$, and thus *the solution $\beta_\infty^\star$ minimises the same potential function that the solution of gradient flow (see Eq. (4)), but with an effective initialisation $\boldsymbol{\alpha}_\infty$*. Also note that for $\gamma_k \equiv \gamma \to 0$ we have $\boldsymbol{\alpha}_\infty \to \boldsymbol{\alpha}$ and $\tilde{\beta}_0 \to \mathbf{0}$ (Proposition 19), recovering the previously known result for gradient flow (4).

**Deviation from gradient flow.** The difference with gradient flow is directly associated with the quantity $\sum_k q(\gamma_k \nabla \mathcal{L}_{\mathcal{B}_k}(\beta_k))$. Also, as the (stochastic) gradients converge to 0 and $q(x) \overset{x\to 0}{\sim} x^2$, one should think of this sum as roughly being $\sum_k \nabla \mathcal{L}_{\mathcal{B}_k}(\beta_k)^2$: the larger this sum, the more the recovered solution differs from that of gradient flow. The full picture of how large stepsizes and stochasticity impact the generalisation properties of $\beta_\infty^\star$ and the recovery of minimum $\ell_1$-norm solution is nuanced as clearly seen in Fig. 1.

### 3.3 Convergence of the iterates

Theorem 1 provides the implicit minimisation problem but says nothing about the convergence of the iterates. Here we show under very reasonable assumptions on the stepsizes that the iterates indeed

converge towards a global optimum. Note that since the loss $F$ is non-convex, such a convergence result is non-trivial and requires an involved analysis.

**Theorem 2** (Convergence of the iterates). *Let $(u_k, v_k)_{k \geqslant 0}$ follow the mini-batch SGD recursion* (3) *initialised at $u_0 = \sqrt{2}\boldsymbol{\alpha} \in \mathbb{R}^d_{>0}$ and $v_0 = \mathbf{0}$, and let $(\beta_k)_{k \geqslant 0} = (u_k \odot v_k)_{k \geqslant 0}$. Recall the "smoothness" parameter $L$ on the minibatch loss defined in the notations. There exist $B > 0$ verifying $B = \tilde{\mathcal{O}}(\min_{\beta^\star \in \mathcal{S}} \|\beta^\star\|_\infty)$ and a numerical constant $c > 0$ such that for stepsizes satisfying $\gamma_k \leqslant \frac{c}{LB}$, the iterates $(\beta_k)_{k \geqslant 0}$ converge almost surely to the interpolator $\beta^\star_\infty$ solution of Eq.* (5).

In fact, we can be more precise by showing an exponential rate of convergence of the losses as well as characterise the rate of convergence of the iterates as follows.

**Proposition 1** (Quantitative convergence rates). *For a uniform initialisation $\boldsymbol{\alpha} = \alpha \mathbf{1}$ and under the assumptions of Theorem* 2, *we have:*

$$\mathbb{E}\left[\mathcal{L}(\beta_k)\right] \leqslant \left(1 - \frac{1}{2}\gamma\alpha^2\lambda_b\right)^k \mathcal{L}(\beta_0) \quad \text{and} \quad \mathbb{E}\left[\left\|\beta_k - \beta^\star_{\alpha_k}\right\|^2\right] \leqslant C \left(1 - \frac{1}{2}\gamma\alpha^2\lambda_b\right)^k,$$

*where $\lambda_b > 0$ is the largest value such that $\lambda_b H \preceq \mathbb{E}_\mathcal{B}[H_\mathcal{B}]$, $C = 2B(\alpha^2\lambda^+_{\min})^{-1}\left(1 + (4B\lambda_{\max})(\alpha^2\lambda^+_{\min})^{-1}\right)\mathcal{L}(\beta_0)$ and $\lambda^+_{\min}, \lambda_{\max} > 0$ are respectively the smallest non-null and the largest eigenevalues of $H$, and $\beta^\star_{\alpha_k}$ is the interpolator that minimises the perturbed hypentropy $h_k$ of parameter $\alpha_k$, as defined in Eq.* (7) *in the next subsection.*

The convergence of the losses is proved directly using the time-varying mirror structure that we exhibit in the next subsection, the convergence of the iterates is proved by studying the curvature of the mirror maps on a small neighborhood around the affine interpolation space.

## 3.4 Sketch of proof through a time varying mirror descent

As in the continuous-time framework, our results heavily rely on showing that the iterates $(\beta_k)_k$ follow a mirror descent recursion with time-varying potentials on the convex loss $\mathcal{L}(\beta)$. To show this, we first define the following quantities:

$$\boldsymbol{\alpha}^2_k := \boldsymbol{\alpha}_{+,k} \odot \boldsymbol{\alpha}_{-,k} \qquad \text{and} \qquad \phi_k := \frac{1}{2}\operatorname{arcsinh}\left(\frac{\boldsymbol{\alpha}^2_{+,k} - \boldsymbol{\alpha}^2_{-,k}}{2\boldsymbol{\alpha}^2_k}\right) \in \mathbb{R}^d,$$

where $\boldsymbol{\alpha}_{\pm,k} := \boldsymbol{\alpha}\exp\left(-\frac{1}{2}\sum_{i=0}^{k-1} q_\pm\left(\gamma_\ell\nabla\mathcal{L}_{\mathcal{B}_\ell}(\beta_\ell)\right)\right) \in \mathbb{R}^d$. Finally for $k \geqslant 0$, we define the potentials $(h_k : \mathbb{R}^d \to \mathbb{R})_{k \geqslant 0}$ as:

$$h_k(\beta) = \psi_{\boldsymbol{\alpha}_k}(\beta) - \langle\phi_k, \beta\rangle. \tag{7}$$

Where $\psi_{\boldsymbol{\alpha}_k}$ is the hyperbolic entropy function defined Eq. (4). Now that all the relevant quantities are defined, we can state the following proposition which explicits the time-varying stochastic mirror descent.

**Proposition 2.** *The iterates $(\beta_k = u_k \odot v_k)_{k \geqslant 0}$ from Eq.* (3) *satisfy the Stochastic Mirror Descent recursion with varying potentials $(h_k)_k$:*

$$\nabla h_{k+1}(\beta_{k+1}) = \nabla h_k(\beta_k) - \gamma_k\nabla\mathcal{L}_{\mathcal{B}_k}(\beta_k),$$

*where $h_k : \mathbb{R}^d \to \mathbb{R}$ for $k \geqslant 0$ are defined Eq.* (7). *Since $\nabla h_0(\beta_0) = 0$ we have:*

$$\nabla h_k(\beta_k) \in \operatorname{span}(x_1, \ldots, x_n). \tag{8}$$

Theorem 1 and 2 and Proposition 1 follow from this key proposition: by suitably modifying classical convex optimization techniques to account for the time-varying potentials, we can prove the convergence of the iterates towards an interpolator $\beta^\star_\infty$ along with that of the relevant quantities $\boldsymbol{\alpha}_{\pm,k}$, $\boldsymbol{\alpha}_k$ and $\phi_k$. The implicit regularisation problem then directly follows from: (1) the limit condition $\nabla h_\infty(\beta_\infty) \in \operatorname{Span}(x_1, \ldots, x_n)$ as seen from Eq. (8) and (2) the interpolation condition $X\beta^\star_\infty = y$. Indeed, these two conditions exactly correspond to the KKT conditions of the convex problem Eq. (5).

# 4 Analysis of the impact of the stepsize and stochasticity on $\alpha_\infty$

In this section, we analyse the effects of large stepsizes and stochasticity on the implicit bias of (S)GD. We focus on how these factors influence the effective initialisation $\alpha_\infty$, which plays a key role as shown in Theorem 1. From its definition in Eq. (6), we see that $\alpha_\infty$ is a function of the vector $\sum_k q(\gamma_k \nabla \mathcal{L}_{\mathcal{B}_k}(\beta_k))$. We henceforth call this quantity the *gain vector*. For simplicity of the discussions, from now on, we consider constant stepsizes $\gamma_k = \gamma$ for all $k \geqslant 0$ and a uniform initialisation of the weights $\alpha = \alpha \mathbf{1}$ with $\alpha > 0$. We can then write the gain vector as:

$$\text{Gain}_\gamma := \ln\left(\frac{\alpha^2}{\alpha_\infty^2}\right) = \sum_k q(\gamma \nabla \mathcal{L}_{\mathcal{B}_k}(\beta_k)) \in \mathbb{R}^d.$$

Following our discussion in section 3.1 on the scale and the shape of $\alpha_\infty$, we recall the link between the scale and shape of $\text{Gain}_\gamma$ and the recovered solution:

**1.** The **scale** of $\text{Gain}_\gamma$, i.e. the magnitude of $\|\text{Gain}_\gamma\|_1$ indicates how much the implicit bias of (S)GD differs from that of gradient flow: $\|\text{Gain}_\gamma\|_1 \sim 0$ implies that $\alpha_\infty \sim \alpha$ and therefore the recovered solution is close to that of gradient flow. On the contrary, $\|\text{Gain}_\gamma\|_1 \gg \ln(1/\alpha)$ implies that $\alpha_\infty$ has effective scale much smaller than $\alpha$ thereby changing the implicit regularisation Eq. (5).

**2.** The **shape** of $\text{Gain}_\gamma$ indicates which coordinates of $\beta$ in the associated minimum weighted $\ell_1$ problem are most penalised. First recall from Section 3.1 that a uniformly large $\text{Gain}_\gamma$ leads to $\psi_{\alpha_\infty}$ being closer to the $\ell_1$-norm. However, with small weight initialisation $\alpha \to 0$, we have,

$$\psi_{\alpha_\infty}(\beta) \sim \ln(\frac{1}{\alpha})\|\beta\|_1 + \sum_{i=1}^d \text{Gain}_\gamma(\mathrm{i})|\beta_\mathrm{i}|\,, \tag{9}$$

In this case, having a heterogeneously large vector $\text{Gain}_\gamma$ leads to a weighted $\ell_1$ norm as the effective implicit regularisation, where the coordinates of $\beta$ corresponding to the largest entries of $\text{Gain}_\gamma$ are less likely to be recovered.

## 4.1 The scale of $\text{Gain}_\gamma$ is increasing with the stepsize

The following proposition highlights the dependencies of the scale of the gain $\|\text{Gain}_\gamma\|_1$ in terms of various problem constants.

**Proposition 3.** *Let $\Lambda_b, \lambda_b > 0$[3] be the largest and smallest values, respectively, such that $\lambda_b H \preceq \mathbb{E}_{\mathcal{B}}\left[H_{\mathcal{B}}^2\right] \preceq \Lambda_b H$. For any stepsize $\gamma > 0$ satisfying $\gamma \leqslant \frac{c}{BL}$ (as in Theorem 2), initialisation $\alpha\mathbf{1}$ and batch size $b \in [n]$, the magnitude of the gain satisfies:*

$$\lambda_b \gamma^2 \sum_k \mathbb{E}\mathcal{L}(\beta_k) \leqslant \mathbb{E}\left[\|\text{Gain}_\gamma\|_1\right] \leqslant 2\Lambda_b \gamma^2 \sum_k \mathbb{E}\mathcal{L}(\beta_k)\,, \tag{10}$$

*where the expectation is over a uniform and independent sampling of the batches $(\mathcal{B}_k)_{k\geqslant 0}$.*

**The slower the training, the larger the gain.** Eq. (10) shows that the slower the training loss converges to 0, the larger the sum of the loss and therefore the larger the scale of $\text{Gain}_\gamma$. This means that the (S)GD trajectory deviates from that of gradient flow if the stepsize and/or noise slows down the training. This supports observations previously made from stochastic gradient flow [48] analysis.

**The bigger the stepsize, the larger the gain.** The effect of the stepsize on the magnitude of the gain is not directly visible in Eq. (10) because a larger stepsize tends to speed up the training. For stepsize $0 < \gamma \leqslant \gamma_{\max} = \frac{c}{BL}$ as in Theorem 2 we have that (see Appendix G.1):

$$\sum_k \gamma^2 \mathcal{L}(\beta_k) = \Theta\left(\gamma \ln\left(\frac{1}{\alpha}\right)\|\beta_{\ell_1}^\star\|_1\right)\,. \tag{11}$$

Eq. (11) clearly shows that increasing the stepsize **boosts** the magnitude $\|\text{Gain}_\gamma\|_1$ up until the limit of $\gamma_{\max}$. Therefore, the larger the stepsize the smaller is the effective scale of $\alpha_\infty$. In turn, larger gap between $\alpha_\infty$ and $\alpha$ leads to a larger deviation of (S)GD from the gradient flow.

---

[3] $\Lambda_b, \lambda_b > 0$ are data-dependent constants; for $b = n$, we have $(\lambda_n, \Lambda_n) = (\lambda_{\min}^+(H), \lambda_{\max}(H))$ where $\lambda_{\min}^+(H)$ is the smallest non-null eigenvalue of $H$; for $b = 1$, we have $\min_i \|x_i\|_2^2 \leqslant \lambda_1 \leqslant \Lambda_1 \leqslant \max_i \|x_i\|_2^2$.

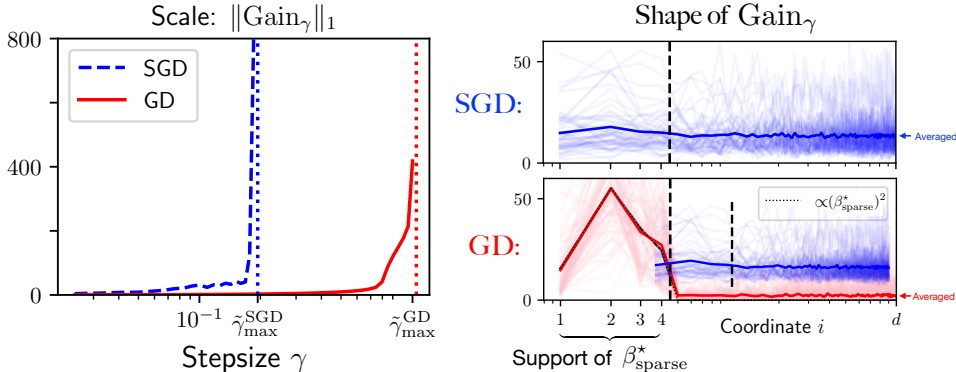

Figure 2: *Left:* the scale of $\mathrm{Gain}_\gamma$ explodes as $\gamma \to \tilde{\gamma}_{\max}$ for both GD and SGD. *Right:* $\beta^\star_{\mathrm{sparse}}$ is fixed, we perform 100 runs of GD and SGD with different feature matrices, and we plot the $d$ coordinates of $\mathrm{Gain}_\gamma$ (for GD and SGD) on the $x$-axis (which is in log scale for better visualisation). The shape of $\mathrm{Gain}_\gamma^{\mathrm{SGD}}$ is homogeneous whereas that of GD is heterogeneous with much higher magnitude on the support of $\beta^\star_{\mathrm{sparse}}$. The shape of $\mathrm{Gain}_\gamma^{\mathrm{GD}}$ is proportional to the expected gradient at initialisation which is $(\beta^\star_{\mathrm{sparse}})^2$.

**Large stepsizes and Edge of Stability.** The previous paragraph holds for stepsizes smaller than $\gamma_{\max}$ for which we can theoretically prove convergence. But what if we use even bigger stepsizes? Let $(\beta^\gamma_k)_k$ denote the iterates generated with stepsize $\gamma$ and let us define $\tilde{\gamma}_{\max} := \sup_{\gamma \geqslant 0}\{\gamma \text{ s.t. } \forall \gamma' \in (0,\gamma), \sum_k \mathcal{L}(\beta^{\gamma'}_k) < \infty\}$, which corresponds to the largest stepsize such that the iterates still converge for a given problem (even if not provably so). From Proposition 3 we have that $\gamma_{\max} \leqslant \tilde{\gamma}_{\max}$. As we approach this upper bound on convergence $\gamma \to \tilde{\gamma}_{\max}$, the sum $\sum_k \mathcal{L}(\beta^\gamma_k)$ diverges. For such large stepsizes, the iterates of gradient descent tend to "bounce" and this regime is commonly referred to as the *Edge of Stability*. In this regime, the convergence of the loss can be made arbitrarily slow due to these bouncing effects. As a consequence, as seen through Eq. (10), the magnitude of $\mathrm{Gain}_\gamma$ can be become arbitrarily big as observed in Fig. 2 (left). In this regime, the recovered solution tends to dramatically differ from the gradient flow solution, as seen in Fig. 1.

**Impact of stochasticity and linear scaling rule.** Assuming inputs $x_i$ sampled from $\mathcal{N}(0, \sigma^2 I_d)$ with $\sigma^2 > 0$, we obtain $\mathbb{E}\left[\|\mathrm{Gain}_\gamma\|_1\right] = \Theta\left(\gamma \frac{\sigma^2 d}{b} \ln\left(\frac{1}{\alpha}\right) \|\beta^\star_{\ell_1}\|_1\right)$, w.h.p. over the dataset (see Appendix G.3, Proposition 17). The scale of $\mathrm{Gain}_\gamma$ decreases with batch size and there exists a factor $n$ between that of SGD and that of GD. Additionally, the magnitude of $\mathrm{Gain}_\gamma$ depends on $\frac{\gamma}{b}$, resembling the **linear scaling rule** commonly used in deep learning [22].

By analysing the magnitude $\|\mathrm{Gain}_\gamma\|_1$, we have explained **the distinct behavior of (S)GD with large stepsizes compared to gradient flow**. However, our current analysis does not qualitatively distinguish the behavior between SGD and GD beyond the linear stepsize scaling rules, in contrast with Fig. 1. A deeper understanding of the shape of $\mathrm{Gain}\gamma$ is needed to explain this disparity.

### 4.2 The shape of $\mathrm{Gain}_\gamma$ explains the differences between GD and SGD

In this section, we restrict our presentation to single batch SGD ($b = 1$) and full batch GD ($b = n$). When visualising the typical shape of $\mathrm{Gain}_\gamma$ for large stepsizes (see Fig. 2 - right), we note that GD and SGD behave very differently. For GD, the magnitude of $\mathrm{Gain}_\gamma$ is higher for coordinates in the support of $\beta^\star_{\ell_1}$ and thus these coordinates are adversely weighted in the asymptotic limit of $\psi_{\boldsymbol{\alpha}_\infty}$ (per (9)). This explains the distinction seed in Fig. 1, where GD in this regime has poor sparse recovery despite having a small scale of $\boldsymbol{\alpha}_\infty$, as opposed to SGD that behaves well.

The **shape** of $\mathrm{Gain}_\gamma$ is determined by the sum of the squared gradients $\sum_k \nabla\mathcal{L}_{\mathcal{B}_k}(\beta_k)^2$, and in particular by the degree of heterogeneity among the coordinates of this sum. Precisely analysing the sum over the whole trajectory of the iterates $(\beta_k)_k$ is technically out of reach. However, we empirically observe for the trajectories shown in Fig. 2 that the shape is largely determined within the first few iterates as formalized in the observation below.

**Observation 1.** $\sum_k \nabla\mathcal{L}_{\mathcal{B}_k}(\beta_k)^2 \asymp \mathbb{E}[\nabla\mathcal{L}_{\mathcal{B}_k}(\beta_0)^2]$.

In the simple case of a Gaussian noiseless sparse recovery problem (where $y_i = \langle \beta^\star_{\mathrm{sparse}}, x_i \rangle$ for some sparse vector $\beta^\star_{\mathrm{sparse}}$), we can control these gradients for GD and SGD (Appendix G.4) as:

$$\nabla \mathcal{L}(\beta_0)^2 = (\beta^\star_{\mathrm{sparse}})^2 + \varepsilon \,, \text{ for some } \varepsilon \text{ verifying } \|\varepsilon\|_\infty << \left\|\beta^\star_{\mathrm{sparse}}\right\|_\infty^2 \,, \tag{12}$$

$$\mathbb{E}_{i_0}[\nabla \mathcal{L}_{i_0}(\beta_0)^2] = \Theta\left(\|\beta^\star_{\mathrm{sparse}}\|_2^2 \mathbf{1}\right) . \tag{13}$$

**The gradient of GD is heterogeneous.** Since $\beta^\star_{\mathrm{sparse}}$ is sparse by definition, we deduce from Eq. (25) that $\nabla \mathcal{L}(\beta_0)$ is heterogeneous with larger values corresponding to the support of $\beta^\star_{\mathrm{sparse}}$. Along with observation 1, this means that $\mathrm{Gain}_\gamma$ **has much larger values on the support of** $\beta^\star_{\mathrm{sparse}}$. The corresponding weighted $\ell_1$-norm therefore penalises the coordinates belonging to the support of $\beta^\star_{\mathrm{sparse}}$, which hinders the recovery of $\beta^\star_{\mathrm{sparse}}$ (as explained in Example 1, Appendix D).

**The stochastic gradient of SGD is homogeneous.** On the contrary, from Eq. (26), we have that the initial stochastic gradients are homogeneous, leading to a weighted $\ell_1$-norm where the weights are roughly balanced. The corresponding weighted $\ell_1$-norm is therefore close to the uniform $\ell_1$-norm and the classical $\ell_1$ recovery guarantees are expected.

**Overall summary of the joint effects of the scale and shape.** In summary we have the following trichotomy which fully explains Fig. 1:

1. for small stepsizes, the scale is small, and (S)GD solutions are close to that of gradient flow;

2. for large stepsizes the scale is significant and the recovered solutions differ from GF:

    - for SGD the shape of $\alpha_\infty$ is uniform, the associated norm is closer to the $\ell_1$-norm and the recovered solution is closer to the sparse solution;

    - for GD, the shape is heterogeneous, the associated norm is weighted such that it hinders the recovery of the sparse solution.

In this last section, we relate heuristically these findings to the *Edge of Stability* phenomenon.

## 5 Edge of Stability: the neural point of view

In recent years it has been noticed that when training neural networks with 'large' stepsizes at the limit of divergence, GD enters the *Edge of Stability (EoS)* regime. In this regime, as seen in Fig. 3, the iterates of GD 'bounce' / 'oscillate'. In this section, we come back to the point of view of the weights $w_k = (u_k, v_k) \in \mathbb{R}^{2d}$ and make the connection between our previous results and the common understanding of the *EoS* phenomenon. The question we seek to answer is: in which case does GD enter the *EoS* regime, and if so, what are the consequences on the trajectory? *Keep in mind that this section aims to provide insights rather than formal statements.* We study the GD trajectory starting from a small initialisation $\alpha = \alpha \mathbf{1}$ where $\alpha \ll 1$ such that we can consider that gradient flow converges close to the sparse interpolator $\beta^\star_{\mathrm{sparse}} = \beta_{w^\star_{\mathrm{sparse}}}$ corresponding to the weights $w^\star_{\mathrm{sparse}} = (\sqrt{|\beta^\star_{\mathrm{sparse}}|}, \mathrm{sign}(\beta^\star_{\mathrm{sparse}})\sqrt{|\beta^\star_{\mathrm{sparse}}|})$ (see Lemma 1 in [49] for the mapping from the predictors to weights for gradient flow). The trajectory of GD as seen in Fig. 3 (left) can be decomposed into up to 3 phases.

**First phase: gradient flow.** The stepsize is appropriate for the local curvature (as seen in Fig. 3, lower right) around initialisation and the iterates of GD remain close to the trajectory of gradient flow (in black in Fig. 3). If the stepsize is such that $\gamma < \frac{2}{\lambda_{\max}(\nabla^2 F(w^\star_{\mathrm{sparse}}))}$, then it is compatible with the local curvature and the iterates can converge: in this case GF and GD converge to the same point (as seen in Fig. 1 for small stepsizes). For larger $\gamma > \frac{2}{\lambda_{\max}(\nabla^2 F(w^\star_{\mathrm{sparse}}))}$ (as is the case for $\gamma_{\mathrm{GD}}$ in Fig. 3, lower right), the iterates cannot converge to $\beta^\star_{\mathrm{sparse}}$ and we enter the oscillating phase.

**Second phase: oscillations.** The iterates start oscillating. The gradient of $F$ writes $\nabla_{(u,v)} F(w) \sim (\nabla \mathcal{L}(\beta) \odot v, \nabla \mathcal{L}(\beta) \odot u)$ and for $w$ in the vicinity of $w^\star_{\mathrm{sparse}}$ we have that $u_i \approx v_i \approx 0$ for $i \notin \mathrm{supp}(\beta^\star_{\mathrm{sparse}})$. Therefore for $w \sim w^\star_{\mathrm{sparse}}$ we have that $\nabla_u F(w)_i \approx \nabla_v F(w)_i \approx 0$ for $i \notin \mathrm{supp}(\beta^\star_{\mathrm{sparse}})$ and the gradients roughly belong to $\mathrm{Span}(e_i, e_{i+d})_{i \in \mathrm{supp}(\beta^\star_{\mathrm{sparse}})}$. This means

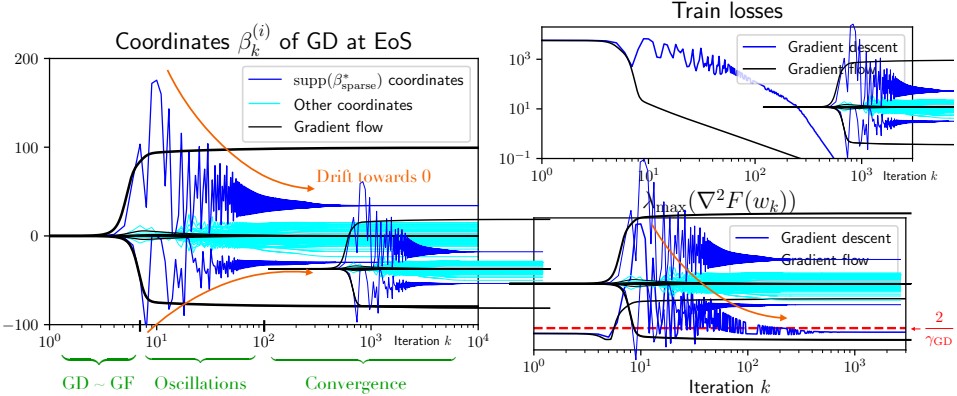

Figure 3: GD at the *EoS*. *Left:* For GD, the coordinates on the support of $\beta^\star_{\mathrm{sparse}}$ oscillate and drift towards 0. *Right, top:* The GD train losses saturate before eventually converging. *Bottom:* GF converges towards a solution that has a high hessian maximum eigenvalue. GD cannot converge towards this solution because of its large stepsize: it therefore drifts towards a solution that has a curvature just below $2/\gamma$.

.

that only the coordinates of the weights $(u_i, v_i)$ for $i \in \mathrm{supp}(\beta^\star_{\mathrm{sparse}})$ can oscillate and similarly for $(\beta_i)_{i\in\mathrm{supp}(\beta^\star_{\mathrm{sparse}})}$ (as seen Fig. 3 left).

**Last phase: convergence.** Due to the oscillations, the iterates gradually drift towards a region of lower curvature (Fig. 3, lower right, the sharpness decreases) where they may (potentially) converge. Theorem 1 enables us to understand where they converge: the coordinates of $\beta_k$ that have oscillated significantly along the trajectory belong to the support of $\beta^\star_{\mathrm{sparse}}$, and therefore $\mathrm{Gain}_\gamma(i)$ becomes much larger for $i \in \mathrm{supp}(\beta^\star_{\mathrm{sparse}})$ than for the other coordinates. Thus, the coordinates of the solution recovered in the *EoS* regime are heavily penalised on the support of the sparse solution. This is observed in Fig. 3 (left): the oscillations of $(\beta_i)_{i\in\mathrm{supp}(\beta^\star_{\mathrm{sparse}})}$ lead to a gradual shift of these coordinates towards 0, hindering an accurate recovery of the solution $\beta^\star_{\mathrm{sparse}}$.

**SGD in the *EoS* regime.** In contrast to the behavior of GD where the oscillations primarily occur on the non-sparse coordinates of ground truth sparse model, for SGD we see a different behavior in Fig. 6 (Appendix A). For stepsizes in the *EoS* regime, just below the non-convergence threshold: the fluctuation of the coordinates occurs evenly over all coordinates, leading to a uniform $\boldsymbol{\alpha}_\infty$. These fluctuations are reminiscent of label-noise SGD [2], that have been shown to recover the sparse interpolator in diagonal linear networks [50].

# 6 Conclusion

We study the effect of stochasticity along with large stepsizes when training DLNs with (S)GD. We prove convergence of the iterates as well as explicitly characterise the recovered solution by exhibiting an implicit regularisation problem which depends on the iterates' trajectory. In essence the impact of stepsize and minibatch size are captured by the effective initialisation parameter $\boldsymbol{\alpha}_\infty$ that depends on these choices in an informative way. We then use our characterisation to explain key empirical differences between SGD and GD and provide further insights on the role of stepsize and stochasticity. In particular, our characterisation explains the fundamentally different generalisation properties of SGD and GD solutions at large stepsizes as seen in Fig. 1: without stochasticity, the use of large stepsizes can prevent the recovery of the sparse interpolator, even though the effective scale of the initialization decreases with larger stepsize for both SGD and GD. We also provide insights on the link between the *Edge of Stability* regime and our results.

**Aknowledgements**

M. Even deeply thanks Laurent Massoulié for making it possible to visit Microsoft Research and the Washington state during an internship supervised by Suriya Gunasekar, the MSR Machine Learning Foundations group for hosting him, and Martin Jaggi for inviting him for a week in Lausanne at EPFL, making it possible to meet and discuss with Scott Pesme and Nicolas Flammarion.

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

# A  Additional experiments and results

## A.1  Uncentered data

When the data is uncentered, the discussion and the conclusion for GD are somewhat different. This paragraph is motivated by the observation of Nacson et al. [44] who notice that GD with large stepsizes helps to recover low $\ell_1$ solutions for uncentered data (Fig. 4). We make the following assumptions on the uncentered inputs.

**Assumption 1.** *There exist $\mu \in \mathbb{R}^d$ and $\delta, c_0, c_1, c_2 > 0$ such that for all $s$-sparse vectors $\beta$ verifying $\langle \mu, \beta \rangle \geqslant c_0 \|\beta\|_\infty \|\mu\|_\infty$, there exists $\varepsilon \in \mathbb{R}^d$ such that $(X^\top X)\beta = \langle \beta, \mu \rangle \mu + \varepsilon$ where $\|\varepsilon\|_2 \leqslant \delta \|\beta\|_2$ and $c_1 \langle \beta, \mu \rangle^2 \mu^2 \leqslant \frac{1}{n} \sum_i x_i^2 \langle x_i, \beta \rangle^2 \leqslant c_2 \langle \beta, \mu \rangle^2 \mu^2$.*

Assumption 1 is not restrictive and holds with high probability for $\mathcal{N}(\mu \mathbf{1}, \sigma^2 I_d)$ inputs when $\mu \gg \sigma \mathbf{1}$ (see Lemma 9 in Appendix). The following lemma characterises the initial shape of SGD and GD gradients for uncentered data.

**Proposition 4** (Shape of the (stochastic) gradient at initialisation). *Under Assumption 1 and if $\langle \mu, \beta^\star_{\mathrm{sparse}} \rangle \geqslant c_0 \|\beta\|_\infty \|\mu\|_\infty$, the squared full batch gradient and the expected stochastic gradient descent at initialisation satisfy, for some $\varepsilon$ satisfying $\|\varepsilon\|_\infty \ll \|\beta_{\mathrm{sparse}}\|_2$:*

$$\nabla \mathcal{L}(\beta_0) = \langle \beta^\star_{\mathrm{sparse}}, \mu \rangle^2 \mu^2 + \varepsilon, \tag{14}$$

$$\mathbb{E}_{i \sim \mathrm{Unif}([\mathrm{n}])}[\nabla \mathcal{L}_i(\beta_0)^2] = \Theta\left( \langle \beta^\star_{\mathrm{sparse}}, \mu \rangle^2 \mu^2 \right). \tag{15}$$

In this case the initial gradients of SGD and of GD **are both homogeneous**, explaining the behaviours of gradient descent in Fig. 4 (App. A): large stepsizes help in the recovery of the sparse solution in the presence of uncentered data, as opposed to centered data. Note that for decentered data with a $\mu \in \mathbb{R}^d$ orthogonal to $\beta^\star_{\mathrm{sparse}}$, there is no effect of decentering on the recovered solution. If the support of $\mu$ is the same as that of $\beta^\star_{\mathrm{sparse}}$, the effect is detrimental and the same discussion as in the centered data case applies.

Fig. 4: for uncentered data the solutions of GD and SGD have similar behaviours, corroborating Proposition 4.

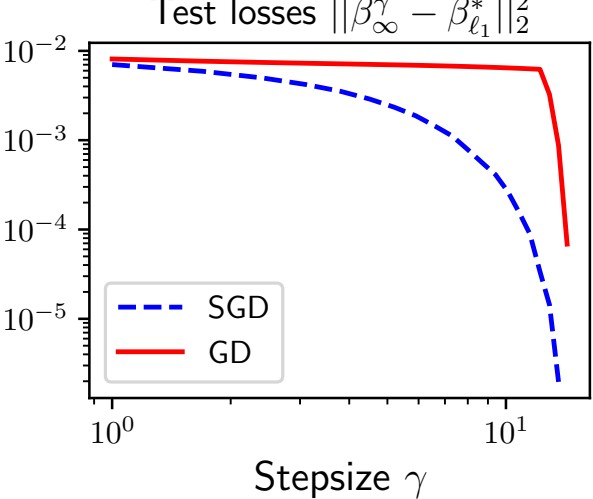

Figure 4: Noiseless sparse regression with a 2-layer DLN with uncentered data $x_i \sim \mathcal{N}(\mu \mathbf{1}, I_d)$ where $\mu = 5$. All the stepsizes lead to convergence to a global solution and the solutions of SGD and GD have similar behaviours, corroborating Proposition 4. The setup corresponds to $(n, d, s, \alpha) = (20, 30, 3, 0.1)$.

## A.2  Behaviour of the maximal value and trace of the hessian

Here in Fig. 5, we provide some additional experiments on the behaviour of: (1) the maximum eigenvalue of the hessian $\nabla^2 F(w^\gamma_\infty)$ at the convergence of the iterates of SGD and GD (2) the trace

of hessian at the convergence of the iterates. As is clearly observed, increasing the stepsize for GD leads to a 'flatter' minimum in terms of the maximum eigenvalue of the hessian, while increasing the stepsize for SGD leads to a 'flatter' minimum in terms of its trace. These two solutions have very different structures. Indeed from the value of the hessian Eq. (22) at a global solution, and (very) roughly assuming that '$X^\top X = I_d$' and that '$\alpha \sim 0$' (pushing the EoS phenomenon), one can see that minimising $\lambda_{\max}(\nabla^2 F(w))$ under the constraints $X(w_+^2 - w_-^2) = y$ and $w_+ \odot w_- = 0$ is equivalent to minimising $\|\beta\|_\infty$ under the constraint $X\beta = y$. On the other hand minimising the trace of the hessian is equivalent to minimising the $\ell_1$-norm.

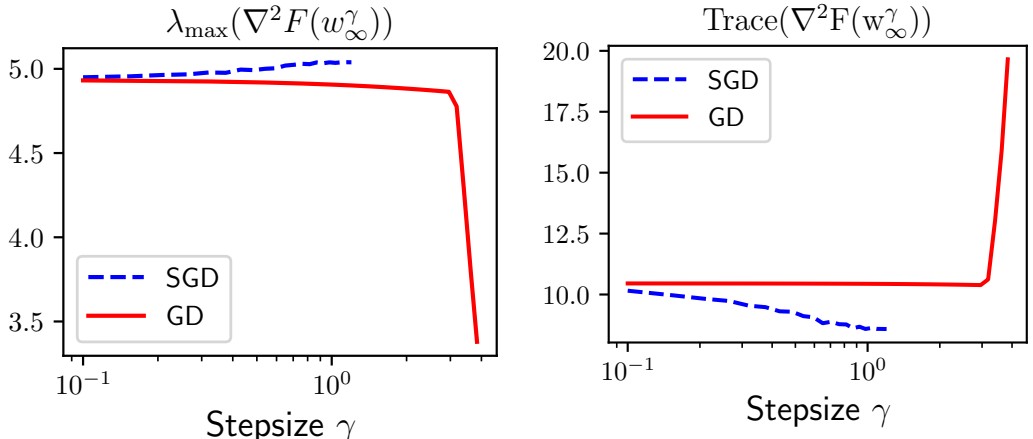

Figure 5: Noiseless sparse regression setting. Diagonal linear network. Centered data. Behaviour of 2 different types of flatness of the recovered solution by SGD and GD depending on the stepsize. The setup corresponds to $(n, d, s, \alpha) = (20, 30, 3, 0.1)$.

### A.3    Edge of Stability for SGD

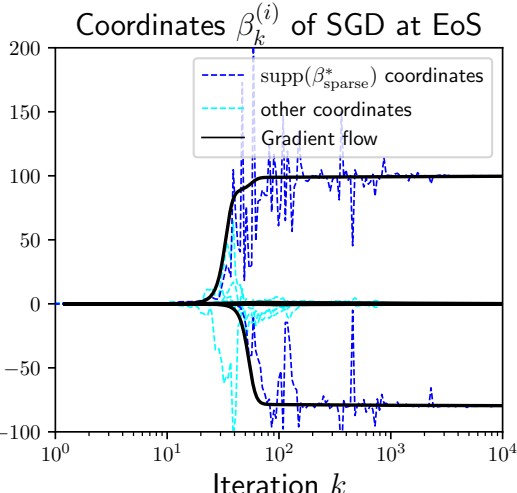

Figure 6: SGD at the edge of stability: all coordinates fluctuate, and the sparse solution is recovered. As opposed to GD at the EoS, since all coordinates fluctuate, the coordinates to recover are not more penalised than the others.

# B   Main ingredients behind the proof of Theorem 1 and Theorem 2

In this section, we show that the iterates $(\beta_k)_{k \geqslant 0}$ follow a *stochastic mirror descent* with *varying potentials*. At the core of our analysis, this result enables us to *(i)* prove convergence of the iterates to an interpolator and *(ii)* completely characterise the inductive bias of the algorithm (SGD or GD). Unveiling a mirror-descent like structure to characterise the implicit bias of a gradient method is classical. For gradient flow over diagonal linear networks [61], the iterates follow a mirror flow with respect to the hypentropy (4) with parameter $\alpha$ the initialisation scale, while for stochastic gradient flow [48] the mirror flow has a continuously evolving potential.

## B.1   Mirror descent and varying potentials

We recall that for a strictly convex reference function $h : \mathbb{R}^d \to \mathbb{R}$, the (stochastic) mirror descent iterates algorithm write as [5, 18], where the minimum is assumed to be attained over $\mathbb{R}^d$ and unique:

$$\beta_{k+1} = \operatorname{argmin}_{\beta \in \mathbb{R}^d} \left\{ \eta_k \langle g_k, \beta \rangle + D_h(\beta, \beta_k) \right\}, \tag{16}$$

for stochastic gradients $g_k$, stepsize $\gamma_k \geqslant 0$, and $D_h(\beta, \beta') = h(\beta) - h(\beta') - \langle \nabla h(\beta'), \beta - \beta' \rangle$ is the Bregman divergence associated to $h$. Iteration (16) can also be cast as

$$\nabla h(\beta_{k+1}) = \nabla h(\beta_k) - \gamma_k g_k. \tag{17}$$

Now, let $(h_k)$ be strictly convex reference functions $\mathbb{R}^d \to \mathbb{R}$. Whilst in continuous time, there is only one natural way to extend mirror flow to varying potentials, in discrete time the varying potentials can be incorporated in (16) (replacing $h$ by $h_k$ and leading to $\nabla h_k(\beta_{k+1}) = \nabla h_k(\beta_k) - \gamma_k g_k$), the mirror descent with varying potentials we study in this paper incorporates $h_{k+1}$ and $h_k$ in (17). The iterates are thus defined as through:

$$\beta_{k+1} = \operatorname{argmin}_{\beta \in \mathbb{R}^d} \left\{ \eta_k \langle g_k, \beta \rangle + D_{h_{k+1}, h_k}(\beta, \beta_k) \right\},$$

where $D_{h_{k+1}, h_k}(\beta, \beta') = h_{k+1}(\beta) - h_k(\beta') - \langle \nabla h_k(\beta'), \beta - \beta' \rangle$, a recursion that can also be cast as:

$$\nabla h_{k+1}(\beta_{k+1}) = \nabla h_k(\beta_k) - \gamma_k g_k.$$

To derive convergence of the iterates, we prove analogs to classical mirror descent lemmas, generalised to time-varying potentials.

## B.2   The iterates $(\beta_k)$ follow a stochastic mirror descent with varying potential recursion

In this section we show and prove that the iterates $(\beta_k)_k$ follow a stochastic mirror descent with varying potentials. Before stating the proposition, we recall the definition of the potentials. To do so we introduce several quantities.

Let $q, q_\pm : \mathbb{R} \to \mathbb{R} \cup \{\infty\}$ be defined as:

$$q_\pm(x) = \mp 2x - \ln\left((1 \mp x)^2\right),$$

$$q(x) = \frac{1}{2}(q_+(x) + q_-(x)) = -\frac{1}{2}\ln\left((1 - x^2)^2\right),$$

with the convention that $q(1) = \infty$. Notice that $q(x) \geqslant 0$ for $|x| \leqslant \sqrt{2}$ and $q(x) < 0$ otherwise. For the iterates $\beta_k = u_k \odot v_k \in \mathbb{R}^d$, we recall the definition of the following quantities:

$$\boldsymbol{\alpha}_{\pm,k} = \boldsymbol{\alpha} \exp\left(-\frac{1}{2} \sum_{i=0}^{k-1} q_\pm(\gamma_\ell \nabla \mathcal{L}_{\mathcal{B}_\ell}(\beta_\ell))\right) \in \mathbb{R}^d_{>0},$$

$$\boldsymbol{\alpha}_k^2 = \boldsymbol{\alpha}_{+,k} \odot \boldsymbol{\alpha}_{-,k},$$

$$\phi_k = \frac{1}{2} \operatorname{arcsinh}\left(\frac{\boldsymbol{\alpha}_{+,k}^2 - \boldsymbol{\alpha}_{-,k}^2}{2\boldsymbol{\alpha}_k^2}\right) \in \mathbb{R}^d.$$

Finally for $k \geqslant 0$, we define the potentials $(h_k : \mathbb{R}^d \to \mathbb{R})_{k \geqslant 0}$ as:

$$h_k(\beta) = \psi_{\boldsymbol{\alpha}_k}(\beta) - \langle \phi_k, \beta \rangle, \tag{18}$$

where $\psi_{\boldsymbol{\alpha}_k}$ is the hyperbolic entropy defined in (4) of scale $\boldsymbol{\alpha}_k$:

$$\psi_{\boldsymbol{\alpha}_k}(\beta) = \frac{1}{2} \sum_{i=1}^{d} \left( \beta_i \mathrm{arcsinh}(\frac{\beta_i}{\alpha_{k,i}^2}) - \sqrt{\beta_i^2 + \alpha_{k,i}^4} + \alpha_{k,i}^2 \right)$$

where $\alpha_{k,i}$ corresponds to the $i^{th}$ coordinate of the vector $\boldsymbol{\alpha}_k$.

Now that all the relevant quantities are define, we can state the following proposition which explicits the time-varying stochastic mirror descent followed by $(\beta_k)_k$

**Proposition 5.** *The iterates* $(\beta_k = u_k \odot v_k)_{k \geqslant 0}$ *from Eq.* (3) *satisfy the Stochastic Mirror Descent recursion with varying potentials* $(h_k)_k$:

$$\nabla h_{k+1}(\beta_{k+1}) = \nabla h_k(\beta_k) - \gamma_k \nabla \mathcal{L}_{\mathcal{B}_k}(\beta_k), \tag{19}$$

*where* $h_k : \mathbb{R}^d \to \mathbb{R}$ *for* $k \geqslant 0$ *are defined Eq.* (18)*. Since* $\nabla h_0(\beta_0) = 0$ *we have:*

$$\nabla h_k(\beta_k) \in \mathrm{span}(x_1, \ldots, x_n)$$

*Proof.* Using Proposition 6, we study the $\frac{1}{2}(w_+^2 - w_-^2)$ parametrisation instead of the $u \odot v$, indeed this is the natural parametrisation to consider when doing the calculations as it "separates" the recursions on $w_+$ and $w_-$.

Let us focus on the recursion of $w_+$:

$$w_{+,k+1} = (1 - \gamma_k \nabla \mathcal{L}_{\mathcal{B}_k}(\beta_k)) \cdot w_{+,k} .$$

We have:

$$\begin{aligned} w_{+,k+1}^2 &= (1 - \gamma_k \nabla \mathcal{L}_{\mathcal{B}_k}(\beta_k))^2 \cdot w_{+,k}^2 \\ &= \exp\left(\ln((1 - \gamma_k \nabla \mathcal{L}_{\mathcal{B}_k}(\beta_k))^2)\right) \cdot w_{+,k}^2 , \end{aligned}$$

with the convention that $\exp(\ln(0)) = 0$. This leads to:

$$\begin{aligned} w_{+,k+1}^2 &= \exp\left(-2\gamma_k \nabla \mathcal{L}_{\mathcal{B}_k}(w_k) + 2\gamma_k \nabla \mathcal{L}_{\mathcal{B}_k}(\beta_k) + \ln((1 - \gamma_k \nabla \mathcal{L}_{\mathcal{B}_k}(\beta_k))^2)\right) \cdot w_{+,k}^2 \\ &= \exp\left(-2\gamma_k \nabla \mathcal{L}_{\mathcal{B}_k}(\beta_k) - q_+(\gamma_k \nabla \mathcal{L}_{\mathcal{B}_k}(\beta_k))\right) \cdot w_{+,k}^2 , \end{aligned}$$

since $q_+(x) = -2x - \ln((1-x)^2)$. Expanding the recursion and using that $w_{+,k=0}$ is initialised at $w_{+,k=0} = \boldsymbol{\alpha}$, we thus obtain:

$$\begin{aligned} w_{+,k}^2 &= \boldsymbol{\alpha}^2 \exp(-\sum_{\ell=0}^{k-1} q_+(\gamma_\ell \nabla \mathcal{L}_{\mathcal{B}_\ell}(\beta_\ell))) \exp\left(-2\sum_{\ell=0}^{k-1} \gamma_\ell \nabla \mathcal{L}_{\mathcal{B}_\ell}(\beta_\ell)\right) \\ &= \boldsymbol{\alpha}_{+,k}^2 \exp\left(-2\sum_{\ell=0}^{k-1} \gamma_\ell \nabla \mathcal{L}_{\mathcal{B}_\ell}(\beta_\ell)\right), \end{aligned}$$

where we recall that $\boldsymbol{\alpha}_{\pm,k}^2 = \boldsymbol{\alpha}^2 \exp(-\sum_{\ell=0}^{k-1} q_\pm(\gamma_\ell g_\ell))$. One can easily check that we similarly get:

$$w_{-,k}^2 = \boldsymbol{\alpha}_{-,k}^2 \exp\left(+2\sum_{\ell=0}^{k-1} \gamma_\ell \nabla \mathcal{L}_{\mathcal{B}_\ell}(\beta_\ell)\right),$$

leading to:

$$\begin{aligned} \beta_k &= \frac{1}{2}(w_{+,k}^2 - w_{-,k}^2) \\ &= \frac{1}{2}\boldsymbol{\alpha}_{+,k}^2 \exp\left(-2\sum_{\ell=0}^{k-1} \gamma_\ell \nabla \mathcal{L}_{\mathcal{B}_\ell}(\beta_\ell)\right) - \frac{1}{2}\boldsymbol{\alpha}_{-,k}^2 \exp\left(+2\sum_{\ell=0}^{k-1} \gamma_\ell \nabla \mathcal{L}_{\mathcal{B}_\ell}(\beta_\ell)\right). \end{aligned}$$

Using Lemma 4, the previous equation can be simplified into:

$$\beta_k = \boldsymbol{\alpha}_{+,k}\boldsymbol{\alpha}_{-,k} \sinh\left(-2\sum_{\ell=0}^{k-1} \gamma_\ell \nabla \mathcal{L}_{\mathcal{B}_\ell}(\beta_\ell) + \mathrm{arcsinh}\left(\frac{\boldsymbol{\alpha}_{+,k}^2 - \boldsymbol{\alpha}_{-,k}^2}{2\boldsymbol{\alpha}_{+,k}\boldsymbol{\alpha}_{-,k}}\right)\right),$$

which writes as:

$$\frac{1}{2}\operatorname{arcsinh}\Big(\frac{\beta_k}{\boldsymbol{\alpha}_k^2}\Big) - \phi_k = -\sum_{\ell=0}^{k-1}\gamma_\ell\nabla\mathcal{L}_{\mathcal{B}_\ell}(\beta_\ell) \in \operatorname{span}(x_1,\dots,x_n)\,,$$

where $\phi_k = \frac{1}{2}\operatorname{arcsinh}\big(\frac{\boldsymbol{\alpha}_{+,k}^2 - \boldsymbol{\alpha}_{-,k}^2}{2\boldsymbol{\alpha}_k^2}\big)$, $\boldsymbol{\alpha}_k^2 = \boldsymbol{\alpha}_{+,k}\odot\boldsymbol{\alpha}_{-,k}$ and since the potentials $h_k$ are defined in Eq. (18) as $h_k = \psi_{\boldsymbol{\alpha}_k} - \langle\phi_k,\cdot\rangle$ with

$$\psi_{\boldsymbol{\alpha}}(\beta) = \frac{1}{2}\sum_{i=1}^{d}\Big(\beta_i\operatorname{arcsinh}(\frac{\beta_i}{\boldsymbol{\alpha}_i^2}) - \sqrt{\beta_i^2 + \boldsymbol{\alpha}_i^4} + \boldsymbol{\alpha}_i^2\Big) \tag{20}$$

specifically such that $\nabla h_k(\beta_k) = \frac{1}{2}\operatorname{arcsinh}\big(\frac{\beta_k}{\boldsymbol{\alpha}_k^2}\big) - \phi_k$. Hence,

$$\nabla h_k(\beta_k) = \sum_{\ell<k}\gamma_\ell\nabla\mathcal{L}_{\mathcal{B}_\ell}(\beta_\ell)\,,$$

so that:

$$\nabla h_{k+1}(\beta_{k+1}) = \nabla h_k(\beta_k) - \gamma_k\nabla\mathcal{L}_{\mathcal{B}_k}(\beta_k)\,,$$

which corresponds to a Mirror Descent with varying potentials $(h_k)_k$. $\qquad\square$

# C Equivalence of the $u\odot v$ and $\frac{1}{2}(w_+^2 - w_-^2)$ parametrisations

We here prove the equivalence between the $\frac{1}{2}(w_+^2 - w_-^2)$ and $u\odot v$ parametrisations, **that we use throughout the proofs in the Appendix.**

**Proposition 6.** *Let $(\beta_k)_{k\geqslant 0}$ and $(\beta_k')_{k\geqslant 0}$ be respectively generated by stochastic gradient descent on the $u\odot v$ and $\frac{1}{2}(w_+^2 - w_-^2)$ parametrisations:*

$$(u_{k+1},v_{k+1}) = (u_k,v_k) - \gamma_k\nabla_{u,v}\big(\mathcal{L}_{\mathcal{B}_k}(u\odot v)\big)(u_k,v_k)\,,$$

*and*

$$w_{\pm,k+1} = w_{\pm,k} - \gamma_k\nabla_{w_\pm}\big(\mathcal{L}_{\mathcal{B}_k}(\frac{1}{2}(w_+^2 - w_-^2))\big)(w_{+,k},w_{-,k})\,,$$

*initialised as $u_0 = \sqrt{2}\boldsymbol{\alpha}, v_0 = 0$ and $w_{+,0} = w_{-,0} = \boldsymbol{\alpha}$. Then for all $k\geqslant 0$, we have $\beta_k = \beta_k'$.*

*Proof.* We have:

$$w_{\pm,0} = \boldsymbol{\alpha}\,, \quad w_{\pm,k+1} = (1\mp\gamma_k\nabla\mathcal{L}_{\mathcal{B}_k}(\beta_k'))w_{\pm,k}\,,$$

and

$$u_0 = \sqrt{2}\boldsymbol{\alpha}\,, \quad v_0 = 0\,, \quad u_{k+1} = u_k - \gamma_k\nabla\mathcal{L}_{\mathcal{B}_k}(\beta_k)v_k\,, \quad v_{k+1} = v_k - \gamma_k\nabla\mathcal{L}(\beta_k)u_k\,.$$

Hence,

$$\beta_{k+1} = (1 + \gamma_k^2\nabla\mathcal{L}(\beta_k)^2)\beta_k - \gamma_k(u_k^2 + v_k^2)\nabla\mathcal{L}_{\mathcal{B}_k}(\beta_k)\,,$$

and

$$\beta_{k+1}' = (1 + \gamma_k^2\nabla\mathcal{L}_{\mathcal{B}_k}(\beta_k')^2)\beta_k' - \gamma_k(w_{+,k}^2 + w_{-,k}^2)\nabla\mathcal{L}_{\mathcal{B}_k}(\beta_k')\,.$$

Then, let $z_k = \frac{1}{2}(u_k^2 - v_k^2)$ and $z_k' = w_{+,k}w_{-,k}$. We have $z_0 = \boldsymbol{\alpha}^2$, $z_0' = \boldsymbol{\alpha}^2$ and:

$$z_{k+1} = (1 - \gamma_k^2\nabla\mathcal{L}_{\mathcal{B}_k}(\beta_k)^2)z_k\,, \quad z_{k+1}' = (1 - \gamma_k^2\nabla\mathcal{L}_{\mathcal{B}_k}(\beta_k')^2)z_k'\,.$$

Using $a^2 + b^2 = \sqrt{(2ab)^2 + (a^2 - b^2)^2}$ for $a,b\in\mathbb{R}$, we finally obtain that:

$$u_k^2 + v_k^2 = \sqrt{(2\beta_k)^2 + (2z_k)^2}\,, \quad w_{+,k}^2 + w_{-,k}^2 = \sqrt{(2\beta_k')^2 + (2z_k')^2}\,.$$

We conclude by observing that $(\beta_k, z_k)$ and $(\beta_k', z_k')$ follow the exact same recursions, initialised at the same value $(0, \boldsymbol{\alpha}^2)$.

$\qquad\square$

# D  Convergence of $\psi_\alpha$ to a weighted $\ell_1$ norm and harmful behaviour

We show that when taking the scale of the initialisation to $0$, one must be careful in the characterisation of the limiting norm, indeed if each entry does not go to zero "at the same speed", then the limit norm is a **weighted $\ell_1$-norm** rather than the classical $\ell_1$ norm.

**Proposition 7.** *For $\alpha \geqslant 0$ and a vector $h \in \mathbb{R}^d$, let $\tilde\alpha = \alpha \exp(-h \ln(1/\alpha)) \in \mathbb{R}^d$. Then we have that for all $\beta \in \mathbb{R}^d$*

$$\psi_{\tilde\alpha}(\beta) \underset{\alpha \to 0}{\sim} \ln(\frac{1}{\alpha}) \cdot \sum_{i=1}^{d}(1 + h_i)|\beta_i|.$$

*Proof.* Recall that

$$\psi_{\tilde\alpha}(\beta) = \frac{1}{2}\sum_{i=1}^{d}\left(\beta_i \mathrm{arcsinh}(\frac{\beta_i}{\tilde\alpha_i^2}) - \sqrt{\beta_i^2 + \tilde\alpha_i^4} + \tilde\alpha_i^2\right)$$

Using that $\mathrm{arcsinh}(x) \underset{|x| \to \infty}{\sim} \mathrm{sgn}(x)\ln(|x|)$, and that $\ln(\frac{1}{\tilde\alpha_i^2}) = (1 + h_i)\ln(\frac{1}{\alpha^2})$ we obtain that

$$\psi_{\tilde\alpha}(\beta) \underset{\alpha \to 0}{\sim} \frac{1}{2}\sum_{i=1}^{d}\mathrm{sgn}(\beta_i)\beta_i(1 + h_i)\ln(\frac{1}{\alpha^2})$$

$$= \frac{1}{2}\ln(\frac{1}{\alpha^2})\sum_{i=1}^{d}(1 + h_i)|\beta_i|.$$

$\square$

The following Fig. 7 illustrates the effect of the non-uniform shape $\boldsymbol\alpha$ on the corresponding potential $\psi_{\boldsymbol\alpha}$.

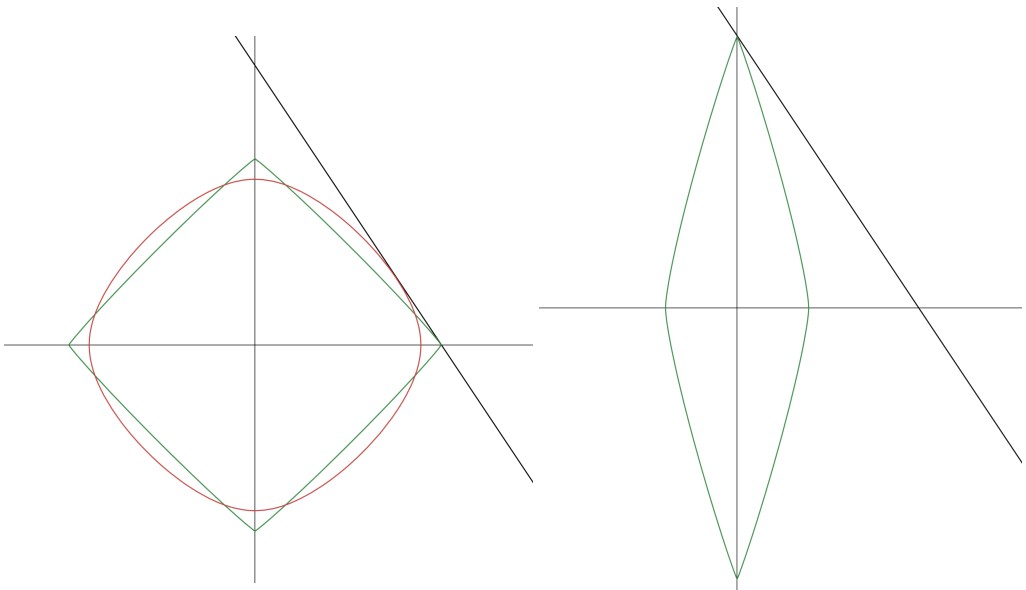

Figure 7: *Left*: Uniform $\boldsymbol\alpha = \alpha\mathbf{1}$: a smaller scale $\alpha$ leads to the potential $\psi_\alpha$ being closer to the $\ell_1$-norm. *Right*: A non uniform $\boldsymbol\alpha$ can lead to the recovery of a solution which is very far from the minimum $\ell_1$-norm solution. The affine line corresponds to the set of interpolators when $n = 1$, $d = 2$ and $s = 1$.

More generally, for $\alpha$ such that $\alpha_i \to 0$ for all $i \in [d]$ at rates such that $\ln(1/\alpha_i) \sim q_i\ln(1/\max_i \alpha_i)$, we retrieve a weighted $\ell_1$ norm:

$$\frac{\psi_\alpha(\beta)}{\ln(1/\alpha^2)} \to \sum_{i=1}^{d}q_i|\beta_i|.$$

Hence, even for arbitrary small $\max_i \alpha_i$, if the *shape* of $\alpha$ is 'bad', the interpolator $\beta_\alpha$ that minimizes $\psi_\alpha$ can be arbitrary far away from $\beta_{\ell 1}^\star$ the interpolator of minimal $\ell_1$ norm.

We illustrate the importance of the previous proposition in the following example.

**Example 1.** *We illustrate how, even for arbitrary small $\max_i \alpha_i$, the interpolator $\beta_\alpha^\star$ that minimizes $\psi_\alpha$ can be far from the minimum $\ell_1$ norm solution, due to the shape of $\boldsymbol{\alpha}$ that is not uniform. The message of this example is that for $\boldsymbol{\alpha} \to 0$ non-uniformly across coordinates, if the coordinates of $\alpha$ that go slowly to $0$ coincide with the non-null coordinates of the sparse interpolator we want to retrieve, then $\beta_\alpha^\star$ will be far from the sparse solution.*

*A simple counterexample can be built: let $\beta_{\mathrm{sparse}}^\star = (1, \ldots, 1, 0, \ldots, 0)$ (with only the $s = o(d)$ first coordinates that are non-null), and let $(x_i)$, $(y_i)$ be generated as $y_i = \langle \beta_{\mathrm{sparse}}^\star, x_i \rangle$ with $x_i \sim \mathcal{N}(0, 1)$. For $n$ large enough ($n$ of order $s \ln(d)$ where $s$ is the sparsity), the design matrix $X$ is RIP [10], so that the minimum $\ell_1$ norm interpolator $\beta_{\ell 1}^\star$ is exactly equal to $\beta_{\mathrm{sparse}}^\star$.*

*However, if $\alpha$ is such that $\max_i \alpha_i \to 0$ with $h_i >> 1$ for $j \leqslant s$ and $h_i = 1$ for $i \geqslant s + 1$ ($h_i$ as in Proposition 7), $\beta_\alpha^\star$ will be forced to verify $\beta_{\alpha,i}^\star = 0$ for $i \leqslant s$ and hence $\|\beta_{\alpha,1}^\star - \beta_{\ell 1}^\star\|_1 \geqslant s$.*

## E  Main descent lemma and boundedness of the iterates

The goal of this section is to prove the following proposition, our main descent lemma: for well-chosen stepsizes, the Bregman divergences $(D_{h_k}(\beta^\star, \beta_k))_{k \geqslant 0}$ decrease. We then use this proposition to bound the iterates for both SGD and GD.

**Proposition 8.** *There exist a constant $c > 0$ and $B > 0$ such that $B = \mathcal{O}(\inf_{\beta^\star \in \mathcal{S}} \|\beta^\star\|_\infty)$ for GD and $B = \mathcal{O}(\ln(1/\alpha) \inf_{\beta^\star \in \mathcal{S}} \|\beta^\star\|_\infty)$ for SGD, such that if $\gamma_k \leqslant \frac{c}{LB}$ for all $k$, then we have, for all $k \geqslant 0$ and any interpolator $\beta^\star \in \mathcal{S}$:*

$$D_{h_{k+1}}(\beta^\star, \beta_{k+1}) \leqslant D_{h_k}(\beta^\star, \beta_k) - \gamma_k \mathcal{L}_{\mathcal{B}_k}(\beta_k) \,.$$

To prove this result, we first provide a general descent lemma for time-varying mirror descent (Proposition 9, appendix E.1), before proving the proposition for fixed iteration $k$ and bound $B > 0$ on the iterates infinity norm in Appendix E.2 (Proposition 10). We finally use this to prove a bound on the iterates infinity norm in appendix E.3.

### E.1  Descent lemma for (stochastic) mirror descent with varying potentials

In the following we adapt a classical mirror descent equality but for time varying potentials, that differentiates from Orabona et al. [47] in that it enables us to prove the decrease of the Bregman divergences of the iterates. Moreover, as for classical MD, it is an equality.

**Proposition 9.** *For $h, g : \mathbb{R}^d \to \mathbb{R}$ functions, let $D_{h,g}(\beta, \beta') = h(\beta) - g(\beta') - \langle \nabla g(\beta'), \beta - \beta' \rangle$[4] for $\beta, \beta' \in \mathbb{R}^d$. Let $(h_k)$ strictly convex functions defined $\mathbb{R}^d$ $\mathcal{L}$ a convex function defined on $\mathbb{R}^d$. Let $(\beta_k)$ defined recursively through $\beta_0 \in \mathbb{R}^d$, and*

$$\beta_{k+1} \in \mathrm{argmin}_{\beta \in \mathbb{R}^d} \left\{ \gamma_k \langle \nabla \mathcal{L}(\beta_k), \beta - \beta_k \rangle + D_{h_{k+1}, h_k}(\beta, \beta_k) \right\},$$

*where we assume that the minimum is unique and attained in $\mathbb{R}^d$. Then, $(\beta_k)$ satisfies*

$$\nabla h_{k+1}(\beta_{k+1}) = \nabla h_k(\beta_k) - \gamma_k \nabla \mathcal{L}(\beta_k) \,,$$

*and for any $\beta \in \mathbb{R}^d$,*

$$D_{h_{k+1}}(\beta, \beta_{k+1}) = D_{h_k}(\beta, \beta_k) - \gamma_k \langle \nabla \mathcal{L}(\beta_k), \beta_k - \beta \rangle + D_{h_{k+1}}(\beta_k, \beta_{k+1})$$
$$- \left(h_{k+1} - h_k\right)(\beta_k) + \left(h_{k+1} - h_k\right)(\beta) \,.$$

*Proof.* Let $\beta \in \mathbb{R}^d$. Since we assume that the minimum through which $\beta_{k+1}$ is computed is attained in $\mathbb{R}^d$, the gradient of the function $V_k(\beta) = \gamma_k \langle \nabla \mathcal{L}(\beta_k), \beta - \beta_k \rangle + D_{h_{k+1}, h_k}(\beta, \beta_k)$ evaluated at $\beta_{k+1}$ is null, leading to $\nabla h_{k+1}(\beta_{k+1}) = \nabla h_k(\beta_k) - \gamma_k \nabla \mathcal{L}(\beta_k)$.

---

[4]for $h = g$, we recover the classical Bregman divergence that we denote $D_h = D_{h,h}$

Then, since $\nabla V_k(\beta_{k+1}) = 0$, we have $D_{V_k}(\beta, \beta_{k+1}) = V_k(\beta) - V_k(\beta_{k+1})$. Using $\nabla^2 V_k = \nabla^2 h_{k+1}$, we also have $D_{V_k} = D_{h_{k+1}}$. Hence:

$$D_{h_{k+1}}(\beta, \beta_{k+1}) = \gamma_k \langle \nabla \mathcal{L}(\beta_k), \beta - \beta_{k+1} \rangle + D_{h_{k+1}, h_k}(\beta, \beta_k) - D_{h_{k+1}, h_k}(\beta_{k+1}, \beta_k).$$

We write $\gamma_k \langle \nabla \mathcal{L}(\beta_k), \beta - \beta_{k+1} \rangle = \gamma_k \langle \nabla \mathcal{L}(\beta_k), \beta - \beta^k \rangle + \gamma_k \langle \nabla \mathcal{L}(\beta_k), \beta_k - \beta_{k+1} \rangle$. We also have $\gamma_k \langle \nabla \mathcal{L}(\beta_k), \beta_k - \beta_{k+1} \rangle = \langle \nabla h_k(\beta_k) - \nabla h_{k+1}(\beta_{k+1}), \beta_k - \beta_{k+1} \rangle = D_{h_k, h_{k+1}}(\beta_k, \beta_{k+1}) + D_{h_{k+1}, h_k}(\beta_{k+1}, \beta^k)$, so that $\gamma_k \langle \nabla \mathcal{L}(\beta_k), \beta_k - \beta_{k+1} \rangle - D_{h_{k+1}, h_k}(\beta_{k+1}, \beta^k) = D_{h_k, h_{k+1}}(\beta_k, \beta_{k+1})$. Thus,

$$D_{h_{k+1}}(\beta, \beta_{k+1}) = D_{h_{k+1}, h_k}(\beta, \beta_k) - \gamma_k \big( D_f(\beta, \beta_k) + D_f(\beta_k, \beta) \big) + D_{h_k, h_{k+1}}(\beta_k, \beta_{k+1}),$$

and writing $D_{h,g}(\beta, \beta') = D_g(\beta, \beta') + h(\beta) - g(\beta)$ concludes the proof. $\qquad\square$

## E.2   Proof of Proposition 10

In next proposition, we use Proposition 9 to prove our main descent lemma. To that end, we bound the error terms that appear in Proposition 9 as functions of $\mathcal{L}_{\mathcal{B}_k}(\beta_k)$ and norms of $\beta_k, \beta_{k+1}$, so that for explicit stepsizes, the error terms can be cancelled by half of the negative quantity $-2\mathcal{L}_{\mathcal{B}_k}(\beta_k)$.

**Additional notation:** let $L_2, L_\infty > 0$ such that $\forall \beta, \|H_{\mathcal{B}}\beta\|_2 \leqslant L\|\beta\|_2, \|H_{\mathcal{B}}\beta\|_\infty \leqslant L\|\beta\|_\infty$ for all batches $\mathcal{B} \subset [n]$ of size $b$.

**Proposition 10.** *Let $k \geqslant 0$ and $B > 0$. Provided that $\|\beta_k\|_\infty, \|\beta_{k+1}\|_\infty, \|\beta^\star\|_\infty \leqslant B$ and $\gamma_k \leqslant \frac{c}{LB}$ where $c > 0$ is some numerical constant, we have:*

$$D_{h_{k+1}}(\beta^\star, \beta_{k+1}) \leqslant D_{h_k}(\beta^\star, \beta_k) - \gamma_k \mathcal{L}_{\mathcal{B}_k}(\beta_k). \tag{21}$$

*Proof.* Let $\beta^\star \in \mathcal{S}$ be any interpolator. From Proposition 9:

$$D_{h_{k+1}}(\beta^\star, \beta_{k+1}) = D_{h_k}(\beta^\star, \beta_k) - 2\gamma_k \mathcal{L}_{\mathcal{B}_k}(\beta_k) + D_{h_{k+1}}(\beta_{k+1}, \beta_k) - (h_{k+1} - h_k)(\beta_k) + (h_{k+1} - h_k)(\beta^\star).$$

We want to bound the last three terms of this equality. First, to bound the last two we apply Lemma 7 assuming that $\|\beta^\star\|_\infty, \|\beta_{k+1}\|_\infty \leqslant B$:

$$-(h_{k+1} - h_k)(\beta_k) + (h_{k+1} - h_k)(\beta^\star) \leqslant 24BL_2\gamma_k^2 \mathcal{L}_{\mathcal{B}_k}(\beta_k)$$

We now bound $D_{h_{k+1}}(\beta_k, \beta_{k+1})$. Classical Bregman manipulations provide that

$$D_{h_{k+1}}(\beta_k, \beta_{k+1}) = D_{h_{k+1}^*}(\nabla h_{k+1}(\beta_{k+1}), \nabla h_{k+1}(\beta_k))$$
$$= D_{h_{k+1}^*}(\nabla h_k(\beta^k) - \gamma_k \nabla \mathcal{L}_{\mathcal{B}_k}(\beta_k), \nabla h_{k+1}(\beta_k)).$$

From Lemma 6 we have that $h_{k+1}$ is $\min(1/(4\alpha_{k+1}^2), 1/(4B))$ strongly convex on the $\ell^\infty$-centered ball of radius $B$ therefore $h_{k+1}^*$ is $\max(4\alpha_{k+1}^2, 4B) = 4B$ (for $\alpha$ small enough or $B$ big enough) smooth on this ball, leading to:

$$D_{h_{k+1}}(\beta_k, \beta_{k+1}) \leqslant 2B\|\nabla h_k(\beta_k) - \gamma_k \nabla \mathcal{L}_{\mathcal{B}_k}(\beta_k) - \nabla h_{k+1}(\beta_k)\|_2^2$$
$$\leqslant 4B\big(\|\nabla h_k(\beta_k) - \nabla h_{k+1}(\beta_k)\|_2^2 + \|\gamma_k \nabla \mathcal{L}_{\mathcal{B}_k}(\beta_k)\|_2^2\big).$$

Using $|\nabla h_k(\beta) - \nabla h_{k+1}(\beta)| \leqslant 2\delta_k$ where $\delta_k = q(\gamma_k \nabla \mathcal{L}_{\mathcal{B}_k}(\beta_k))$, we get that:

$$D_{h_{k+1}}(\beta_k, \beta_{k+1}) \leqslant 8B\|\delta_k\|_2^2 + 4BL\gamma_k^2 \mathcal{L}_{\mathcal{B}_k}(\beta_k).$$

Now, $\|\delta_k\|_2^2 \leqslant \|\delta_k\|_1 \|\delta_k\|_\infty$ and using Lemma 5, $\|\delta_k\|_1 \|\delta_k\|_\infty \leqslant 4\|\gamma_k \nabla \mathcal{L}_{\mathcal{B}_k}(\beta_k)\|_2^2 \|\gamma_k \nabla \mathcal{L}_{\mathcal{B}_k}(\beta_k)\|_\infty^2 \leqslant 2\|\gamma_k \nabla \mathcal{L}_{\mathcal{B}_k}(\beta_k)\|_2^2$ since $\|\gamma_k \nabla \mathcal{L}_{\mathcal{B}_k}(\beta_k)\|_\infty \leqslant \gamma_k L_\infty \|\beta_k - \beta_\infty\| \leqslant \gamma_k \times 2LB \leqslant 1/2$ is verified for $\gamma_k \leqslant 1/(4LB)$. Thus,

$$D_{h_{k+1}}(\beta_k, \beta_{k+1}) \leqslant 40BL_2\gamma_k^2 \mathcal{L}_{\mathcal{B}_k}(\beta_k).$$

Hence, provided that $\|\beta_k\|_\infty \leqslant B, \|\beta_{k+1}\|_\infty \leqslant B$ and $\gamma_k \leqslant 1/(4LB)$, we have:

$$D_{h_{k+1}}(\beta^\star, \beta_{k+1}) \leqslant D_{h_k}(\beta^\star, \beta_k) - 2\gamma_k \mathcal{L}_{\mathcal{B}_k}(\beta_k) + 64L_2\gamma_k^2 B\mathcal{L}_{\mathcal{B}_k}(\beta_k),$$

and thus

$$D_{h_{k+1}}(\beta^\star, \beta_{k+1}) \leqslant D_{h_k}(\beta^\star, \beta_k) - \gamma_k \mathcal{L}_{\mathcal{B}_k}(\beta_k).$$

if $\gamma_k \leqslant \frac{c}{BL}$, where $c = \frac{1}{64}$.

$\qquad\square$

## E.3 Bound on the iterates

We now bound the iterates $(\beta_k)$ by an explicit constant $B$ that depends on $\|\beta^\star\|_1$ (for any fixed $\beta^\star \in \mathcal{S}$).

The first bound we prove holds for both SGD and GD, and is of the form $\mathcal{O}(\|\beta^\star\|_1 \ln(1/\alpha^2))$ while the second bound, that holds only for GD ($b = n$) is of order $\mathcal{O}(\|\beta^\star\|_1)$ (independent of $\alpha$). While a bound independent of $\alpha$ is only proved for GD, we believe that such a result also holds for SGD, and in both cases $B$ should be thought of order $\mathcal{O}(\|\beta^\star\|_1)$.

### E.3.1 Bound that depends on $\alpha$ for GD and SGD

A consequence of Proposition 10 is the boundedness of the iterates, as shown in next corollary. Hence, Proposition 10 can be applied using $B$ a uniform bound on the iterates $\ell^\infty$ norm.

**Corollary 1.** *Let $B = 3\|\beta^\star\|_1 \ln\left(1 + \frac{\|\beta^\star\|_1}{\alpha^2}\right)$. For stepsizes $\gamma_k \leqslant \frac{c}{BL}$, we have $\|\beta_k\|_\infty \leqslant B$ for all $k \geqslant 0$.*

*Proof.* We proceed by induction. Let $k \geqslant 0$ such that $\|\beta_k\|_\infty \leqslant B$ for some $B > 0$ and $D_{h_k}(\beta^\star, \beta_k) \leqslant D_{h_0}(\beta^\star, \beta_0)$ (note that these two properties are verified for $k = 0$, since $\beta_0 = 0$). For $\gamma_k$ sufficiently small (*i.e.*, that satisfies $\gamma_k \leqslant \frac{c}{B'L}$ where $B' \geqslant \|\beta_{k+1}\|_\infty, \|\beta_k\|_\infty, \|\beta^\star\|_\infty$), using Proposition 10, we have $D_{h_{k+1}}(\beta^\star, \beta_{k+1}) \leqslant D_{h_k}(\beta^\star, \beta_k)$ so that $D_{h_{k+1}}(\beta^\star, \beta_{k+1}) \leqslant D_{h_0}(\beta^\star, \beta_0)$, which can be rewritten as:

$$\sum_{i=1}^d \alpha_{k+1,i}^2 \left(\sqrt{1 + (\frac{\beta_{k+1,i}}{\alpha_{k+1,i}^2})^2} - 1\right) \leqslant \sum_{i=1}^d \beta_i^\star \operatorname{arcsinh}(\frac{\beta_{k+1,i}}{\alpha^2}).$$

Hence, $\|\beta_{k+1}\|_1 \leqslant \|\beta^\star\|_1 \ln(1 + \frac{\|\beta_{k+1}\|_1}{\alpha^2})$. We then notice that for $x, y > 0$, $x \leqslant y \ln(1 + x) \implies x \leqslant 3y \ln(1 + y)$: if $x > y \ln(1 + y)$ and $x > y$, we have that $y \ln(1 + y) < y \ln(1 + x)$, so that $1 + y < 1 + x$, which contradicts our assumption. Hence, $x \leqslant \max(y, y \ln(1 + y))$. In our case, $x = \|\beta^{k+1}\|_1 / \alpha^2$, $y = \|\beta^\star\|_1 / \alpha^2$ so that for small alpha, $\ln(1 + y) \geqslant 1$.

Hence, we deduce that $\|\beta_{k+1}\|_1 \leqslant B$, where $B = \|\beta^\star\|_1 \ln(1 + \frac{\|\beta^\star\|_1}{\alpha^2})$.

This is true as long as $\gamma_k$ is tuned using $B'$ a bound on $\max(\|\beta_k\|_\infty, \|\beta_{k+1}\|_\infty)$. Using the continuity of $\beta_{k+1}$ as a function of $\gamma_k$ ($\beta_k$ being fixed), we show that $\gamma_k \leqslant \frac{1}{2} \times \frac{c}{BL}$ can be used using this $B$. Indeed, let $\phi : \mathbb{R}^+ \to \mathbb{R}^d$ be the function that takes as entry $\gamma_k \geqslant 0$ and outputs the corresponding $\|\beta_{k+1}\|_\infty$: $\phi$ is continuous. Let $\gamma_r = \frac{1}{2} \times \frac{c}{rL}$ for $r > 0$ and $\bar{r} = \sup\{r \geqslant 0 : B < \phi(\gamma_r)\}$ (the set is upper-bounded; if is is empty, we do not need what follows since it means that any stepsize leads to $\|\beta_{k+1}\|_\infty \leqslant B$). By continuity of $\phi$, $\phi(\gamma_{\bar{r}}) = B$. Furthermore, for all $r$ that satisfies $r \geqslant \max(\phi(\gamma_r), B) \geqslant \max(\phi(\gamma_r), \|\beta_k\|_\infty, \|\beta^\star\|_\infty)$, we have, using what is proved just above, that $\|\beta_{k+1}\|_\infty \leqslant B$ and thus $\phi(\gamma_r) \leqslant B$ for such a $r$:

**Lemma 1.** *For $r > 0$ such that $r \geqslant \max(\phi(\gamma_r), B)$, we have $\phi(\gamma_r) \leqslant B$.*

Now, if $\bar{r} > B$, by definition of $\bar{r}$ and by continuity of $\phi$, since $\phi(\bar{r}) = B$, there exists some $B < r < \bar{r}$ such that $\phi(\gamma_r) > B$ (definition of the supremum) and $\phi(\gamma_r) \leqslant 2B$ (continuity of $\phi$). This particular choice of $r$ thus satisfies $r > B$ and and $\phi(\gamma_r) \leqslant 2B \leqslant 2r$, leading to $\phi(\gamma_r) \leqslant B$, using Lemma 1, hence a contradiction: we thus have $\bar{r} \leqslant B$.

This concludes the induction: for all $r \geqslant B$, we have $r \geqslant \bar{r}$ so that $\phi(\gamma_r) \leqslant B$ and thus for all stepsizes $\gamma \leqslant \frac{c}{2LB}$, we have $\|\beta_{k+1}\|_\infty \leqslant B$.

$\square$

### E.3.2 Bound independent of $\alpha$

We here assume in this subsection that $b = n$. We prove that for gradient descent, the iterates are bounded by a constant that does not depend on $\alpha$.

**Proposition 11.** *Assume that $b = n$ (full batch setting). There exists some $B = \mathcal{O}(\|\beta^\star\|_1)$ such that for stepsizes $\gamma_k \leqslant \frac{c}{BL}$, we have $\|\beta_k\|_\infty \leqslant B$ for all $k \geqslant 0$.*

*Proof.* We first begin by proving the following proposition: for sufficiently small stepsizes, the loss values decrease. In the following lemma we provide a bound on the gradient descent iterates $(w_{+,k}, w_{-,k})$ which will be useful to show that the loss is decreasing.

**Proposition 12.** *For $\gamma_k \leqslant \frac{c}{LB}$ where $B \geqslant \max(\|\beta_k\|_\infty, \|\beta_{k+1}\|_\infty)$, we have $\mathcal{L}(\beta_{k+1}) \leqslant \mathcal{L}(\beta_k)$*

*Proof.* Oddly, using the time-varying mirror descent recursion is not the easiest way to show the decrease of the loss, due to the error terms which come up. Therefore to show that the loss is decreasing we use the gradient descent recursion. Recall that the iterates $w_k = (w_{+,k}, w_{-,k}) \in \mathbb{R}^{2d}$ follow a gradient descent on the non convex loss $F(w) = \frac{1}{2}\|y - \frac{1}{2}X(w_+^2 - w_-^2)\|_2$.

For $k \geqslant 0$, using the Taylor formula we have that $F(w_{k+1}) \leqslant F(w_k) - \gamma_k(1 - \frac{\gamma_k L_k}{2})\|\nabla F(w_k)\|^2$ with the local smoothness $L_k = \sup_{w \in [w_k, w_{k+1}]} \lambda_{\max}(\nabla^2 F(w))$. Hence if $\gamma_k \leqslant 1/L_k$ for all $k$ we get that the loss is non-increasing. We now bound $L_k$. Computing the hessian ot $F$, we obtain that:

$$
\nabla^2 F(w_k) = \begin{pmatrix} \mathrm{diag}(\nabla\mathcal{L}(\beta_k)) & 0 \\ 0 & -\mathrm{diag}(\nabla\mathcal{L}(\beta_k)) \end{pmatrix}
$$
$$
+ \begin{pmatrix} \mathrm{diag}(w_{+,k})H\,\mathrm{diag}(w_{+,k}) & -\mathrm{diag}(w_{-,k})H\,\mathrm{diag}(w_{+,k}) \\ -\mathrm{diag}(w_{+,k})H\,\mathrm{diag}(w_{-,k}) & \mathrm{diag}(w_{-,k})H\,\mathrm{diag}(w_{-,k}) \end{pmatrix}. \tag{22}
$$

Let us denote by $M = \begin{pmatrix} M_+ & M_{+,-} \\ M_{+,-} & M_- \end{pmatrix} \in \mathbb{R}^{2d \times 2d}$ the second matrix in the previous equality. With this notation $\|\nabla^2 F(w_k)\| \leqslant \|\nabla\mathcal{L}(\beta_k)\|_\infty + 2\|M\|$ (where the norm corresponds to the Schatten 2-norm which is the largest eigenvalue for symmetric matrices). Now, notice that:

$$
\|M\|^2 = \sup_{u \in \mathbb{R}^{2d}, \|u\|=1} \|Mu\|^2
$$
$$
= \sup_{\substack{u_+ \in \mathbb{R}^d, \|u_+\|=1 \\ u_- \in \mathbb{R}^d, \|u_-\|=1 \\ (a,b) \in \mathbb{R}^2, a^2+b^2=1}} \left\| M \begin{pmatrix} a \cdot u_+ \\ b \cdot u_- \end{pmatrix} \right\|^2.
$$

We have:

$$
\left\| M \begin{pmatrix} a \cdot u_+ \\ b \cdot u_- \end{pmatrix} \right\|^2 = \left\| \begin{pmatrix} aM_+u_+ + bM_{+-}u_- \\ aM_{+-}u_+ + bM_-u_- \end{pmatrix} \right\|^2
$$
$$
= \|aM_+u_+ + bM_{+-}u_-\|^2 + \|aM_{+-}u_+ + bM_-u_-\|^2
$$
$$
\leqslant 2\left( a^2\|M_+u_+\|^2 + b^2\|M_{+-}u_-\|^2 + a^2\|M_{+-}u_+\|^2 + b^2\|M_-u_-\|^2 \right)
$$
$$
\leqslant 2\left( \|M_+\|^2 + \|M_{+-}\|^2 + \|M_-\|^2 \right).
$$

Since $\|M_\pm\| \leqslant \lambda_{max} \cdot \|w_\pm\|_\infty^2$ and $\|M_{+-}\| \leqslant \lambda_{max}\|w_+\|_\infty\|w_-\|_\infty$ we finally get that

$$
\|M\|^2 \leqslant 6\lambda_{max}^2 \cdot \max(\|w_+\|_\infty^2, \|w_-\|_\infty^2)^2
$$
$$
\leqslant 6\lambda_{max}^2(\|w_+^2\|_\infty + \|w_-^2\|_\infty)^2
$$
$$
\leqslant 12\lambda_{max}^2\|w_+^2 + w_-^2\|_\infty^2.
$$

We now upper bound this quantity in the following lemma.

**Lemma 2.** *For all $k \geqslant 0$, the following inequality holds component-wise:*

$$
w_{+,k}^2 + w_{-,k}^2 = \sqrt{4\boldsymbol{\alpha}_k^4 + \beta_k^2}.
$$

*Proof.* Notice from the definition of $w_{+,k}$ and $w_{-,k}$ given in the proof of Proposition 5 that:

$$
|w_{+,k}||w_{-,k}| = \boldsymbol{\alpha}_{-,k}\boldsymbol{\alpha}_{+,k} = \boldsymbol{\alpha}_k^2. \tag{23}
$$

And $\boldsymbol{\alpha}_0 = \boldsymbol{\alpha}^2$. Now since $\boldsymbol{\alpha}_k$ is decreasing coordinate-wise (under our assumptions on the stepsizes, $\gamma_k^2 \nabla \mathcal{L}(\beta_k)^2 \leqslant (1/2)^2 < 1$), we get that.:

$$w_{+,k}^2 + w_{-,k}^2 = 2\sqrt{\boldsymbol{\alpha}_k^4 + \beta_k^2} \leqslant 2\sqrt{\boldsymbol{\alpha}^4 + \beta_k^2}$$

leading to $w_{+,k}^2 + w_{-,k}^2 \leqslant \sqrt{4\boldsymbol{\alpha}^4 + B^2}$. $\hfill \square$

From Lemma 2, $w_{+,k}^2 + w_{-,k}^2$ is bounded by $2\sqrt{\boldsymbol{\alpha}^4 + B^2}$. Putting things together we finally get that $\|\nabla^2 F(w)\| \leqslant \|\nabla \mathcal{L}(\beta)\|_\infty + 8\lambda_{max}\sqrt{4\|\boldsymbol{\alpha}\|_\infty^4 + B^2}$. Hence,

$$L_k \leqslant \sup_{\|\beta\|_\infty \leqslant B} \|\nabla \mathcal{L}(\beta)\|_\infty + 8\lambda_{\max}\sqrt{\|\boldsymbol{\alpha}\|_\infty^4 + B^2} \leqslant LB + 8\lambda_{\max}\sqrt{\|\boldsymbol{\alpha}\|_\infty^4 + B^2} \leqslant 10LB\,,$$

for $B \geqslant \|\boldsymbol{\alpha}\|_\infty^2$. $\hfill \square$

We finally prove the bound on $\|\beta_k\|_\infty$ independent of $\alpha$ for a uniform initialisation $\boldsymbol{\alpha} = \alpha\mathbf{1}$, using the monotonic property of $\mathcal{L}$.

**Proposition 13.** *Assume that $b = n$ (full batch setting). There exists some $B = \mathcal{O}(\|\beta^\star\|_1)$ such that for stepsizes $\gamma_k \leqslant \frac{c}{BL}$, we have $\|\beta_k\|_\infty \leqslant B$ for all $k \geqslant 0$.*

*Proof.* In this proof, we first let $B$ be a bound on the iterates. Tuning stepsizes using this bound, we prove that the iterates are bounded by a some $B' = \mathcal{O}(\|\beta^\star\|_1)$. Finally, we conclude by using the continuity of the iterates (at a finite horizon) that this explicit bound can be used to tune the stepsizes.

Writing the mirror descent with varying potentials, we have, since $\nabla h_0(\beta_0) = 0$,

$$\nabla h_k(\beta_k) = -\sum_{\ell < k} \gamma_\ell \nabla \mathcal{L}(\beta_\ell)\,,$$

leading to, by convexity of $h_k$:

$$h_k(\beta_k) - h_k(\beta^\star) \leqslant \langle \nabla h_k(\beta_k), \beta_k - \beta^\star \rangle = -\sum_{\ell < k} \langle \gamma_\ell \nabla \mathcal{L}(\beta_\ell), \beta_k - \beta^\star \rangle\,.$$

We then write, using $\nabla \mathcal{L}(\beta) = H(\beta - \beta^\star)$ for $H = XX^\top$, that $-\sum_{\ell < k}\langle \gamma_\ell \nabla \mathcal{L}(\beta_\ell), \beta_k - \beta^\star \rangle = -\sum_{\ell < k} \gamma_\ell \langle X^\top(\bar{\beta}_k - \beta^\star), X^\top(\beta_k - \beta^\star) \rangle \leqslant \sum_{\ell < k} \gamma_\ell \sqrt{\mathcal{L}(\bar{\beta}_k)\mathcal{L}(\beta_k)}$, leading to:

$$h_k(\beta_k) - h_k(\beta^\star) \leqslant 2\sqrt{\sum_{\ell < k}\gamma_\ell \mathcal{L}(\bar{\beta}_k) \sum_{\ell < k}\gamma_\ell \mathcal{L}(\beta_k)} \leqslant 2\sum_{\ell < k}\gamma_\ell \mathcal{L}(\bar{\beta}_k) \leqslant 2D_{h_0}(\beta^\star, \beta^0)\,,$$

where the last inequality holds provided that $\gamma_k \leqslant \frac{1}{CLB}$. Thus,

$$\psi_{\boldsymbol{\alpha}_k}(\beta_k) \leqslant \psi_{\boldsymbol{\alpha}_k}(\beta^\star) + 2\psi_{\boldsymbol{\alpha}_0}(\beta^\star) + \langle \phi_k, \beta_k - \beta^\star \rangle\,.$$

Then, $\langle \phi_k, \beta_k - \beta^\star \rangle \leqslant \|\phi_k\|_1 \|\beta_k - \beta^\star\|_\infty$ and $\|\phi_k\|_1 \leqslant C\lambda_{\max}\sum_{k < K}\gamma_k^2 \mathcal{L}(\beta^k) \leqslant C\lambda_{\max}\gamma_{\max}h_0(\beta^\star)$. Then, using

$$\|\beta\|_\infty - \frac{1}{\ln(1/\alpha^2)} \leqslant \frac{\psi_\alpha(\beta)}{\ln(1/\alpha^2)} \leqslant \|\beta\|_1\big(1 + \frac{\ln(\|\beta\|_1 + \alpha^2)}{\ln(1/\alpha^2)}\big)\,,$$

we have:

$$\begin{aligned}\|\beta_k\|_\infty &\leqslant \frac{1}{\ln(1/\alpha^2)} + \|\beta^\star\|_1\big(1 + \frac{\ln(\|\beta^\star\|_1 + \alpha^2)}{\ln(1/\alpha^2)}\big) + \|\beta^\star\|_1\big(1 + \frac{\ln(\|\beta^\star\|_1 + \alpha^2)}{\ln(1/\alpha^2)}\big) \\ &\quad + B_0 C\lambda_{\max}\gamma_{\max}h_0(\beta^\star)/\ln(1/\alpha^2) \\ &\leqslant R + B_0 C\lambda_{\max}\gamma_{\max}h_0(\beta^\star)/\ln(1/\alpha^2)\,,\end{aligned}$$

where $R = \mathcal{O}(\|\beta^\star\|_1)$ is independent of $\alpha$. Hence, since $B_0 = \sup_{k < \infty}\|\beta_k\|_\infty < \infty$, we have:

$$B_0(1 - C\lambda_{\max}\gamma_{\max}h_0(\beta^\star)/\ln(1/\alpha^2)) \leqslant R \implies B_0 \leqslant 2R\,,$$

provided that $\gamma_{\max} \leqslant 1/(2C\lambda_{\max}h_0(\beta^\star)/\ln(1/\alpha^2))$ (note that $h_0(\beta^\star)/\ln(1/\alpha^2)$ is independent of $\alpha^2$).

Hence, if for all $k$ we have $\gamma_k \leqslant \frac{1}{C'LB}$ where $B$ bounds all $\|\beta_k\|_\infty$, we have $\|\beta_k\|_\infty \leqslant 2R$ for all $k$, where $R = \mathcal{O}(\|\beta^\star\|_1)$ is independent of $\alpha$ and stepsizes $\gamma_k$.

Let $K > 0$ be fixed, and

$$\bar{\gamma} = \inf \left\{ \gamma > 0 \quad \text{s.t.} \quad \sup_{k \leqslant K} \|\beta_k\|_\infty > 2R \right\}.$$

For $\gamma \geqslant 0$ a constant stepsize, let

$$\varphi(\gamma) = \sup_{k \leqslant K} \|\beta_k\|_\infty,$$

which is a continuous function of $\gamma$. For $r > 0$, let $\gamma_r = \frac{1}{C'Lr}$.

An important feature to notice is that if $\gamma < \gamma_r$ and $r$ bounds all $\|\beta_k\|_\infty, k \leqslant K$, then $\varphi(\gamma) \leqslant R$, as shown above. We will show that we have $\bar{\gamma} \geqslant \gamma_{2R}$. Reasoning by contradiction, if $\bar{\gamma} < \gamma_{2R}$: by continuity of $\varphi$, we have $\varphi(\bar{\gamma}) \leqslant R$ and thus, there exists some small $0 < \varepsilon < \gamma_{2R} - \bar{\gamma}$ such that for all $\gamma \in [\bar{\gamma}, \bar{\gamma} + \varepsilon]$, we have $\varphi(\bar{\gamma}) \leqslant 2R$.

However, such $\gamma$'s verify both $\varphi(\gamma) \leqslant 2R$ (since $\gamma \in [\bar{\gamma}, \bar{\gamma} + \varepsilon]$ and by definition of $\varepsilon$) and $\gamma \leqslant \gamma_{2R}$ (by definition of $\varepsilon$), and hence $\varphi(\gamma) \leqslant R$. This contradicts the infimum of $\bar{\gamma}$, and hence $\bar{\gamma} \geqslant \gamma_{2R}$. Thus, for $\gamma \leqslant \gamma_{2R} = \frac{1}{2C'LR}$, we have $\|\beta_k\|_\infty \leqslant R$. $\qquad\square$

$\square$

# F    Proof of Theorem 1 and 2, and of Proposition 1

## F.1    Proof of Theorem 1 and 2

We are now equipped to prove Theorem 1 and Theorem 2, condensed in the following Theorem.

**Theorem 3.** *Let $(u_k, v_k)_{k \geqslant 0}$ follow the mini-batch SGD recursion (3) initialised at $u_0 = \sqrt{2}\alpha \in \mathbb{R}^d_{>0}$ and $v_0 = \mathbf{0}$, and let $(\beta_k)_{k \geqslant 0} = (u_k \odot v_k)_{k \geqslant 0}$. There exists and explicit $B > 0$ and a numerical constant $c > 0$ such that:*

1. *For stepsizes satisfying $\gamma_k \leqslant \frac{c}{LB}$, the iterates satisfy $\|\gamma_k \nabla \mathcal{L}_{\mathcal{B}_k}(\beta_k)\|_\infty \leqslant 1$ and $\|\beta_k\|_\infty \leqslant B$ for all $k$;*

2. *For stepsizes satisfying $\gamma_k \leqslant \frac{c}{LB}$, $(\beta_k)_{k \geqslant 0}$ converges almost surely to some $\beta^\star_\infty \in \mathcal{S}$,*

3. *If $(\beta_k)_k$ and the neurons $(u_k, v_k)_k$ respectively converge to a model $\beta^\star_\infty$ and neurons $(u_\infty, v_\infty)$ satisfying $\beta^\star_\infty \in \mathcal{S}$ (and $\beta^\star_\infty = u_\infty \odot v_\infty$), then for almost all stepsizes (with respect to the Lebesgue measure), the limit $\beta^\star_\infty$ satisfies:*

$$\beta^\star_\infty = \underset{\beta^\star \in \mathcal{S}}{\arg\min} \, D_{\psi_{\boldsymbol{\alpha}_\infty}}(\beta^\star, \tilde{\beta}_0),$$

*for $\boldsymbol{\alpha}_\infty \in \mathbb{R}^d_{>0}$ and $\tilde{\beta}_0 \in \mathbb{R}^d$ satisfying*

$$\boldsymbol{\alpha}^2_\infty = \boldsymbol{\alpha}^2 \odot \exp\left(-\sum_{k=0}^\infty q\big(\gamma_k \nabla \mathcal{L}_{\mathcal{B}_k}(\beta_k)\big)\right),$$

*where $q(x) = -\frac{1}{2}\ln((1-x^2)^2) \geqslant 0$ for $|x| \leqslant \sqrt{2}$, and $\tilde{\beta}_0$ is a perturbation term equal to:*

$$\tilde{\beta}_0 = \frac{1}{2}\big(\boldsymbol{\alpha}^2_+ - \boldsymbol{\alpha}^2_-\big),$$

*where, $q_\pm(x) = \mp 2x - \ln((1 \mp x)^2)$, and $\boldsymbol{\alpha}^2_\pm = \boldsymbol{\alpha}^2 \odot \exp\left(-\sum_{k=0}^\infty q_\pm(\gamma_k \nabla \mathcal{L}_{\mathcal{B}_k}(\beta_k))\right)$.*

*Proof.* **Point 1.** The first point of the Theorem is a direct consequence of Corollary 1 and the bounds proved in appendix E.3.

**Point 2.** Then, for stepsizes $\gamma_k \leqslant \frac{c}{LB}$, using Proposition 8 for any interpolator $\beta^\star \in \mathcal{S}$:

$$D_{h_{k+1}}(\beta^\star, \beta_{k+1}) \leqslant D_{h_k}(\beta^\star, \beta_k) - \gamma_k \mathcal{L}_{\mathcal{B}_k}(\beta_k). \tag{24}$$

Hence, summing:

$$\sum_k \gamma_k \mathcal{L}_{\mathcal{B}_k}(\beta_k) \leqslant D_{h_0}(\beta^\star, \beta_0),$$

so that the series converges.

Under our stepsize rule, $\|\gamma_k \nabla \mathcal{L}_{\mathcal{B}_k}(\beta_k)\|_\infty \leqslant \frac{1}{2}$, leading to $\|q(\gamma_k \nabla \mathcal{L}_{\mathcal{B}_k}(\beta_k)\|_\infty \leqslant 3\|\gamma_k \nabla \mathcal{L}_{\mathcal{B}_k}(\beta_k)\|_\infty^2$ by Lemma 5. Using $\|\nabla \mathcal{L}_{\mathcal{B}_k}(\beta_k)\|^2 \leqslant 2L_2 \mathcal{L}_{\mathcal{B}_k}(\beta_k)$, we have that $\ln(\boldsymbol{\alpha}_{\pm,k})$, $\ln(\boldsymbol{\alpha}_k)$ all converge.

We now show that $\sum_k \gamma_k \mathcal{L}(\beta_k) < \infty$. We have:

$$\sum_{\ell < k} \mathcal{L}(\beta_k) = \sum_{\ell < k} \gamma_k \mathcal{L}_{\mathcal{B}_k}(\beta_k) + M_k,$$

where $M_k = \sum_{\ell < k} \gamma_k(\mathcal{L}(\beta_k) - \mathcal{L}_{\mathcal{B}_k}(\beta_k))$. We have that $(M_k)$ is a martingale with respect to the filtration $(\mathcal{F}_k)$ defined as $\mathcal{F}_k = \sigma(\beta_\ell, \ell \leqslant k)$. Using our upper-bound on $\sum_{\ell < k} \gamma_k \mathcal{L}_{\mathcal{B}_k}(\beta_k)$, we have:

$$M_k \geqslant \sum_{\ell < k} \gamma_k \mathcal{L}(\beta_k) - \sum_{\ell < k} \gamma_k \mathcal{L}_{\mathcal{B}_k}(\beta_k) \geqslant -D_{h_0}(\beta^\star, \beta_0),$$

and hence $(M_k)$ is a lower bounded martingale. Using Doob's first martingale convergence theorem (a lower bounded super-martingale converges almost surely, Doob [17]), $(M_k)$ converges almost surely. Consequently, since $\sum_{\ell < k} \gamma_k \mathcal{L}(\beta_k) = \sum_{\ell < k} \gamma_k \mathcal{L}_{\mathcal{B}_k}(\beta_k) + M_k$, we have that $\sum_{\ell < k} \gamma_k \mathcal{L}(\beta_k)$ converges almost surely (the first term is upper bounded, the second converges almost surely).

We now prove the convergence of $(\beta_k)$. Since it is a bounded sequence, let $\beta_{\sigma(k)}$ be a convergent sub-sequence and let $\beta_\infty^\star$ denote its limit: $\beta_{\sigma(k)} \to \beta_\infty^\star$.

Almost surely, $\sum_k \gamma_k \mathcal{L}(\beta_k) < \infty$ and so $\gamma_k \mathcal{L}(\beta_k) \to 0$, leading to $\mathcal{L}(\beta_k) \to 0$ since the stepsizes are lower bounded, so that $\mathcal{L}(\beta_{\sigma(k)}) \to 0$, and hence $\mathcal{L}(\beta_\infty^\star) = 0$: this means that $\beta_\infty^\star$ is an interpolator.

Since the quantities $(\boldsymbol{\alpha}_k)_k$, $(\boldsymbol{\alpha}_{\pm,k})_k$ and $(\phi_k)_k$ converge almost surely to $\boldsymbol{\alpha}_\infty$, $\boldsymbol{\alpha}_\pm$ and $\phi_\infty$, we get that the potentials $h_k$ uniformly converge to $h_\infty = \psi_{\boldsymbol{\alpha}_\infty} - \langle \phi_\infty, \cdot \rangle$ on all compact sets. Now notice that we can decompose $\nabla h_\infty(\beta_\infty^\star)$ as:

$$\nabla h_\infty(\beta_\infty^\star) = \left(\nabla h_\infty(\beta_\infty^\star) - \nabla h_\infty(\beta_{\sigma(k)})\right) + \left(\nabla h_\infty(\beta_{\sigma(k)}) - \nabla h_{\sigma(k)}(\beta_{\sigma(k)})\right) + \nabla h_{\sigma(k)}(\beta_{\sigma(k)}).$$

The first two terms converge to 0: the first is a direct consequence of the convergence of the extracted subsequence, the second is a consequence of the uniform convergence of $h_{\sigma(k)}$ to $h_\infty$ on compact sets. Finally the last term is always in $\mathrm{Span}(x_1, \ldots, x_n)$ due to Proposition 5, leading to $\nabla h_\infty(\beta_\infty^\star) \in \mathrm{Span}(x_1, \ldots, x_n)$. Consequently, $\nabla h_\infty(\beta_\infty^\star) \in \mathrm{Span}(x_1, \ldots, x_n)$. Notice that from the definition of $h_\infty$, we have that $\nabla h_\infty(\beta_\infty^\star) = \nabla \psi_{\boldsymbol{\alpha}_\infty}(\beta_\infty^\star) - \phi_\infty$. Now since $\phi_\infty = \frac{1}{2} \mathrm{arcsinh}(\frac{\boldsymbol{\alpha}_+^2 - \boldsymbol{\alpha}_-^2}{2\boldsymbol{\alpha}_\infty^2})$, one can notice that $\tilde{\beta}_0$ is precisely defined such that $\nabla \psi_{\boldsymbol{\alpha}_\infty}(\tilde{\beta}_0) = \phi_\infty$. Therefore $\nabla \psi_{\boldsymbol{\alpha}_\infty}(\beta_\infty^\star) - \nabla \psi_{\boldsymbol{\alpha}_\infty}(\tilde{\beta}_0) \in \mathrm{Span}(x_1, \ldots, x_n)$. This condition along with the fact that $\beta_\infty^\star$ is an interpolator are exactly the optimality conditions of the convex minimisation problem:

$$\min_{\beta^\star \in \mathcal{S}} D_{\psi_{\boldsymbol{\alpha}_\infty}}(\beta^\star, \tilde{\beta}_0)$$

Therefore $\beta_\infty^\star$ must be equal to the unique minimiser of this problem. Since this is true for any sub-sequence we get that $\beta_k$ converges almost surely to:

$$\beta_\infty^\star = \underset{\beta \in \mathcal{S}}{\mathrm{argmin}} \; D_{\psi_{\boldsymbol{\alpha}_\infty}}(\beta^\star, \tilde{\beta}_0).$$

**Point 3.** From what we just proved, note that it is sufficient to prove that $\boldsymbol{\alpha}_k, \boldsymbol{\alpha}_{\pm,k}, \phi_k$ converge to limits $\boldsymbol{\alpha}_\infty, \boldsymbol{\alpha}_{\pm,\infty}, \phi_\infty$ satisfying $\boldsymbol{\alpha}_\infty, \boldsymbol{\alpha}_{\pm,\infty} \in \mathbb{R}_{>0}^d$ (with positive and non-null coordinates) and $\phi_\infty \in \mathbb{R}^d$. Indeed, if this holds and since we assume that the iterates converge to some interpolator, we proved just above that this interpolator is uniquely defined through the desired implicit regularization problem. We thus prove the convergence of $\boldsymbol{\alpha}_k, \boldsymbol{\alpha}_{\pm,k}, \phi_k$.

Note that the convergence of $u_k, v_k$ is equivalent to the convergence of $w_{\pm,k}$ in the $w_+^2 - w_-^2$ parameterisation used in our proofs, that we use there too. We have:

$$w_{\pm,k+1} = (1 \mp \gamma_k \nabla \mathcal{L}_{\mathcal{B}_k}(\beta_k)) \odot w_{\pm,k}\,,$$

so that

$$\ln(w_{\pm,k}^2) = \sum_{\ell < k} \ln((1 \mp \gamma_\ell \nabla \mathcal{L}_{\mathcal{B}_\ell}(\beta_\ell))^2)\,.$$

We now assume that stepsizes are such that for all $\ell \geqslant 0$ and $i \in [d]$, stepsizes are such that we have $|\gamma_\ell \nabla_i \mathcal{L}_{\mathcal{B}_\ell}(\beta_\ell)| \neq 1$: this is true for all stepsizes except a countable number of stepsizes, and so this is true for almost all stepsizes. Since we assume that the iterates $\beta_k$ converge to some interpolator, this leads to $\gamma_\ell \nabla \mathcal{L}_{\mathcal{B}_\ell}(\beta_\ell) \to 0$ if we assume that stepsizes do not diverge.

Taking the limit, we have

$$\ln(w_{\pm,\infty}^2) = \sum_{\ell < \infty} \ln((1 \mp \gamma_\ell \nabla \mathcal{L}_{\mathcal{B}_\ell}(\beta_\ell))^2)\,.$$

This limit is in $(\{-\infty\} \cup \mathbb{R})^d$ (since $w_{\pm,\infty} \in \mathbb{R}^d$), and a coordinate of the limit is equal to $-\infty$ if and only if the sum on the RHS diverges to $-\infty$ (note that from our assumption just above, no term of the sum can be equal to $-\infty$).

We have $\ln((1 \mp \gamma_\ell \nabla \mathcal{L}_{\mathcal{B}_\ell}(\beta_\ell))^2) \sim \mp 2\gamma_\ell \nabla \mathcal{L}_{\mathcal{B}_\ell}(\beta_\ell)$ as $\ell \to \infty$, so that if for some coordinate $i$ we have $\sum_\ell \gamma_\ell \nabla_i \mathcal{L}_{\mathcal{B}_\ell}(\beta_\ell) = \mp\infty$, then the coordinate $i$ of the limit satisfies $\ln(w_{i,\pm,\infty}^2) = +\infty$, which is impossible. Hence, the sum $\sum_\ell \gamma_\ell \nabla \mathcal{L}_{\mathcal{B}_\ell}(\beta_\ell)$ is in $\mathbb{R}^d$ (and is thus converging); consequently, $\sum_\ell \gamma_\ell^2 \nabla \mathcal{L}_{\mathcal{B}_\ell}(\beta_\ell)^2$ converges and thus $\sum_\ell q(\gamma_\ell \nabla \mathcal{L}_{\mathcal{B}_\ell}(\beta_\ell))$ and $\sum_\ell q_\pm(\gamma_\ell \nabla \mathcal{L}_{\mathcal{B}_\ell}(\beta_\ell))$ all converge: the sequences $\alpha_k, \alpha_{\pm,k}$ thus converge to limits in $\mathbb{R}_{>0}^d$, and $\phi_k$ converges, concluding our proof.

$\square$

### F.2 Proof of Proposition 1

We begin with the following Lemma, that explicits the curvature of $D_h$ around the set of interpolators.

**Lemma 3.** *For all* $k \geqslant 0$, *if* $\mathcal{L}(\beta_k) \leqslant \frac{1}{2\lambda_{\max}}(\alpha^2 \lambda_{\min}^+)^2$, *we have* $\left\| \beta_k - \beta_{\alpha_k}^\star \right\|^2 \leqslant 2B(\alpha^2 \lambda_{\min}^+)^{-1} \mathcal{L}(\beta_k)$.

*Proof.* Recall that the sequence $\mathbf{z}^k = \nabla h_k(\beta^k)$ satisfies $\mathbf{z}^0 = 0$ and $\mathbf{z}^{k+1} = \mathbf{z}^k - \gamma_k \mathcal{L}(\beta^k)$, so that we have that $\mathbf{z}^k \in V = \mathrm{Im}(\mathbf{XX}^\top)$ for all $k \geqslant 0$. Then, let $\beta_k^\alpha$ be the unique minimizer of $h_k$ over $\mathcal{S}$ the space of interpolators: $\beta_k^\alpha$ is exactly characterized by $\mathbf{X}^\top \beta_k^\alpha = \mathbf{Y}$ and $\nabla h_k(\beta_k^\alpha) \in V$. We define $\mathbf{z}_k^\alpha \in V$ as $\mathbf{z}_k^\alpha = \nabla h_k(\beta_k^\alpha)$.

Now, fix $\mathbf{z}^\alpha = \mathbf{z}_k^\alpha$ and $h = h_k$, and let us define $\psi : \mathbf{z} \in V \to D_{h^*}(\mathbf{z}, \mathbf{z}^\alpha)$ and $\phi : \mathbf{z} \in V \to \mathcal{L}(\nabla h^*(\mathbf{z}))$. We next show that for all $\mathbf{z} \in V$, there exists $\mu_z$ such that $\nabla^2 \phi(\mathbf{z}) \geqslant \mu_z \nabla^2 \psi(\mathbf{z})$, and that $\mu_z \geqslant \mu$ for $\mathbf{z}$ in an open convex set of $V$ around $\mathbf{z}^\alpha$, for some $\mu > 0$. For $A \in \mathbb{R}^{d \times d}$ an operator/matrix on $\mathbb{R}^d$, let us denote $A_V$ its restriction/co-restriction to $V$.

First, for $\mathbf{z} \in V$, we have $\nabla^2 \psi(\mathbf{z}) = \nabla^2(h^*(\mathbf{z}) - h^*(\mathbf{z}) - \langle \nabla h^*(\mathbf{z}^\alpha), z - z^\alpha \rangle)(\mathbf{z}) = \nabla^2 h^*(\mathbf{z})_V$. Then, $\nabla \phi(\mathbf{z}) = \nabla^2 h^*(\mathbf{z}) \nabla \mathcal{L}(\nabla h^*(\mathbf{z}))$, so that $\nabla^2 \phi(\mathbf{z}) = \left(\nabla^2 h^*(\mathbf{z}) \nabla^2 \mathcal{L}(\nabla h^*(\mathbf{z})) \nabla^2 h^*(\mathbf{z})\right)_V + \nabla^3 h^*(\mathbf{z})(\nabla \mathcal{L}(\nabla h^*(\mathbf{z})), \cdot, \cdot)_V$.

Since $h$ is $1/(2\alpha^2)$ smooth (on $\mathbb{R}^d$ and thus on $V$), $h^*$ is $2\alpha^2$ strongly convex (on $V$ and on $\mathbb{R}^d$). Using $V = \mathrm{Im}(\mathbf{XX}^\top)$ and $\nabla^2 \mathcal{L} \equiv \mathbf{XX}^\top$, we have $\left(\nabla^2 h^*(\mathbf{z}) \nabla^2 \mathcal{L}(\nabla h^*(\mathbf{z})) \nabla^2 h^*(\mathbf{z})\right)_V = \nabla^2 h^*(\mathbf{z})_V \nabla^2 \mathcal{L}(\nabla h^*(\mathbf{z}))_V \nabla^2 h^*(\mathbf{z})_V$, and thus $\left(\nabla^2 h^*(\mathbf{z}) \nabla^2 \mathcal{L}(\nabla h^*(\mathbf{z})) \nabla^2 h^*(\mathbf{z})\right)_V \succeq 2\alpha^2 \lambda_{\min}^+ \nabla^2 h^*(\mathbf{z})_V$.

For the other term of $\nabla^2 \phi$, namely $\nabla^3 h^*(\mathbf{z})(\nabla \mathcal{L}(\nabla h^*(\mathbf{z})), \cdot, \cdot)_V$, we compute $\nabla_{ijk}^3 h^*(\mathbf{z}) = \mathbf{1}_{i=j=k} 2\alpha_{i,k}^2 \sinh(\mathbf{z}_i)$, leading to: $\nabla^3 h^*(\mathbf{z})(\nabla \mathcal{L}(\nabla h^*(\mathbf{z})), \cdot, \cdot)_V = \mathrm{diag}(2\alpha^2 \sinh(\mathbf{z}) \odot (\mathbf{XX}^\top(2\alpha^2 \sinh(\mathbf{z}) - \beta^\alpha)))_V$. Thus, writing $\beta_{\mathbf{z}} = 2\alpha_{i,k}^2 \sinh(\mathbf{z}) = \nabla h^*(\mathbf{z})$ the primal surrogate of

$\mathbf{z}$, we have:

$$\nabla^3 h^*(\mathbf{z})(\nabla\mathcal{L}(\nabla h^*(\mathbf{z})),\cdot,\cdot)_V = \mathrm{diag}(2\alpha_{i,k}^2\sinh(\mathbf{z})\odot(\mathbf{X}\mathbf{X}^\top(\beta_\mathbf{z}-\beta_k^\alpha)))_V$$
$$\succeq -\left\|\mathbf{X}\mathbf{X}^\top(\beta_\mathbf{z}-\beta_k^\alpha)\right\|_\infty\mathrm{diag}(2\alpha_k^2\odot|\sinh(\mathbf{z})|)_V$$
$$\succeq -\left\|\mathbf{X}\mathbf{X}^\top(\beta_\mathbf{z}-\beta_k^\alpha)\right\|_\infty\mathrm{diag}(2\alpha_k^2\odot\cosh(\mathbf{z}))_V$$
$$= -\left\|\mathbf{X}\mathbf{X}^\top(\beta_\mathbf{z}-\beta_k^\alpha)\right\|_\infty\nabla^2\psi(\mathbf{z})\,.$$

Wrapping things together,

$$\nabla^2\phi(\mathbf{z})\succeq\left(2\alpha^2\lambda_{\min}^+ - \left\|\mathbf{X}\mathbf{X}^\top(\beta_\mathbf{z}-\beta^\alpha)\right\|_\infty\right)\nabla^2\psi(\mathbf{z})\,.$$

Let $\mathcal{Z} = \left\{\mathbf{z}\in V : \left\|\mathbf{X}\mathbf{X}^\top(\beta_\mathbf{z}-\beta_k^\alpha)\right\|_\infty < \alpha^2\lambda_{\min}^+\right\}$ that satisfies $\left\{\beta\in V : \mathcal{L}(\beta_\mathbf{z}) < \frac{1}{2\lambda_{\max}}(\alpha^2\lambda_{\min}^+)^2\right\} \subset \mathcal{Z}$. $\mathcal{Z}$ is an open convex set of $V$ containing $\mathbf{z}^\alpha$. On $\mathcal{Z}$, $\nabla^2\phi\succeq\alpha^2\lambda_{\min}^+\nabla^2\psi$, and $\psi(\mathbf{z}^\alpha)=\phi(\mathbf{z}^\alpha)=0$, so that for all $\mathbf{z}\in\mathcal{Z}$, we have $\phi(\mathbf{z})\geqslant\alpha^2\lambda_{\min}^+\psi(\mathbf{z})$. Hence, for all $\mathbf{z}\in\mathcal{Z}$, we have $D_{h_k}(\beta_k^\alpha,\beta_\mathbf{z})\leqslant D_{h^\star}(\mathbf{z},\mathbf{z}^\alpha)\leqslant(\alpha^2\lambda_{\min}^+)^{-1}\mathcal{L}(\beta_\mathbf{z})$, and using the fact that $D_{h_k}$ is $\frac{1}{4B}$ strongly convex, we obtain, for $\beta_\mathbf{z}=\beta_k$ (since $\mathbf{z}^k\in V$): if $\mathcal{L}(\beta_k)\leqslant\frac{1}{2\lambda_{\max}}(\alpha^2\lambda_{\min}^+)^2$, we have $\|\beta_k^\alpha-\beta_k\|_2^2\leqslant(\alpha^2\lambda_{\min}^+)^{-1}\mathcal{L}(\beta_k)$. $\qquad\square$

**Proposition 14.** *As assume $\mathcal{L}$ is $L_r$-relatively smooth with respect to all the $h_k$'s. Then for all $\beta$ we have the following inequality.*

$$\gamma_k(\mathcal{L}(\beta_{k+1})-\mathcal{L}(\beta)) \leqslant D_{h_k}(\beta,\beta_k) - D_{h_{k+1}}(\beta,\beta_{k+1}) - (1-\gamma_k L_r)D_{h_k}(\beta_{k+1},\beta_k)$$
$$+ (h_{k+1}-h_k)(\beta) - (h_{k+1}-h_k)(\beta_{k+1})\,.$$

*Proof.* For any $\beta,\beta_k,\beta_{k+1}$, the following holds (three points identity for time varying potentials, Proposition 9):

$$D_{h_k}(\beta,\beta_k) - D_{h_{k+1}}(\beta,\beta_{k+1}) = \big[h_k(\beta) - (h_k(\beta_k) + \langle\nabla h_k(\beta_k),\beta-\beta_k\rangle)\big]$$
$$- \big[h_{k+1}(\beta) - (h_{k+1}(\beta_{k+1}) + \langle\nabla h_{k+1}(\beta_{k+1}),\beta-\beta_{k+1}\rangle)\big]$$
$$= h_k(\beta) - h_{k+1}(\beta) + \langle\nabla h_{k+1}(\beta_{k+1}) - \nabla h_k(\beta_k),\beta-\beta_{k+1}\rangle$$
$$+ h_{k+1}(\beta_{k+1}) - \big[h_k(\beta_k) + \langle\nabla h_k(\beta_k),\beta_{k+1}-\beta_k\rangle\big]$$
$$= h_k(\beta) - h_{k+1}(\beta) + \langle\nabla h_{k+1}(\beta_{k+1}) - \nabla h_k(\beta_k),\beta-\beta_{k+1}\rangle$$
$$+ h_{k+1}(\beta_{k+1}) - h_k(\beta_{k+1}) + D_{h_k}(\beta_{k+1},\beta_k).$$

Rearranging and plugging in our mirror update we obtain that for all $\beta$:

$$\gamma_k\langle\nabla\mathcal{L}(\beta_k),\beta_{k+1}-\beta\rangle = D_{h_k}(\beta,\beta_k) - D_{h_{k+1}}(\beta,\beta_{k+1})$$
$$- D_{h_k}(\beta_{k+1},\beta_k) - (h_{k+1}-h_k)(\beta_{k+1}) + (h_{k+1}-h_k)(\beta).$$

From the convexity of $\mathcal{L}$ and its $L_r$-relative smoothness we also have that:

$$\mathcal{L}(\beta_{k+1})\leqslant\mathcal{L}(\beta) + \langle\nabla\mathcal{L}(\beta_k),\beta_{k+1}-\beta\rangle + L_r D_{h_k}(\beta_{k+1},\beta_k),$$

Finally:

$$\gamma_k(\mathcal{L}(\beta_{k+1})-\mathcal{L}(\beta)) \leqslant D_{h_k}(\beta,\beta_k) - D_{h_{k+1}}(\beta,\beta_{k+1}) - (1-\gamma_k L_r)D_{h_k}(\beta_{k+1},\beta_k)$$
$$+ (h_{k+1}-h_k)(\beta) - (h_{k+1}-h_k)(\beta_{k+1}).$$

Note that in our setting, for any $\beta$, $k\mapsto h_k(\beta)$ is **increasing**. We can therefore write that:

$$\gamma_k(\mathcal{L}(\beta_{k+1})-\mathcal{L}(\beta))\leqslant D_{h_k}(\beta,\beta_k) - D_{h_{k+1}}(\beta,\beta_{k+1}) - (1-\gamma_k L_r)D_{h_k}(\beta_{k+1},\beta_k) + (h_{k+1}-h_k)(\beta).$$

In particular, for $\beta=\beta^*$:

$$\gamma_k\mathcal{L}(\beta_{k+1})\leqslant D_{h_k}(\beta^*,\beta_k) - D_{h_{k+1}}(\beta^*,\beta_{k+1}) - (1-\gamma_k L)D_{h_k}(\beta_{k+1},\beta_k) + (h_{k+1}-h_k)(\beta^*)$$
$$- (h_{k+1}-h_k)(\beta_{k+1})$$
$$\leqslant D_{h_k}(\beta^*,\beta_k) - D_{h_{k+1}}(\beta^*,\beta_{k+1}) - (1-\gamma_k L_r)D_{h_k}(\beta_{k+1},\beta_k) + (h_{k+1}-h_k)(\beta^*)$$

and in $\beta=\beta_k$:

$$\gamma_k\mathcal{L}(\beta_{k+1})\leqslant\gamma_k\mathcal{L}(\beta_k) - D_{h_{k+1}}(\beta_k,\beta_{k+1}) - (1-\gamma_k L_r)D_{h_k}(\beta_{k+1},\beta_k) + (h_{k+1}-h_k)(\beta_k)$$
$$- (h_{k+1}-h_k)(\beta_{k+1})$$
$$\leqslant\gamma_k\mathcal{L}(\beta_k) - D_{h_{k+1}}(\beta_k,\beta_{k+1}) - (1-\gamma_k L_r)D_{h_k}(\beta_{k+1},\beta_k) + (h_{k+1}-h_k)(\beta_k)$$

$$\square$$

*Proof of Proposition 1.* We apply Proposition 14 for $\beta = \beta_k$, with $L_r = 4BL$ (using Lemma 6) and replacing $\mathcal{L}$ by $\mathcal{L}_{\mathcal{B}_k}$, to obtain:

$$\gamma_k(\mathcal{L}_{\mathcal{B}_k}(\beta_{k+1}) - \mathcal{L}_{\mathcal{B}_k}(\beta_k)) \leqslant -D_{h_{k+1}}(\beta_k, \beta_{k+1}) - (1 - \gamma_k L_r)D_{h_k}(\beta_{k+1}, \beta_k)$$
$$+ (h_{k+1} - h_k)(\beta_k) - (h_{k+1} - h_k)(\beta_{k+1}),$$

and thus, taking the mean wrt $\mathcal{B}_k$,

$$\gamma_k(\mathbb{E}_{\mathcal{B}_k}\mathcal{L}(\beta_{k+1}) - \mathcal{L}(\beta_k)) \leqslant -\mathbb{E}_{\mathcal{B}_k}D_{h_{k+1}}(\beta_k, \beta_{k+1}) - (1 - \gamma_k L_r)\mathbb{E}_{\mathcal{B}_k}D_{h_k}(\beta_{k+1}, \beta_k)$$
$$+ \mathbb{E}_{\mathcal{B}_k}(h_{k+1} - h_k)(\beta_k) - \mathbb{E}_{\mathcal{B}_k}(h_{k+1} - h_k)(\beta_{k+1})$$
$$\leqslant -(1 - \gamma_k L_r)\mathbb{E}_{\mathcal{B}_k}D_{h_k}(\beta_{k+1}, \beta_k)$$
$$+ \mathbb{E}_{\mathcal{B}_k}(h_{k+1} - h_k)(\beta_k) - \mathbb{E}_{\mathcal{B}_k}(h_{k+1} - h_k)(\beta_{k+1}).$$

First, as in the proof of Proposition 10, using the fact that $h_k$ is $\ln(1/\alpha_k)$ smooth,

$$D_{h_k}(\beta_{k+1}, \beta_k) \geqslant \frac{1}{2\ln(1/\alpha_k)}\|\nabla h_k(\beta_k) - \gamma_k\nabla\mathcal{L}_{\mathcal{B}_k}(\beta_k) - \nabla h_k(\beta_k) + \nabla h_{k+1}(\beta_{k+1}) - \nabla h_k(\beta_{k+1})\|_2^2$$

$$\geqslant -\frac{1}{2\ln(1/\alpha_k)}\|\nabla h_k(\beta_k) - \nabla h_{k+1}(\beta_k)\|_2^2 + \frac{1}{4\ln(1/\alpha_k)}\|\gamma_k\nabla\mathcal{L}_{\mathcal{B}_k}(\beta_k)\|_2^2,$$

and thus

$$\mathbb{E}D_{h_k}(\beta_{k+1}, \beta_k) \geqslant \mathbb{E}\left[-\frac{1}{2\ln(1/\alpha_k)}\|\nabla h_k(\beta_k) - \nabla h_{k+1}(\beta_k)\|_2^2 + \frac{\lambda_b}{2\ln(1/\alpha_k)}\gamma_k^2\mathcal{L}_{\mathcal{B}}(\beta_k)\right].$$

Now, we apply Lemma 7 assuming that $\|\beta^\star\|_\infty, \|\beta_{k+1}\|_\infty \leqslant B$ (which is satisfied since we are under the assumption of Theorem 2):

$$(h_{k+1} - h_k)(\beta_k) - (h_{k+1} - h_k)(\beta^\star) \leqslant 24BL\gamma_k^2\mathcal{L}_{\mathcal{B}_k}(\beta_k).$$

Using $|\nabla h_k(\beta) - \nabla h_{k+1}(\beta)| \leqslant 2\delta_k$ where $\delta_k = q(\gamma_k\nabla\mathcal{L}_{\mathcal{B}_k}(\beta_k))$ as in Proposition 10, we have:

$$\mathbb{E}\|\nabla h_k(\beta_k) - \nabla h_{k+1}(\beta_k)\|_2^2 \leqslant 16B\gamma_k^2\mathbb{E}\|\nabla\mathcal{L}_{\mathcal{B}_k}(\beta_k)\|^2 \leqslant 32BL\gamma_k^2\mathbb{E}\mathcal{L}(\beta_k).$$

Wrapping everything together,

$$\mathbb{E}\left[\mathcal{L}(\beta_{k+1}) - \mathcal{L}(\beta_k)\right] \leqslant -(1 - \gamma_k 4BL)\frac{\lambda_b}{2\ln(1/\alpha_k)}\gamma_k\mathbb{E}\mathcal{L}(\beta_k)$$

$$+ \left(\gamma_k^2(1 - 4\gamma_k BL)24BL + \frac{32BL}{\ln(1/\alpha_k)}\right)\gamma_k^2\mathbb{E}\mathcal{L}(\beta_k).$$

Thus, for $\gamma_k \leqslant \frac{c'}{LB\ln(1/(\min_i \alpha_{k,i}))}$, we have the first part of Proposition 1.

Using Lemma 3, we then have:

$$\mathbb{E}\left[\|\beta_k - \beta_{\alpha_k}^\star\|^2\right] = \mathbb{E}\left[\mathbf{1}_{\left\{\mathcal{L}(\beta_k) \leqslant \frac{1}{2\lambda_{\max}}(\alpha^2\lambda_{\min}^+)^2\right\}}\|\beta_k - \beta_{\alpha_k}^\star\|^2\right]$$

$$+ \mathbb{E}\left[\mathbf{1}_{\left\{\mathcal{L}(\beta_k) > \frac{1}{2\lambda_{\max}}(\alpha^2\lambda_{\min}^+)^2\right\}}\|\beta_k - \beta_{\alpha_k}^\star\|^2\right]$$

$$\leqslant \mathbb{E}\left[\mathbf{1}_{\left\{\mathcal{L}(\beta_k) \leqslant \frac{1}{2\lambda_{\max}}(\alpha^2\lambda_{\min}^+)^2\right\}}2B(\alpha^2\lambda_{\min}^+)^{-1}\mathcal{L}(\beta_k)\right]$$

$$+ \mathbb{P}\left(\mathcal{L}(\beta_k) > \frac{1}{2\lambda_{\max}}(\alpha^2\lambda_{\min}^+)^2\right) \times 4B^2$$

$$\leqslant 2B(\alpha^2\lambda_{\min}^+)^{-1}\mathbb{E}\left[\mathcal{L}(\beta_k)\right]$$

$$+ \frac{\mathbb{E}\left[\mathcal{L}(\beta_k)\right]}{\frac{1}{2\lambda_{\max}}(\alpha^2\lambda_{\min}^+)^2} \times 4B^2$$

$$= 2B(\alpha^2\lambda_{\min}^+)^{-1}\left(1 + \frac{4B\lambda_{\max}}{\alpha^2\lambda_{\min}^+}\right)\mathbb{E}\left[\mathcal{L}(\beta_k)\right].$$

$\square$

# G  Proof of miscellaneous results mentioned in the main text

In this section, we provide proofs for results mentioned in the main text and that are not directly directed to the proof of Theorem 3.

## G.1  Proof of Proposition 3 and the sum of the losses

We start by proving the following proposition, present as is in the first 9 pages of this paper. We then continue with upper and lower bounds (of similar magnitude) on the sum of the losses.

**Proposition 3.** *Let $\Lambda_b, \lambda_b > 0$ [5] be the largest and smallest values, respectively, such that $\lambda_b H \preceq \mathbb{E}_\mathcal{B}\left[H_\mathcal{B}^2\right] \preceq \Lambda_b H$. For any stepsize $\gamma > 0$ satisfying $\gamma \leqslant \frac{c}{BL}$ (as in Theorem 2), initialisation $\alpha\mathbf{1}$ and batch size $b \in [n]$, the magnitude of the gain satisfies:*

$$\lambda_b\gamma^2 \sum_k \mathbb{E}\mathcal{L}(\beta_k) \leqslant \mathbb{E}\left[\|\mathrm{Gain}_\gamma\|_1\right] \leqslant 2\Lambda_b\gamma^2 \sum_k \mathbb{E}\mathcal{L}(\beta_k), \tag{10}$$

*where the expectation is over a uniform and independent sampling of the batches $(\mathcal{B}_k)_{k\geqslant 0}$.*

*Proof.* From Lemma 5, for all $-1/2 \leqslant x \leqslant 1/2$, it holds that $x^2 \leqslant q(x) \leqslant 2x^2$. We have, using $\|\gamma_k\nabla\mathcal{L}_{\mathcal{B}_k}(\beta_k)\|_\infty \leqslant 1/2$ (which holds under the stepsize assumption):

$$\begin{aligned}
\mathbb{E}\|\mathrm{Gain}_\gamma\|_1 &= -\mathbb{E}\sum_i \ln\left(\frac{\alpha_{\infty,i}}{\alpha}\right) \\
&= \sum_{\ell<\infty}\sum_i \mathbb{E}q\big(\gamma_\ell\nabla_i\mathcal{L}_{\mathcal{B}_\ell}(\beta_\ell)\big) \\
&\leqslant 2\sum_{\ell<\infty}\sum_i \mathbb{E}\big(\gamma_\ell\nabla_i\mathcal{L}_{\mathcal{B}_\ell}(\beta_\ell)\big)^2 \\
&= \sum_{\ell<\infty}\gamma_\ell^2\mathbb{E}\|\nabla\mathcal{L}_{\mathcal{B}_\ell}(\beta_\ell)\|_2^2 \\
&\leqslant 4\Lambda_b\sum_{\ell<\infty}\gamma_\ell^2\mathbb{E}\mathcal{L}_{\mathcal{B}_\ell}(\beta_\ell),
\end{aligned}$$

since $\mathbb{E}\|\nabla\mathcal{L}_{\mathcal{B}_\ell}(\beta_\ell)\|_2^2 \leqslant 2\Lambda_b\mathcal{L}_{\mathcal{B}_\ell}(\beta_\ell)$. For the left handside we use $q(x) \geqslant x^2$ for $|x| \leqslant 1/2$ and $\mathbb{E}\|\nabla\mathcal{L}_{\mathcal{B}_\ell}(\beta_\ell)\|_2^2 \geqslant 2\lambda_b\mathcal{L}_{\mathcal{B}_k}(\beta_\ell)$. Finally, since $\mathcal{B}_\ell$ independent freom $\beta_\ell$, we have $\mathbb{E}\mathcal{L}_{\mathcal{B}_\ell}(\beta_\ell) = \mathbb{E}\mathcal{L}(\beta_\ell)$. □

**Proposition 15.** *For stepsizes $\gamma_k \equiv \gamma \leqslant \frac{c}{LB}$ (as in Theorem 2), we have:*

$$\sum_{k\geqslant 0}\gamma^2\mathbb{E}\mathcal{L}(\beta_k) = \Theta\left(\gamma\|\beta^\star\|_1\ln(1/\alpha)\right).$$

*Proof.* We first lower bound $\sum_{k<\infty}\gamma_k^2\mathcal{L}_{\mathcal{B}_k}(\beta_k)$. We have the following equality, that holds for any $k$:

$$\begin{aligned}
D_{h_{k+1}}(\beta^\star, \beta_{k+1}) &= D_{h_k}(\beta^\star, \beta_k) - 2\gamma\mathcal{L}_{\mathcal{B}_k}(\beta_k) + D_{h_{k+1}}(\beta_k, \beta_{k+1}) \\
&\quad + \big(h_k - h_{k+1}\big)(\beta_k) - \big(h_k - h_{k+1}\big)(\beta^\star),
\end{aligned}$$

leading to, by summing for $k \in \mathbb{N}$:

$$\sum_{k<\infty}2\gamma\mathcal{L}_{\mathcal{B}_k}(\beta_k) = D_{h_0}(\beta^\star, \beta_0) - \lim_{k\to\infty}D_{h_k}(\beta^\star, \beta_k) + \sum_{k<\infty}D_{h_{k+1}}(\beta_k, \beta_{k+1}) + \sum_{k<\infty}\big(h_k - h_{k+1}\big)(\beta_k) - \big(h_k - h_{k+1}\big)(\beta^\star).$$

First, since $h_k \to h_\infty, \beta_k \to \beta_\infty$, we have $\lim_{k\to\infty}D_{h_k}(\beta^\star, \beta_k) = 0$. Then, $D_{h_{k+1}}(\beta_k, \beta_{k+1}) \geqslant 0$. Finally, $\left|\big(h_k - h_{k+1}\big)(\beta_k) - \big(h_k - h_{k+1}\big)(\beta^\star)\right| \leqslant 16BL_2\gamma^2\mathcal{L}_{\mathcal{B}_k}(\beta_k)$. Hence :

$$\sum_{k<\infty}2\gamma(1 + 16\gamma BL_2)\mathcal{L}_{\mathcal{B}_k}(\beta_k) \geqslant D_{h_0}(\beta^\star, \beta_0),$$

---

[5]$\Lambda_b, \lambda_b > 0$ are data-dependent constants; for $b = n$, we have $(\lambda_n, \Lambda_n) = (\lambda_{\min}^+(H), \lambda_{\max}(H))$ where $\lambda_{\min}^+(H)$ is the smallest non-null eigenvalue of $H$; for $b = 1$, we have $\min_i\|x_i\|_2^2 \leqslant \lambda_1 \leqslant \Lambda_1 \leqslant \max_i\|x_i\|_2^2$.

and thus $\sum_{k<\infty}\gamma\mathcal{L}_{\mathcal{B}_k}(\beta_k) \geqslant D_{h_0}(\beta^\star,\beta_0)/4$ for $\gamma \leqslant c/(BL)$ (with $c \geqslant 16$). This gives the RHS inequality. The LHS is a direct consequence of bounds proved in previous subsections.

Hence, we have that

$$\gamma^2 \sum_k \mathcal{L}(\beta_k) = \Theta\left(\gamma D_{h_0}(\beta^\star,\beta_0)\right).$$

Noting that $D_{h_0}(\beta^\star,\beta_0) = h_0(\beta^\star) = \Theta\left(\ln(1/\alpha)\|\beta^\star\|_1\right)$ concludes the proof. □

## G.2 $\tilde{\beta}_0$ is negligible

In the following proposition we show that $\tilde{\beta}_0$ is close to $\mathbf{0}$ and therefore one should think of the implicit regularization problem as $\beta^\star_\infty = \arg\min_{\beta^\star \in S} \psi_{\alpha_\infty}(\beta^\star)$

**Proposition 16.** *Under the assumptions of Theorem 2,*

$$|\tilde{\beta}_0| \leqslant \alpha^2,$$

*where the inequality must be understood coordinate-wise.*

*Proof.*

$$\begin{aligned}
|\tilde{\beta}_0| &= \frac{1}{2}|\alpha_+^2 - \alpha_-^2| \\
&= \frac{1}{2}\alpha^2 \left| \exp\left(-\sum_k q_+(\gamma_k \nabla\mathcal{L}(\beta_k))\right) - \exp\left(-\sum_k q_-(\gamma_k \nabla\mathcal{L}(\beta_k))\right) \right| \\
&\leqslant \alpha^2,
\end{aligned}$$

where the inequality is because $q_+(\gamma_k \nabla\mathcal{L}(\beta_k)) \geqslant 0$, $q_-(\gamma_k \nabla\mathcal{L}(\beta_k)) \geqslant 0$ for all $k$.

□

## G.3 Impact of stochasticity and linear scaling rule

**Proposition 17.** *With probability $1 - 2ne^{-d/16} - 3/n^2$ over the $x_i \sim_{\mathrm{iid}} \mathcal{N}(0,\sigma^2 I_d)$, $c_1 \frac{d\sigma^2}{b}(1 + o(1)) \leqslant \lambda_b \leqslant \Lambda_b \leqslant c_2 \frac{d\sigma^2}{b}(1 + o(1))$,*

so that under these assumptions,

$$\sum_k \gamma_k \mathbb{E}\mathcal{L}(\beta_k) = \Theta\left(\frac{\gamma}{b}\sigma^2\|\beta^\star\|_1 \ln(1/\alpha)\right).$$

*Proof.* The bound on $\lambda_b, \Lambda_b$ is a direct consequence of the concentration bound provided in Lemma 13.
□

## G.4 (Stochastic) gradients at the initialisation

To understand the behaviour and the effects of the stochasticity and the stepsize on the shape of $\mathrm{Gain}_\gamma$, we analyse a noiseless sparse recovery problem under the following standard assumption 2 [10] and as common in the sparse recovery literature, we make the following assumption 3 on the inputs.

**Assumption 2.** *There exists an $s$-sparse ground truth vector $\beta^\star_{\mathrm{sparse}}$ where $s$ verifies $n = \Omega(s\ln(d))$, such that $y_i = \langle\beta^\star_{\mathrm{sparse}}, x_i\rangle$ for all $i \in [n]$.*

**Assumption 3.** *There exists $\delta, c_1, c_2 > 0$ such that for all $s$-sparse vectors $\beta$, there exists $\varepsilon \in \mathbb{R}^d$ such that $(X^\top X)\beta = \beta + \varepsilon$ where $\|\varepsilon\|_\infty \leqslant \delta\|\beta\|_2$ and $c_1\|\beta\|_2^2 \mathbf{1} \leqslant \frac{1}{n}\sum_i x_i^2 \langle x_i, \beta\rangle^2 \leqslant c_2\|\beta\|_2^2 \mathbf{1}$.*

The first part of Assumption 3 closely resembles the classical restricted isometry property (RIP) and is relevant for GD while the second part is relevant for SGD. Such an assumption is not restrictive and holds with high probability for Gaussian inputs $\mathcal{N}(0,\sigma^2 I_d)$ (see Lemma 10 in Appendix).

Based on the claim above, we analyse the shape of the (stochastic) gradient at initialisation. For GD and SGD, it respectively writes, where $g_0 = \nabla \mathcal{L}_{i_0}(\beta_0)^2$, $i_0 \sim \mathrm{Unif}([n])$:

$$\nabla \mathcal{L}(\beta_0)^2 = [X^\top X \beta^\star]^2, \quad \mathbb{E}_{i_0}[g_0] = \frac{1}{n} \sum_i x_i^2 \langle x_i, \beta^\star \rangle^2.$$

The following lemma then shows that while the initial stochastic gradients of SGD are homogeneous, it is not the case for that of GD.

**Proposition 18.** *Under Assumption 3, the squared full batch gradient and the expected stochastic gradient at initialisation satisfy, for some $\varepsilon$ verifying $\|\varepsilon\|_\infty << \left\|\beta^\star_{\mathrm{sparse}}\right\|_\infty^2$:*

$$\nabla \mathcal{L}(\beta_0)^2 = (\beta^\star_{\mathrm{sparse}})^2 + \varepsilon, \tag{25}$$

$$\mathbb{E}_{i_0}[\nabla \mathcal{L}_{i_0}(\beta_0)^2] = \Theta\left(\|\beta^\star\|_2^2 \mathbf{1}\right). \tag{26}$$

*Proof of Proposition 18.* Under Assumption 3, we have using:

$$\begin{aligned}
\nabla \mathcal{L}(\beta_0)^2 &= (X^\top X \beta^\star_{\mathrm{sparse}}) \\
&= (\beta^\star_{\mathrm{sparse}} + \varepsilon)^2 \\
&= \beta^\star_{\mathrm{sparse}}{}^2 + \varepsilon^2 + 2\varepsilon \beta^\star_{\mathrm{sparse}}.
\end{aligned}$$

We have $\left\|\varepsilon^2 + 2\varepsilon \beta^\star_{\mathrm{sparse}}\right\|_\infty \leqslant \|\varepsilon\|_\infty^2 + 2\|\varepsilon\|_\infty \left\|\beta^\star_{\mathrm{sparse}}\right\|_\infty$, and we conclude by using $\|\varepsilon\|_\infty \leqslant \delta \left\|\beta^\star_{\mathrm{sparse}}\right\|_2$.

Then,

$$\mathbb{E}_{i \sim \mathrm{Unif}([n])}[\nabla \mathcal{L}_i(\beta_0)^2] = \frac{1}{n} x_i^2 \langle x_i, \beta^\star_{\mathrm{sparse}} \rangle,$$

and we conclude using Assumption 3.

$\square$

*Proof of Proposition 4.* The proof proceeds as that of Proposition 18. $\square$

## G.5  Convergence of $\alpha_\infty$ and $\tilde{\beta}_0$ for $\gamma \to 0$

**Proposition 19.** *Let $\tilde{\beta}_0(\gamma), \alpha_\infty(\gamma)$ be as defined in Theorem 1, for constant stepsizes $\gamma_k \equiv \gamma$. We have:*

$$\tilde{\beta}_0(\gamma) \to 0, \quad \boldsymbol{\alpha}_\infty \to \alpha \mathbf{1},$$

*when $\gamma \to 0$.*

*Proof.* We have, as proved previously, that

$$\begin{aligned}
\left\| \sum_k \gamma^2 \nabla \mathcal{L}_{\mathcal{B}_k}(\beta_k)^2 \right\|_1 &\leqslant \sum_k \gamma^2 \left\| \nabla \mathcal{L}_{\mathcal{B}_k}(\beta_k)^2 \right\|_1 \\
&= \sum_k \gamma^2 \left\| \nabla \mathcal{L}_{\mathcal{B}_k}(\beta_k) \right\|_2^2 \\
&\leqslant 2L\gamma^2 \sum_k \mathcal{L}_{\mathcal{B}_k}(\beta_k) \\
&\leqslant 2L\gamma D_{h_0}(\beta^\star, \beta_0),
\end{aligned}$$

for $\gamma \leqslant \frac{c}{BL}$. Thus, $\sum_k \gamma^2 \nabla \mathcal{L}_{\mathcal{B}_k}(\beta_k)^2 \to 0$ as $\gamma \to 0$ (note that $\beta_k$ implicitly depends on $\gamma$, so that this result is not immediate).

Then, for $\gamma \leqslant \frac{c}{LB}$,

$$\left\| \ln(\boldsymbol{\alpha}_\infty^2/\alpha^2) \right\|_1 \leqslant \sum_k \|q(\gamma \mathcal{L}(\beta_k)\|_1 \leqslant 2 \sum_k \gamma^2 \left\| \nabla \mathcal{L}_{\mathcal{B}_k}(\beta_k)^2 \right\|_1,$$

which tends to 0 as $\gamma \to 0$. Similarly, $\left\|\ln(\alpha_{+,\infty}^2/\alpha^2)\right\|_1 \to 0$ and $\left\|\ln(\alpha_{-,\infty}^2/\alpha^2)\right\|_1 \to 0$ as $\gamma \to 0$, leading to $\tilde{\beta}_0(\gamma) \to 0$ as $\gamma \to 0$.

$\square$

## H  Technical lemmas

In this section we present a few technical lemmas, used and referred to throughout the proof of **??**.

**Lemma 4.** *Let $\alpha_+, \alpha_- > 0$ and $x \in \mathbb{R}$, and $\beta = \alpha_+^2 e^x - \alpha_-^2 e^{-x}$. We have:*

$$\operatorname{arcsinh}\left(\frac{\beta}{2\alpha_+\alpha_-}\right) = x + \ln\left(\frac{\alpha_+}{\alpha_-}\right) = x + \operatorname{arcsinh}\left(\frac{\alpha_+^2 - \alpha_-^2}{2\alpha_+\alpha_-}\right).$$

*Proof.* First,

$$\begin{aligned}
\frac{\beta}{2\alpha_+\alpha_-} &= \frac{1}{2}\left(\frac{\alpha_+}{\alpha_-}e^x - \left(\frac{\alpha_+}{\alpha_-}\right)^{-1}e^{-x}\right) \\
&= \frac{e^{x+\ln(\alpha_+/\alpha_-)} - e^{-x-\ln(\alpha_+/\alpha_-)}}{2} \\
&= \sinh(x + \ln(\alpha_+/\alpha_-)),
\end{aligned}$$

hence the result by taking the arcsinh of both sides. Note also that we have $\ln(\alpha_+/\alpha_-) = \operatorname{arcsinh}(\frac{\alpha_+^2 - \alpha_-^2}{2\alpha_+\alpha_-})$. $\square$

**Lemma 5.** *If $|x| \leqslant 1/2$ then $x^2 \leqslant q(x) \leqslant 2x^2$*

**Lemma 6.** *On the $\ell_\infty$ ball of radius $B$, the quadratic loss function $\beta \mapsto \mathcal{L}(\beta)$ is $4\lambda_{\max}\max(B, \alpha^2)$-relatively smooth w.r.t all the $h_k$'s.*

*Proof.* We have:

$$\nabla^2 h_k(\beta) = \operatorname{diag}\left(\frac{1}{2\sqrt{\alpha_k^4 + \beta^2}}\right) \succeq \operatorname{diag}\left(\frac{1}{2\sqrt{\alpha^4 + \beta^2}}\right),$$

since $\alpha_k \leqslant \alpha$ component-wise. Thus, $\nabla^2 h_k(\beta) \succeq \frac{1}{2}\min\left(\min_{1\leqslant i\leqslant d}\frac{1}{2|\beta_i|}, \frac{1}{2\alpha^2}\right)I_d = \frac{1}{\max(4\|\beta\|_\infty, 4\alpha^2)}I_d$, and $h_k$ is $\frac{1}{\max(4B, 4\alpha^2)}$-strongly convex on the $\ell^\infty$ norm of radius $B$. Since $\mathcal{L}$ is $\lambda_{\max}$-smooth over $\mathbb{R}^d$, we have our result. $\square$

**Lemma 7.** *For $k \geqslant 0$ and for all $\beta \in \mathbb{R}^d$:*

$$|h_{k+1}(\beta) - h_k(\beta)| \leqslant 8L_2\gamma_k^2 \mathcal{L}_{\mathcal{B}_k}(\beta_k)\|\beta\|_\infty.$$

*Proof.* We have $\alpha_{+,k+1}^2 = \alpha_{+,k}^2 e^{-\delta_{+,k}}$ and $\alpha_{-,k+1}^2 = \alpha_{-,k}^2 e^{-\delta_{-,k}}$, for $\delta_{+,k} = \tilde{q}(\gamma_k\nabla\mathcal{L}_{\mathcal{B}_k}(\beta_k))$ and $\delta_{-,k} = \tilde{q}(-\gamma_k\nabla\mathcal{L}_{\mathcal{B}_k}(\beta_k))$. And $\alpha_{k+1} = \alpha_k\exp(-\delta_k)$ where $\delta_k := \delta_{+,k} + \delta_{-,k} = q(\gamma_k\nabla\mathcal{L}_{\mathcal{B}_k}(\beta_k))$. To prove the result we will use that for $\beta \in \mathbb{R}^d$, we have $|(h_{k+1} - h_k)(\beta)| \leqslant \sum_{i=1}^d \int_0^{|\beta_i|} |\nabla_i h_{k+1}(x) - \nabla_i h_k(x)|\mathrm{d}x$.

First, using that $|\operatorname{arcsinh}(a) - \operatorname{arcsinh}(b)| \leqslant |\ln(a/b)|$ for $ab > 0$. We have that

$$\begin{aligned}
\left|\operatorname{arcsinh}\left(\frac{x}{\alpha_{k+1}^2}\right) - \operatorname{arcsinh}\left(\frac{x}{\alpha_k^2}\right)\right| &\leqslant \ln\left(\frac{\alpha_k^2}{\alpha_{k+1}^2}\right) \\
&= \delta_k,
\end{aligned}$$

since $\delta_k \geqslant 0$ due to our stepsize condition.

We now prove that $|\phi_{k+1} - \phi_k| \leqslant \frac{|\delta_{+,k} - \delta_{-,k}|}{2}$. We have $\phi_k = \operatorname{arcsinh}\left(\frac{\alpha_{+,k}^2 - \alpha_{-,k}^2}{2\alpha_{+,k}\alpha_{-,k}}\right)$ and hence,

$$|\phi_{k+1} - \phi_k| = \left|\operatorname{arcsinh}\left(\frac{\alpha_{+,k}^2 - \alpha_{-,k}^2}{2\alpha_{+,k}\alpha_{-,k}}\right) - \operatorname{arcsinh}\left(\frac{\alpha_{+,k+1}^2 - \alpha_{-,k+1}^2}{2\alpha_{+,k+1}\alpha_{-,k+1}}\right)\right|.$$

Then, assuming that $\alpha_{+,k,i} \geqslant \alpha_{-,k,i}$, we have:

$$\frac{\alpha_{+,k+1,i}^2 - \alpha_{-,k+1,i}^2}{2\alpha_{+,k+1,i}\alpha_{-,k+1,i}} = e^{\delta_{k,i}/2}\frac{\alpha_{+,k,i}^2 e^{-\delta_{+,k,i}} - \alpha_{-,k,i}^2 e^{-\delta_{-,k,i}}}{2\alpha_{+,k,i}\alpha_{-,k,i}}$$

$$\begin{cases} \leqslant \begin{cases} e^{\frac{\delta_{+,k,i}-\delta_{-,k,i}}{2}}\dfrac{\alpha_{+,k,i}^2 - \alpha_{-,k,i}^2}{2\alpha_{+,k,i}\alpha_{-,k,i}} & \text{if} \quad \delta_{+,k,i} \geqslant \delta_{-,k,i} \\[2ex] e^{\frac{\delta_{-,k,i}-\delta_{+,k,i}}{2}}\dfrac{\alpha_{+,k,i}^2 - \alpha_{-,k,i}^2}{2\alpha_{+,k,i}\alpha_{-,k,i}} & \text{if} \quad \delta_{-,k,i} \geqslant \delta_{+,k,i} \end{cases} \\[6ex] \geqslant \begin{cases} e^{-\frac{\delta_{+,k,i}-\delta_{-,k,i}}{2}}\dfrac{\alpha_{+,k,i}^2 - \alpha_{-,k,i}^2}{2\alpha_{+,k,i}\alpha_{-,k,i}} & \text{if} \quad \delta_{+,k,i} \geqslant \delta_{-,k,i} \\[2ex] e^{-\frac{\delta_{-,k,i}-\delta_{+,k,i}}{2}}\dfrac{\alpha_{+,k,i}^2 - \alpha_{-,k,i}^2}{2\alpha_{+,k,i}\alpha_{-,k,i}} & \text{if} \quad \delta_{-,k,i} \geqslant \delta_{+,k,i} \end{cases} \end{cases}.$$

We thus have $\frac{\alpha_{+,k+1,i}^2-\alpha_{-,k+1,i}^2}{2\alpha_{+,k+1,i}\alpha_{-,k+1,i}} \in \left[e^{-\frac{|\delta_{+,k,i}-\delta_{-,k,i}|}{2}}, e^{\frac{|\delta_{+,k,i}-\delta_{-,k,i}|}{2}}\right] \times \frac{\alpha_{+,k,i}^2-\alpha_{-,k,i}^2}{2\alpha_{+,k,i}\alpha_{-,k,i}}$, and this holds

similarly if $\alpha_{+,k,i} \leqslant \alpha_{-,k,i}$. Then, using $|\operatorname{arcsinh}(a) - \operatorname{arcsinh}(b)| \leqslant |\ln(a/b)|$ we obtain that:

$$|\phi_{k+1} - \phi_k| = \left|\operatorname{arcsinh}\left(\frac{\alpha_{+,k}^2 - \alpha_{-,k}^2}{2\alpha_{+,k}\alpha_{-,k}}\right) - \operatorname{arcsinh}\left(\frac{\alpha_{+,k+1}^2 - \alpha_{-,k+1}^2}{2\alpha_{+,k+1}\alpha_{-,k+1}}\right)\right|$$

$$\leqslant \frac{|\delta_{+,k} - \delta_{-,k}|}{2}.$$

Wrapping things up, we have:

$$|\nabla h_k(\beta) - \nabla h_{k+1}(\beta)| \leqslant \delta_k + \frac{|\delta_{+,k} - \delta_{-,k}|}{2} \leqslant 2\delta_k,$$

This leads to the following bound:

$$|h_{k+1}(\beta) - h_k(\beta)| \leqslant \langle|2\delta_k|, |\beta|\rangle$$
$$\leqslant 2\|\delta_k\|_1\|\beta\|_\infty.$$

Recall that $\delta_k = q(\gamma_k\nabla\mathcal{L}_{\mathcal{B}_k}(\beta_k))$, hence from Lemma 5 if $\gamma_k\|\nabla\mathcal{L}_{\mathcal{B}_k}(\beta_k)\|_\infty \leqslant 1/2$, we get that

$$\|\delta_k\|_1 \leqslant 2\gamma_k^2\|\nabla\mathcal{L}_{\mathcal{B}_k}(\beta_k)\|_2^2 \leqslant 4L_2\gamma_k^2\mathcal{L}_{\mathcal{B}_k}(\beta_k).$$

Putting things together we obtain that

$$|h_{k+1}(\beta) - h_k(\beta)| \leqslant \langle|2\delta_k|, |\beta|\rangle$$
$$\leqslant 8L_2\gamma_k^2\mathcal{L}_{\mathcal{B}_k}(\beta_k)\|\beta\|_\infty.$$

$\square$

# I  Concentration inequalities for matrices

In this last section of the appendix, we provide and prove several concentration bounds for random vectors and matrices, with (possibly uncentered) isotropic gaussian inputs. These inequalities can easily be generalized to subgaussian random variables via more refined concentration bounds, and to non-isotropic subgaussian random variables [19], leading to a dependence on an effective dimension and on the subgaussian matrix $\Sigma$. We present these lemmas before proving them in a row.

The next two lemmas closely ressemble the RIP assumption, for centered and then for uncentered gaussians.

**Lemma 8.** *Let $x_1, \ldots, x_n \in \mathbb{R}^d$ be i.i.d. random variables of law $\mathcal{N}(0, I_d)$ and $H = \frac{1}{n}\sum_{i=1}^n x_i x_i^\top$. Then, denoting by $\mathcal{C}$ the set of all $s$-sparse vector $\beta \in \mathbb{R}^d$ satisfying $\|\beta\|_2 \leqslant 1$, there exist $C_4, C_5 > 0$ such that for any $\varepsilon > 0$, if $n \geqslant C_4 s \ln(d)\varepsilon^{-2}$,*

$$\mathbb{P}\left(\sup_{\beta\in\mathcal{S}}\|H\beta - \beta\|_\infty \geqslant \varepsilon\right) \leqslant e^{-C_5 n}.$$

**Lemma 9.** *Let* $x_1, \dots, x_n \in \mathbb{R}^d$ *be i.i.d. random variables of law* $\mathcal{N}(\mu, \sigma^2 I_d)$ *and* $H = \frac{1}{n} \sum_{i=1}^n x_i x_i^\top$. *Then, denoting by* $\mathcal{C}$ *the set of all $s$-sparse vector* $\beta \in \mathbb{R}^d$ *satisfying* $\|\beta\|_2 \leqslant 1$, *there exist* $C_4, C_5 > 0$ *such that for any* $\varepsilon > 0$, *if* $n \geqslant C_4 s \ln(d)\varepsilon^{-2}$,

$$\mathbb{P}\left(\sup_{\beta \in \mathcal{S}} \|H\beta - \mu\langle \mu, \beta\rangle - \sigma^2\beta\|_\infty \geqslant \varepsilon\right) \leqslant e^{-C_5 n}.$$

We then provide two lemmas that estimate the mean Hessian of SGD.

**Lemma 10.** *Let* $x_1, \dots, x_n$ *be i.i.d. random variables of law* $\mathcal{N}(0, I_d)$. *Then, there exist* $c_1, c_2 > 0$ *such that with probability* $1 - \frac{1}{d^2}$ *and if* $n = \Omega(s^{5/4}\ln(d))$, *we have for all $s$-sparse vectors* $\beta$:

$$c_1 \|\beta\|_2^2 \mathbf{1} \leqslant \frac{1}{n}\sum_{i=1}^n x_i^2 \langle x_i, \beta\rangle^2 \leqslant c_2 \|\beta\|_2^2 \mathbf{1},$$

*where the inequality is meant component-wise.*

**Lemma 11.** *Let* $x_1, \dots, x_n$ *be i.i.d. random variables of law* $\mathcal{N}(\mu, \sigma^2 I_d)$. *Then, there exist* $c_0, c_1, c_2 > 0$ *such that with probability* $1 - \frac{c_0}{d^2} - \frac{1}{nd}$ *and if* $n = \Omega(s^{5/4}\ln(d))$ *and* $\mu \geqslant 4\sigma\sqrt{\ln(d)}\mathbf{1}$, *we have for all $s$-sparse vectors* $\beta$:

$$\frac{\mu^2}{2}\left(\langle \mu, \beta\rangle^2 + \frac{1}{2}\sigma^2\|\beta\|_2^2\right) \leqslant \frac{1}{n}\sum_i x_i^2 \langle x_i, \beta\rangle^2 \leqslant 4\mu^2\left(\langle \mu, \beta\rangle^2 + 2\sigma^2\|\beta\|_2^2\right).$$

*where the inequality is meant component-wise.*

Finally, next two lemmas are used to estimate $\lambda_b, \Lambda_b$ in our paper.

**Lemma 12.** *Let* $x_1, \dots, x_n \in \mathbb{R}^d$ *be i.i.d. random variables of law* $\mathcal{N}(\mu\mathbf{1}, \sigma^2 I_d)$. *Let* $H = \frac{1}{n}\sum_{i=1}^n x_i x_i^\top$ *and* $\tilde{H} = \frac{1}{n}\sum_{i=1}^n \|x_i\|^2 x_i x_i^\top$. *There exist numerical constants* $C_2, C_3 > 0$ *such that*

$$\mathbb{P}\left(C_2(\mu^2 + \sigma^2)dH \preceq \tilde{H} \preceq C_3(\mu^2 + \sigma^2)dH\right) \geqslant 1 - 2ne^{-d/16}.$$

**Lemma 13.** *Let* $x_1, \dots, x_n \in \mathbb{R}^d$ *be i.i.d. random variables of law* $\mathcal{N}(\mu\mathbf{1}, \sigma^2 I_d)$ *for some* $\mu \in \mathbb{R}$. *Let* $H = \frac{1}{n}\sum_{i=1}^n x_i x_i^\top$ *and for* $1 \leqslant b \leqslant n$ *let* $\tilde{H}_b = \mathbb{E}_{\mathcal{B}}\left[\left(\frac{1}{b}\sum_{i \in \mathcal{B}} x_i x_i^\top\right)^2\right]$ *where* $\mathcal{B} \subset [n]$ *is sampled uniformly at random in* $\{\mathcal{B} \subset [n] \text{ s.t. } |\mathcal{B}| = b\}$. *With probability* $1 - 2ne^{-d/16} - 3/n^2$, *we have, for some numerical constants* $c_1, c_2, c_3, C > 0$:

$$\left(c_1 \frac{d(\mu^2 + \sigma^2)}{b} - c_2 \frac{(\sigma^2 + \mu^2)\ln(n)}{\sqrt{d}} - c_3 \frac{\mu^2 d}{n}\right)H \preceq \tilde{H}_b \preceq C\left(\frac{d(\mu^2 + \sigma^2)}{b} + \frac{(\sigma^2 + \mu^2)\ln(n)}{\sqrt{d}} + \mu^2 d\right)$$

*Proof of Lemma 8.* For $j \in [d]$, we have:

$$\begin{aligned}
(H\beta)_j &= \frac{1}{n}\sum_{i=1}^n x_{ij}\langle x_i, \beta\rangle \\
&= \frac{1}{n}\sum_{i=1}^n \sum_{j'=1}^d x_{ij}x_{ij'}\beta_{j'} \\
&= \frac{1}{n}\sum_{i=1}^n x_{ij}^2\beta_j + \frac{1}{n}\sum_{i=1}^n \sum_{j'\neq j} x_{ij}x_{ij'}\beta_{j'} \\
&= \frac{\beta_j}{n}\sum_{i=1}^n x_{ij}^2 + \frac{1}{n}\sum_{i=1}^n x_{ij}\sum_{j'\neq j} x_{ij'}\beta_{j'}.
\end{aligned}$$

We thus notice that $\mathbb{E}[H\beta] = \beta$, and

$$(H\beta)_j = \beta_j + \frac{\beta_j}{n}\sum_{i=1}^n (x_{ij}^2 - 1) + \frac{1}{n}\sum_{i=1}^n z_i,$$

where $z_i = x_{ij} \sum_{j' \neq j} x_{ij'} \beta_{j'}$, and $\sum_{j' \neq j} x_{ij'} \beta_{j'} \sim \mathcal{N}(0, \|\beta\|^2 - \beta_j^2)$ and $\|\beta\|^2 - \beta_j^2 \leqslant 1$. Hence, $z_j + x_{ij}^2 - 1$ is a centered subexponential random variables (with a subexponential parameter of order 1). Thus, for $t \leqslant 1$:

$$\mathbb{P}\left( \left| \frac{\beta_j}{n} \sum_{i=1}^n (x_{ij}^2 - 1) + \frac{1}{n} \sum_{i=1}^n z_i \right| \geqslant t \right) \leqslant 2e^{-cnt^2} \, .$$

Hence, using an $\varepsilon$-net of $\mathcal{C} = \left\{ \beta \in \mathbb{R}^d : \|\beta\|_2 \leqslant 1, \|\beta\|_0 \right\}$ (of cardinality less than $d^s \times (C/\varepsilon)^s$, and for $\varepsilon$ of order 1), we have, using the classical $\varepsilon$-net trick explained in [Chapt. 9, [58]] or [App. C, Even and Massoulie [19]]:

$$\mathbb{P}\left( \sup_{\beta \in \mathcal{C}, \, j \in [d]} |(H\beta)_j - \beta_j| \geqslant t \right) \leqslant d \times d^s (C/\varepsilon)^s \times 2e^{-cnt^2} = \exp\left( -c \ln(2)nt^2 + (s+1)\ln(d) + s\ln(C/\varepsilon) \right) \, .$$

Consequently, for $t = \varepsilon$ and if $n \geqslant C_4 s \ln(d)/\varepsilon^2$, we have:

$$\mathbb{P}\left( \sup_{\beta \in \mathcal{C}, \, j \in [d]} |(H\beta)_j - \beta_j| \geqslant t \right) \leqslant \exp\left( -C_5 nt^2 \right) \, .$$

$\square$

*Proof of Lemma 9.* We write $x_i = \sigma z_i + \mu$ where $z_i \sim \mathcal{N}(0, I_d)$. We have:

$$X^\top X \beta = \frac{1}{n} \sum_{i=1}^n (\mu + \sigma z_i) \langle \mu + \sigma z_i, \beta \rangle$$

$$= \mu \langle \mu, \beta \rangle + \frac{\sigma^2}{n} \sum_{i=1}^n z_i \langle z_i, \beta \rangle + \frac{\sigma}{n} \sum_{i=1}^n \mu \langle z_i, \beta \rangle + \frac{\sigma}{n} \sum_{i=1}^n z_i \langle \mu, \beta \rangle$$

$$= \mu \langle \mu, \beta \rangle + \frac{\sigma^2}{n} \sum_{i=1}^n z_i \langle z_i, \beta \rangle + \sigma \mu \langle \frac{1}{n} \sum_{i=1}^n z_i, \beta \rangle + \frac{\sigma \langle \mu, \beta \rangle}{n} \sum_{i=1}^n z_i \, .$$

The first term is deterministic and is to be kept. The second one is of order $\sigma^2 \beta$ whp using Lemma 8. Then, $\frac{1}{n} \sum_{i=1}^n z_i \sim \mathcal{N}(0, I_d/n)$, so that

$$\mathbb{P}\left( \left| \langle \frac{1}{n} \sum_{i=1}^n z_i, \beta \rangle \right| \geqslant t \right) \leqslant 2e^{-nt^2/(2\|\beta\|_2^2)} \, ,$$

and

$$\mathbb{P}\left( \left| \frac{1}{n} \sum_{i=1}^n z_{ij} \right| \geqslant t \right) \leqslant 2e^{-nt^2/2} \, .$$

Hence,

$$\mathbb{P}\left( \sup_{\beta \in \mathcal{C}} \left\| \frac{1}{n} \sum_{i=1}^n z_{ij} \right\|_\infty \geqslant t, \, \sup_{\beta \in \mathcal{C}} \left| \langle \frac{1}{n} \sum_{i=1}^n z_i, \beta \rangle \right| \geqslant t \right) \leqslant 4e^{cs \ln(d)} e^{-nt^2/2} \, .$$

Thus, with probability $1 - Ce^{-n\varepsilon^2}$ and under the assumptions of Lemma 8, we have $\left\| X^\top X \beta - \mu \langle \mu, \beta \rangle - \sigma^2 \beta \right\|_\infty \leqslant \varepsilon$

$\square$

*Proof of Lemma 10.* To ease notations, we assume that $\sigma = 1$. We remind (O'Donnell [46], Chapter 9 and Tao [54]) that for *i.i.d.* real random variables $a_1, \ldots, a_n$ that satisfy a tail inequality of the form

$$\mathbb{P}\left( |a_1 - \mathbb{E}a_1| \geqslant t \right) \leqslant Ce^{-ct^p} \, , \tag{27}$$

for $p < 1$, then for all $\varepsilon > 0$ there exists $C', c'$ such that for all $t$,

$$\mathbb{P}\left( \left| \frac{1}{n} \sum_{i=1}^n a_i - \mathbb{E}a_1 \right| \geqslant t \right) \leqslant C' e^{-c' nt^{p-\varepsilon}} \, .$$

We now expand $\frac{1}{n}\sum_{i=1}^{n}x_i^2\langle x_i,\beta\rangle^2$:

$$\frac{1}{n}\sum_{i=1}^{n}x_i^2\langle x_i,\beta\rangle^2 = \frac{1}{n}\sum_{i\in[n],k,\ell\in[d]}x_i^2 x_{ik}x_{i\ell}\beta_k\beta_\ell$$

$$= \frac{1}{n}\sum_{i\in[n],k\in[d]}x_i^2 x_{ik}^2\beta_k^2 + \frac{1}{n}\sum_{i\in[n],k\neq\ell\in[d]}x_i^2 x_{ik}x_{i\ell}\beta_k\beta_\ell\,.$$

Thus, for $j\in[d]$,

$$\left(\frac{1}{n}\sum_{i=1}^{n}x_i^2\langle x_i,\beta\rangle^2\right)_j = \sum_{k\in[d]}\frac{\beta_k^2}{n}\sum_{i\in[n]}x_{ij}^2 x_{ik}^2 + \sum_{k\neq\ell\in[d]}\frac{\beta_k\beta_\ell}{n}\sum_{i\in[n]}x_{ij}^2 x_{ik}x_{i\ell}\,.$$

We notice that for all indices, all $x_{ij}^2 x_{ik}x_{i\ell}$ and $x_{ij}^2 x_{ik}^2$ satisfy the tail inequality Eq. (27) for $C=8$, $c=1/2$ and $p=1/2$, so that for $\varepsilon=1/4$:

$$\mathbb{P}(|\frac{1}{n}\sum_{i=1}^{n}x_{ij}^2 x_{ik}x_{i\ell}| \geqslant t) \leqslant C'e^{-c'nt^{1/4}} \quad,\quad \mathbb{P}(|\frac{1}{n}\sum_{i=1}^{n}x_{ij}^2 x_{ik}^2 - \mathbb{E}\left[x_{ij}^2 x_{ik}^2\right]| \geqslant t) \leqslant C'e^{-c'nt^{1/4}}\,.$$

For $j\neq k$, we have $\mathbb{E}\left[x_{ij}^2 x_{ik}^2\right]=1$ while for $j=k$, we have $\mathbb{E}\left[x_{ij}^2 x_{ik}^2\right]=\mathbb{E}\left[x_{ij}^4\right]=3$. Hence,

$$\mathbb{P}\left(\exists j,k\neq\ell,\,|\frac{1}{n}\sum_{i=1}^{n}x_{ij}^2 x_{ik}x_{i\ell}| \geqslant t\,,\,|\frac{1}{n}\sum_{i=1}^{n}x_{ij}^2 x_{ik}^2 - \mathbb{E}\left[x_{ij}^2 x_{ik}^2\right]| \geqslant t\right) \leqslant C'd^2 e^{-c'nt^{1/4}}\,.$$

Thus, with probability $1 - C'd^2 e^{-c'nt^{1/4}}$, for all $j\in[d]$,

$$\left|\left(\frac{1}{n}\sum_{i=1}^{n}x_i^2\langle x_i,\beta\rangle^2\right)_j - 2\beta_j^2 - \|\beta\|_2^2\right| \leqslant t\sum_{k,\ell}|\beta_k||\beta_\ell| = t\|\beta\|_1^2\,.$$

Using the classical technique of Baraniuk et al. [4], to make a union bound on all $s$-sparse vectors, we consider an $\varepsilon$-net of the set of $s$-sparse vectors of $\ell^2$-norm smaller than 1. This $\varepsilon$-net is of cardinality less than $(C_0/\varepsilon)^s d^s$, and we only need to take $\varepsilon$ of order 1 to obtain the result for all $s$-sparse vectors. This leads to:

$$\mathbb{P}\left(\exists\beta\in\mathbb{R}^d\ s\text{-sparse and } \|\beta\|_2\leqslant 1\,,\,\exists j\in\mathbb{R}^d,\,\left|\left(\frac{1}{n}\sum_{i=1}^{n}x_i^2\langle x_i,\beta\rangle^2\right)_j - 2\beta_j^2 - \|\beta\|_2^2\right| \geqslant t\|\beta\|_1^2\right)$$
$$\leqslant C'd^2 e^{c_1 s + s\ln(d)}e^{-c'nt^{1/4}}\,.$$

This probability is equal to $C'/d^2$ for $t = \left(\frac{(s+4)\ln(d)+c_1 s}{c'n}\right)^4$. We conclude that with probability $1 - C'/d^2$, all $s$-sparse vectors $\beta$ satisfy:

$$\left|\left(\frac{1}{n}\sum_{i=1}^{n}x_i^2\langle x_i,\beta\rangle^2\right)_j - 2\beta_j^2 - \|\beta\|_2^2\right| \leqslant \left(\frac{(s+4)\ln(d)+c_1 s}{c'n}\right)^4\|\beta\|_1^2 \leqslant \left(\frac{(s+4)\ln(d)+c_1 s}{c'n}\right)^4 s\|\beta\|_2^2\,,$$

and the RHS is smaller than $\|\beta\|_2^2/2$ for $n \geqslant \Omega(s^{5/4}\ln(d))$. $\qquad\square$

*Proof of Lemma 11.* We write $x_i = \mu + \sigma z_i$ where $x_i \sim \mathcal{N}(0,1)$. We have:

$$\mathbb{P}\left(\forall i\in[n],\forall j\in[d],\,|z_{ij}|\geqslant t\right) \leqslant e^{\ln(nd)-t^2/2} = \frac{1}{nd}\,,$$

for $t = 2\sqrt{\ln(nd)}$. Thus, if $\mu \geqslant 4\sigma\sqrt{\ln(nd)}$ we have $\frac{\mu}{2} \leqslant x_i \leqslant 2\mu$, so that

$$\frac{\mu^2}{2n}\sum_i\langle x_i,\beta\rangle^2 \leqslant \frac{1}{n}\sum_i x_i^2\langle x_i,\beta\rangle^2 \leqslant \frac{4\mu^2}{n}\sum_i\langle x_i,\beta\rangle^2\,.$$

Then, $\langle x_i, \beta \rangle \sim \mathcal{N}(\langle \mu, \beta \rangle, \sigma^2 \|\beta\|_2^2)$. For now, we assume that $\|\beta\|_2 = 1$. We have $\mathbb{P}(|\langle x_i, \beta \rangle^2 - \langle \mu, \beta \rangle^2 - \sigma^2 \|\beta\|_2^2| \geqslant t) \leqslant Ce^{-ct/\sigma^2}$, and for $t \leqslant 1$, using concentration of subexponential random variables [58]:

$$\mathbb{P}\left(\left| \frac{1}{n} \sum_i \langle x_i, \beta \rangle^2 - \langle \mu, \beta \rangle^2 - \sigma^2 \|\beta\|_2^2 \right| \geqslant t \right) \leqslant C' e^{-nc't^2/\sigma^4},$$

and using the $\varepsilon$-net trick of Baraniuk et al. [4],

$$\mathbb{P}\left( \sup_{\beta \in \mathcal{C}} \left| \frac{1}{n} \sum_i \langle x_i, \beta \rangle^2 - \langle \mu, \beta \rangle^2 - \sigma^2 \|\beta\|_2^2 \right| \geqslant t \right) \leqslant C' e^{s \ln(d) - nc't^2/\sigma^4} = \frac{C'}{d^2},$$

for $t = \sigma^2 \|\beta\|_2^2 \sqrt{\frac{2(cs+2)\ln(d)}{n}}$. Consequently, we have, with probability $1 - \frac{C'}{d^2} - \frac{1}{nd}$:

$$\frac{\mu^2}{2}\left( \langle \mu, \beta \rangle^2 + \frac{1}{2}\sigma^2 \|\beta\|_2^2 \right) \leqslant \frac{1}{n} \sum_i x_i^2 \langle x_i, \beta \rangle^2 \leqslant 4\mu^2 \left( \langle \mu, \beta \rangle^2 + 2\sigma^2 \|\beta\|_2^2 \right).$$

$\square$

*Proof of Lemma 12.* First, we write $x_i = \mu \mathbf{1} + \sigma z_i$, where $z_i \sim \mathcal{N}(0, I)$, leading to:

$$\frac{1}{n} \sum_{i \in [n]} \|x_i\|_2^2 x_i x_i^\top = \frac{1}{n} \sum_{i \in [n]} \left( \sigma^2 \|z_i\|_2^2 + d\mu^2 + 2\sigma\mu \langle \mathbf{1}, z_i \rangle \right) x_i x_i^\top$$

We use concentration of $\chi_d^2$ random variables around $d$:

$$\mathbb{P}(\chi_d^2 > d + 2t + 2\sqrt{dt}) \geqslant t) \leqslant e^{-t} \quad \text{and} \quad \mathbb{P}(\chi_d^2 > d - 2\sqrt{dt}) \leqslant t) \leqslant e^{-t},$$

so that for all $i \in [n]$,

$$\mathbb{P}(\|z_i\|_2^2 \notin [d - 2\sqrt{dt}, d + 2t + 2\sqrt{dt}]) \leqslant 2e^{-t}.$$

Thus,

$$\mathbb{P}(\forall i \in [n], \|z_i\|_2^2 \in [d - 2\sqrt{dt}, d + 2t + 2\sqrt{dt}]) \geqslant 1 - 2ne^{-t}.$$

Taking $t = d/16$,

$$\mathbb{P}(\forall i \in [n], \|z_i\|_2^2 \in [\frac{d}{2}, 13d/8]) \geqslant 1 - 2ne^{-d/16}.$$

Then, for all $i$, $\langle \mathbf{1}, z_i \rangle$ is of law $\mathcal{N}(0, d)$, so that $\mathbb{P}(|\langle \mathbf{1}, z_i \rangle| \geqslant t) \leqslant 2e^{-t^2/(2d)}$ and

$$\mathbb{P}(\forall i \in [n], |\langle \mathbf{1}, z_i \rangle| \geqslant t) \leqslant 2ne^{-\frac{t^2}{2d}}.$$

Taking $t = \sqrt{2}d^{3/4}$,

$$\mathbb{P}(\forall i \in [n], |\langle \mathbf{1}, z_i \rangle| \geqslant d^{3/4}) \leqslant 2ne^{-d^{1/2}}.$$

Thus, with probability $1 - 2n(e^{-d/16} + e^{-\sqrt{d}})$, we have $\forall i \in [n]$, $|\langle \mathbf{1}, z_i \rangle| \geqslant d^{3/4}$ and $\|z_i\|_2^2 \in [\frac{d}{2}, 13d/8]$, so that

$$\left( \frac{d}{2}\sigma^2 + d\mu^2 - 2\mu\sigma d^{3/4} \right) H \preceq \tilde{H} \preceq \left( \frac{13d}{8}\sigma^2 + d\mu^2 + 2\mu\sigma d^{3/4} \right) H,$$

leading to the desired result. $\square$

*Proof of Lemma 13.* We have:

$$\tilde{H}_b = \mathbb{E}\left[ \frac{1}{b^2} \sum_{i,j \in \mathcal{B}} \langle x_i, x_j \rangle x_i x_j^\top \right]$$

$$= \mathbb{E}\left[ \frac{1}{b^2} \sum_{i \in \mathcal{B}} \|x_i\|_2^2 x_i x_i^\top + \frac{1}{b^2} \sum_{i,j \in \mathcal{B}, i \neq j} \langle x_i, x_j \rangle x_i x_j^\top \right]$$

$$= \frac{1}{b^2} \sum_{i \in [n]} \mathbb{P}(i \in \mathcal{B})\|x_i\|_2^2 x_i x_i^\top + \frac{1}{b^2} \sum_{i \neq j} \mathbb{P}(i, j \in \mathcal{B})\langle x_i, x_j \rangle x_i x_j^\top.$$

Then, since $\mathbb{P}(i \in \mathcal{B}) = \frac{b}{n}$ and $\mathbb{P}(i, j \in \mathcal{B}) = \frac{b(b-1)}{n(n-1)}$ for $i \neq j$, we get that:

$$\tilde{H}_b = \frac{1}{bn} \sum_{i \in [n]} \|x_i\|_2^2 x_i x_i^\top + \frac{(b-1)}{bn(n-1)} \sum_{i \neq j} \langle x_i, x_j \rangle x_i x_j^\top .$$

Using Lemma 12, the first term satisfies:

$$\mathbb{P}\left( \frac{d(\mu^2 + \sigma^2)}{b} C_2 H \preceq \frac{1}{bn} \sum_{i \in [n]} \|x_i\|_2^2 x_i x_i^\top \preceq \frac{d(\mu^2 + \sigma^2)}{b} C_3 H \right) \geqslant 1 - 2ne^{-d/16} .$$

We now show that the second term is of smaller order. Writing $x_i = \mu \mathbf{1} + \sigma z_i$ where $z_i \sim \mathcal{N}(0, I_d)$, we have:

$$\frac{(b-1)}{bn(n-1)} \sum_{i \neq j} \langle x_i, x_j \rangle x_i x_j^\top = \frac{(b-1)}{bn(n-1)} \sum_{i \neq j} \langle x_i, x_j \rangle x_i x_j^\top$$

For $i \neq j$, $\langle x_i, x_j \rangle = \sum_{k=1}^{d} x_{ik} x_{jk} = \sum_{k=1}^{d} a_k$ where $a_k = x_{ik} x_{jk}$ satisfies $\mathbb{E} a_k = 0$, $\mathbb{E} a_k^2 = 1$ and $\mathbb{P}(a_k \geqslant t) \leqslant 2\mathbb{P}(|x_{ik}| \geqslant \sqrt{t}) \leqslant 4e^{-t/2}$. Hence, $a_k$ is a centered subexponential random variables. Using concentration of subexponential random variables [58], for $t \leqslant 1$,

$$\mathbb{P}\left( \frac{1}{d} |\langle x_i, x_j \rangle| \geqslant t \right) \leqslant 2e^{-cdt^2} .$$

Thus,

$$\mathbb{P}\left( \forall i \neq j, \frac{1}{d} |\langle x_i, x_j \rangle| \leqslant t \right) \geqslant 1 - n(n-1)e^{-cdt^2} .$$

Then, taking $t = d^{-1/2} 4 \ln(n)/c$, we have:

$$\mathbb{P}\left( \forall i \neq j, \frac{1}{d} |\langle x_i, x_j \rangle| \leqslant \frac{4 \ln(n)}{c\sqrt{d}} \right) \geqslant 1 - \frac{1}{n^2} .$$

Going back to our second term,

$$\frac{(b-1)}{bn(n-1)} \sum_{i \neq j} \langle x_i, x_j \rangle x_i x_j^\top = \frac{(b-1)}{bn(n-1)} \sum_{i < j} \langle x_i, x_j \rangle \left( x_i x_j^\top + x_j x_i^\top \right)$$

$$\preceq \frac{(b-1)}{bn(n-1)} \sum_{i < j} |\langle x_i, x_j \rangle| \left( x_i x_i^\top + x_j x_j^\top \right) ,$$

where we used $x_i x_j^\top + x_j x_i^\top \preceq x_i x_i^\top + x_j x_j^\top$. Thus,

$$\frac{(b-1)}{bn(n-1)} \sum_{i \neq j} \langle x_i, x_j \rangle x_i x_j^\top \preceq \sup_{i \neq j} |\langle x_i, x_j \rangle| \times \frac{(b-1)}{bn(n-1)} \sum_{i < j} \left( x_i x_i^\top + x_j x_j^\top \right)$$

$$= \sup_{i \neq j} |\langle x_i, x_j \rangle| \times \frac{b-1}{b} \frac{1}{n-1} \sum_{i=1}^{n} x_i x_i^\top$$

$$= \sup_{i \neq j} |\langle x_i, x_j \rangle| \times \frac{b-1}{b} \frac{n}{n-1} H .$$

Similarly, we have

$$\frac{(b-1)}{bn(n-1)} \sum_{i \neq j} \langle x_i, x_j \rangle x_i x_j^\top \succeq -\sup_{i \neq j} |\langle x_i, x_j \rangle| \times \frac{b-1}{b} \frac{n}{n-1} H .$$

Hence, with probability $1 - 1/n^2$,

$$-\frac{4 \ln(n)}{c\sqrt{d}} \times \frac{b-1}{b} \frac{n}{n-1} H \preceq \frac{(b-1)}{bn(n-1)} \sum_{i \neq j} \langle x_i, x_j \rangle x_i x_j^\top \preceq \frac{4 \ln(n)}{c\sqrt{d}} \times \frac{b-1}{b} \frac{n}{n-1} H .$$

Wrapping things up, with probability $1 - 1/n^2 - 2ne^{-d/16}$,

$$\left(-\frac{4\ln(n)}{c\sqrt{d}}\frac{b-1}{b}\frac{n}{n-1} + C_2\frac{d}{b}\right) \times H \preceq \tilde{H}_b \preceq \left(\frac{4\ln(n)}{c\sqrt{d}}\frac{b-1}{b}\frac{n}{n-1} + C_3\frac{d}{b}\right) \times H.$$

Thus, provided that $\frac{4\ln(n)}{c\sqrt{d}} \leqslant \frac{C_2 d}{2b}$ and $d \geqslant 48\ln(n)$, we have with probability $1 - 3/n^2$:

$$C_2'\frac{d}{b} \times H \preceq \tilde{H}_b \preceq C_3'\frac{d}{b} \times H.$$

$\square$

*Proof of Lemma 13.* We have:

$$\tilde{H}_b = \mathbb{E}\left[\frac{1}{b^2}\sum_{i,j\in\mathcal{B}}\langle x_i, x_j\rangle x_i x_j^\top\right]$$

$$= \mathbb{E}\left[\frac{1}{b^2}\sum_{i\in\mathcal{B}}\|x_i\|_2^2 x_i x_i^\top + \frac{1}{b^2}\sum_{i,j\in\mathcal{B}, i\neq j}\langle x_i, x_j\rangle x_i x_j^\top\right]$$

$$= \frac{1}{b^2}\sum_{i\in[n]}\mathbb{P}(i\in\mathcal{B})\|x_i\|_2^2 x_i x_i^\top + \frac{1}{b^2}\sum_{i\neq j}\mathbb{P}(i,j\in\mathcal{B})\langle x_i, x_j\rangle x_i x_j^\top.$$

Then, since $\mathbb{P}(i\in\mathcal{B}) = \frac{b}{n}$ and $\mathbb{P}(i,j\in\mathcal{B}) = \frac{b(b-1)}{n(n-1)}$ for $i\neq j$, we get that:

$$\tilde{H}_b = \frac{1}{bn}\sum_{i\in[n]}\|x_i\|_2^2 x_i x_i^\top + \frac{(b-1)}{bn(n-1)}\sum_{i\neq j}\langle x_i, x_j\rangle x_i x_j^\top.$$

Using Lemma 12, the first term satisfies:

$$\mathbb{P}\left(\frac{d(\mu^2 + \sigma^2)}{b}C_2 H \preceq \frac{1}{bn}\sum_{i\in[n]}\|x_i\|_2^2 x_i x_i^\top \preceq \frac{d(\mu^2 + \sigma^2)}{b}C_3 H\right) \geqslant 1 - 2ne^{-d/16}.$$

We now show that the second term is of smaller order. Writing $x_i = \mu\mathbf{1} + \sigma z_i$ where $z_i \sim \mathcal{N}(0, I_d)$, we have:

$$\frac{(b-1)}{bn(n-1)}\sum_{i\neq j}\langle x_i, x_j\rangle x_i x_j^\top = \frac{(b-1)}{bn(n-1)}\sum_{i\neq j}\left(\sigma^2\langle z_i, z_j\rangle + \sigma\mu\langle\mathbf{1}, z_i + z_j\rangle + \mu^2 d\right)x_i x_j^\top$$

$$= \frac{(b-1)}{bn(n-1)}\sum_{i\neq j}\left(\sigma^2\langle z_i, z_j\rangle + \sigma\mu\langle\mathbf{1}, z_i + z_j\rangle\right)x_i x_j^\top + \frac{(b-1)}{bn(n-1)}\mu^2 d\sum_{i\neq j}x_i x_j^\top$$

For $i\neq j$, $\langle z_i, z_j\rangle = \sum_{k=1}^d z_{ik}z_{jk} = \sum_{k=1}^d a_k$ where $a_k = z_{ik}z_{jk}$ satisfies $\mathbb{E}a_k = 0$, $\mathbb{E}a_k^2 = 1$ and $\mathbb{P}(a_k \geqslant t) \leqslant 2\mathbb{P}(|z_{ik}| \geqslant \sqrt{t}) \leqslant 4e^{-t/2}$. Hence, $a_k$ is a centered subexponential random variables. Using concentration of subexponential random variables [58], for $t \leqslant 1$,

$$\mathbb{P}\left(\frac{1}{d}|\langle x_i, x_j\rangle| \geqslant t\right) \leqslant 2e^{-cdt^2}.$$

Thus,

$$\mathbb{P}\left(\forall i\neq j, \frac{1}{d}|\langle x_i, x_j\rangle| \leqslant t\right) \geqslant 1 - n(n-1)e^{-cdt^2}.$$

Then, taking $t = d^{-1/2}4\ln(n)/c$, we have:

$$\mathbb{P}\left(\forall i\neq j, \frac{1}{d}|\langle x_i, x_j\rangle| \leqslant \frac{4\ln(n)}{c\sqrt{d}}\right) \geqslant 1 - \frac{1}{n^2}.$$

For $i\in[n]$, $\langle\mathbf{1}, z_i\rangle \sim \mathcal{N}(0, d)$ so that $\mathbb{P}(|\langle\mathbf{1}, z_i\rangle| \geqslant t) \leqslant 2e^{-t^2/(2d)}$, and

$$\mathbb{P}(\forall i\in[n], |\langle\mathbf{1}, z_i\rangle| \leqslant t) \geqslant 1 - 2ne^{-t^2/(2d)} = 1 - \frac{2}{n^2},$$

for $t = 3\sqrt{d}\ln(n)$. Hence, with probability $1 - 3/n^2$, for all $i \neq j$ we have $|\sigma^2\langle z_i, z_j\rangle + \sigma\mu\langle \mathbf{1}, z_i + z_j\rangle| \leqslant (\sigma^2 + \sigma\mu)C\ln(n)/\sqrt{d}$.

Now,

$$\frac{(b-1)}{bn(n-1)}\sum_{i \neq j}\left(\sigma^2\langle z_i, z_j\rangle + \sigma\mu\langle \mathbf{1}, z_i + z_j\rangle\right)x_i x_j^\top = \frac{(b-1)}{bn(n-1)}\sum_{i<j}\left(\sigma^2\langle z_i, z_j\rangle + \sigma\mu\langle \mathbf{1}, z_i + z_j\rangle\right)(x_i x_j^\top + x_j x_i^\top)$$

$$\preceq \frac{(b-1)}{bn(n-1)}\sum_{i<j}\left|\sigma^2\langle z_i, z_j\rangle + \sigma\mu\langle \mathbf{1}, z_i + z_j\rangle)\right|(x_i x_i^\top + x_j x_j^\top),$$

where we used $x_i x_j^\top + x_j x_i^\top \preceq x_i x_i^\top + x_j x_j^\top$. Thus,

$$\frac{(b-1)}{bn(n-1)}\sum_{i \neq j}\left(\sigma^2\langle z_i, z_j\rangle + \sigma\mu\langle \mathbf{1}, z_i + z_j\rangle\right)x_i x_j^\top \preceq \sup_{i \neq j}\left|\sigma^2\langle z_i, z_j\rangle + \sigma\mu\langle \mathbf{1}, z_i + z_j\rangle)\right| \times \frac{(b-1)}{bn(n-1)}\sum_{i<j}\left(x_i x_i^\top + x_j x_j^\top\right)$$

$$= \sup_{i \neq j}\left|\sigma^2\langle z_i, z_j\rangle + \sigma\mu\langle \mathbf{1}, z_i + z_j\rangle\right| \times \frac{b-1}{b}\frac{1}{n-1}\sum_{i=1}^{n}x_i x_i^\top$$

$$= \sup_{i \neq j}\left|\sigma^2\langle z_i, z_j\rangle + \sigma\mu\langle \mathbf{1}, z_i + z_j\rangle\right| \times \frac{b-1}{b}\frac{n}{n-1}H\,.$$

Similarly, we have

$$\frac{(b-1)}{bn(n-1)}\sum_{i \neq j}\left(\sigma^2\langle z_i, z_j\rangle + \sigma\mu\langle \mathbf{1}, z_i + z_j\rangle\right)x_i x_j^\top \succeq -\sup_{i \neq j}\left|\sigma^2\langle z_i, z_j\rangle + \sigma\mu\langle \mathbf{1}, z_i + z_j\rangle)\right| \times \frac{b-1}{b}\frac{n}{n-1}H\,.$$

Hence, with probability $1 - 3/n^2$,

$$-\frac{(\sigma^2 + \sigma\mu)C\ln(n)}{\sqrt{d}} \times \frac{b-1}{b}\frac{n}{n-1}H \preceq \frac{(b-1)}{bn(n-1)}\sum_{i \neq j}\left(\sigma^2\langle z_i, z_j\rangle + \sigma\mu\langle \mathbf{1}, z_i + z_j\rangle\right)x_i x_j^\top$$

$$\preceq \frac{(\sigma^2 + \sigma\mu)C\ln(n)}{\sqrt{d}} \times \frac{b-1}{b}\frac{n}{n-1}H\,.$$

We thus have shown that this term (the one in the middle of the above inequality) is of smaller order.

We are hence left with $\frac{(b-1)}{bn(n-1)}\mu^2 d\sum_{i \neq j}x_i x_j^\top$. Denoting $\bar{x} = \frac{1}{n}\sum_i x_i$, we have $\frac{1}{n^2}\sum_{i \neq j}x_i x_j^\top = \frac{1}{n^2}\sum_{i,j}x_i x_j^\top - \frac{1}{n^2}\sum_i x_i x_i^\top$, so that:

$$\frac{(b-1)}{bn(n-1)}\mu^2 d\sum_{i \neq j}x_i x_j^\top = \frac{(b-1)n}{b(n-1)}\mu^2 d\left(\bar{x}\bar{x}^\top - \frac{1}{n}H\right)\,.$$

We note that we have $H = \frac{1}{n}\sum_i x_i x_i^\top = \frac{1}{n^2}\sum_{i<j}x_i x_i^\top + x_j x_j^\top \succeq \frac{1}{n^2}\sum_{i<j}x_i x_j^\top + x_j x_i^\top = \bar{x}\bar{x}^\top$ using $x_i x_i^\top + x_j x_j^\top \succeq x_i x_j^\top + x_j x_i^\top$. Thus, $H \succeq \bar{x}\bar{x}^\top \succeq 0$, and:

$$-\frac{(b-1)n}{b(n-1)}\mu^2 d\frac{1}{n}H \preceq \frac{(b-1)}{bn(n-1)}\mu^2 d\sum_{i \neq j}x_i x_j^\top \preceq \frac{(b-1)n}{b(n-1)}\mu^2 d(1 - 1/n)H\,.$$

We are now able to wrap everything together. With probability $1 - 2ne^{-d/16} - 3/n^2$, we have, for some numerical constants $c_1, c_2, c_3, C > 0$:

$$\left(c_1\frac{d(\mu^2 + \sigma^2)}{b} - c_2\frac{(\sigma^2 + \mu^2)\ln(n)}{\sqrt{d}} - c_3\frac{\mu^2 d}{n}\right)H \preceq \tilde{H}_b \preceq C\left(\frac{d(\mu^2 + \sigma^2)}{b} + \frac{(\sigma^2 + \mu^2)\ln(n)}{\sqrt{d}} + \mu^2 d\right)$$

$\square$