$\qquad\qquad\qquad\qquad\qquad\qquad\qquad\qquad\qquad\qquad\qquad\qquad\qquad\qquad\qquad\qquad\qquad\square$

## F  Proof of Theorem 1 and 2, and of Proposition 1

### F.1  Proof of Theorem 1 and 2

We are now equipped to prove Theorem 1 and Theorem 2, condensed in the following Theorem.

**Theorem 3.** *Let $(u_k, v_k)_{k \geqslant 0}$ follow the mini-batch SGD recursion (3) initialised at $u_0 = \sqrt{2}\alpha \in \mathbb{R}^d_{>0}$ and $v_0 = \mathbf{0}$, and let $(\beta_k)_{k \geqslant 0} = (u_k \odot v_k)_{k \geqslant 0}$. There exists and explicit $B > 0$ and a numerical constant $c > 0$ such that:*

1. *For stepsizes satisfying $\gamma_k \leqslant \frac{c}{LB}$, the iterates satisfy $\|\gamma_k \nabla \mathcal{L}_{\mathcal{B}_k}(\beta_k)\|_\infty \leqslant 1$ and $\|\beta_k\|_\infty \leqslant B$ for all $k$;*

2. *For stepsizes satisfying $\gamma_k \leqslant \frac{c}{LB}$, $(\beta_k)_{k \geqslant 0}$ converges almost surely to some $\beta^\star_\infty \in \mathcal{S}$,*

3. *If $(\beta_k)_k$ and the neurons $(u_k, v_k)_k$ respectively converge to a model $\beta^\star_\infty$ and neurons $(u_\infty, v_\infty)$ satisfying $\beta^\star_\infty \in \mathcal{S}$ (and $\beta^\star_\infty = u_\infty \odot v_\infty$), then for almost all stepsizes (with respect to the Lebesgue measure), the limit $\beta^\star_\infty$ satisfies:*

$$\beta^\star_\infty = \operatorname*{argmin}_{\beta^\star \in \mathcal{S}} D_{\psi_{\alpha_\infty}}(\beta^\star, \tilde{\beta}_0)\,,$$

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

where we used $x_ix_j^\top + x_jx_i^\top \preceq x_ix_i^\top + x_jx_j^\top$. Thus,

$$\frac{(b-1)}{bn(n-1)}\sum_{i\neq j}\left(\sigma^2\langle z_i, z_j\rangle + \sigma\mu\langle \mathbf{1}, z_i + z_j\rangle\right)x_ix_j^\top \preceq \sup_{i\neq j}\left|\sigma^2\langle z_i, z_j\rangle + \sigma\mu\langle \mathbf{1}, z_i + z_j\rangle\right| \times \frac{(b-1)}{bn(n-1)}\sum_{i<j}\left(x_ix_i^\top + x_jx_j^\top\right)$$

$$= \sup_{i\neq j}\left|\sigma^2\langle z_i, z_j\rangle + \sigma\mu\langle \mathbf{1}, z_i + z_j\rangle\right| \times \frac{b-1}{b}\frac{1}{n-1}\sum_{i=1}^{n}x_ix_i^\top$$

$$= \sup_{i\neq j}\left|\sigma^2\langle z_i, z_j\rangle + \sigma\mu\langle \mathbf{1}, z_i + z_j\rangle\right| \times \frac{b-1}{b}\frac{n}{n-1}H.$$

Similarly, we have

$$\frac{(b-1)}{bn(n-1)}\sum_{i\neq j}\left(\sigma^2\langle z_i, z_j\rangle + \sigma\mu\langle \mathbf{1}, z_i + z_j\rangle\right)x_ix_j^\top \succeq -\sup_{i\neq j}\left|\sigma^2\langle z_i, z_j\rangle + \sigma\mu\langle \mathbf{1}, z_i + z_j\rangle\right| \times \frac{b-1}{b}\frac{n}{n-1}H.$$

Hence, with probability $1 - 3/n^2$,

$$-\frac{(\sigma^2 + \sigma\mu)C\ln(n)}{\sqrt{d}} \times \frac{b-1}{b}\frac{n}{n-1}H \preceq \frac{(b-1)}{bn(n-1)}\sum_{i\neq j}\left(\sigma^2\langle z_i, z_j\rangle + \sigma\mu\langle \mathbf{1}, z_i + z_j\rangle\right)x_ix_j^\top$$

$$\preceq \frac{(\sigma^2 + \sigma\mu)C\ln(n)}{\sqrt{d}} \times \frac{b-1}{b}\frac{n}{n-1}H.$$

We thus have shown that this term (the one in the middle of the above inequality) is of smaller order.

We are hence left with $\frac{(b-1)}{bn(n-1)}\mu^2 d\sum_{i\neq j}x_ix_j^\top$. Denoting $\bar{x} = \frac{1}{n}\sum_i x_i$, we have $\frac{1}{n^2}\sum_{i\neq j}x_ix_j^\top = \frac{1}{n^2}\sum_{i,j}x_ix_j^\top - \frac{1}{n^2}\sum_i x_ix_i^\top$, so that:

$$\frac{(b-1)}{bn(n-1)}\mu^2 d\sum_{i\neq j}x_ix_j^\top = \frac{(b-1)n}{b(n-1)}\mu^2 d\left(\bar{x}\bar{x}^\top - \frac{1}{n}H\right).$$

We note that we have $H = \frac{1}{n}\sum_i x_ix_i^\top = \frac{1}{n^2}\sum_{i<j}x_ix_i^\top + x_jx_j^\top \succeq \frac{1}{n^2}\sum_{i<j}x_ix_j^\top + x_jx_i^\top = \bar{x}\bar{x}^\top$ using $x_ix_i^\top + x_jx_j^\top \succeq x_ix_j^\top + x_jx_i^\top$. Thus, $H \succeq \bar{x}\bar{x}^\top \succeq 0$, and:

$$-\frac{(b-1)n}{b(n-1)}\mu^2 d\frac{1}{n}H \preceq \frac{(b-1)}{bn(n-1)}\mu^2 d\sum_{i\neq j}x_ix_j^\top \preceq \frac{(b-1)n}{b(n-1)}\mu^2 d(1 - 1/n)H.$$

We are now able to wrap everything together. With probability $1 - 2ne^{-d/16} - 3/n^2$, we have, for some numerical constants $c_1, c_2, c_3, C > 0$:

$$\left(c_1\frac{d(\mu^2 + \sigma^2)}{b} - c_2\frac{(\sigma^2 + \mu^2)\ln(n)}{\sqrt{d}} - c_3\frac{\mu^2 d}{n}\right)H \preceq \tilde{H}_b \preceq C\left(\frac{d(\mu^2 + \sigma^2)}{b} + \frac{(\sigma^2 + \mu^2)\ln(n)}{\sqrt{d}} + \mu^2 d\right)$$

$\square$