# OpenReview forum: "(S)GD over Diagonal Linear Networks: Implicit bias, Large Stepsizes and Edge of Stability"
_NeurIPS.cc/2023/Conference — NeurIPS 2023 poster_

### Official Review · Reviewer_ZVE8 · 2023-07-06

**Soundness:** 3 good
**Presentation:** 3 good
**Contribution:** 3 good
**Rating:** 6
**Confidence:** 2

**Summary:**

This paper analyzes the implicit biases of GD and SGD on two-layer diagonal linear networks, specifically with large step sizes. This paper shows that SGD converges to a limit that is determined by the trajectory, and specifically by a certain effective initialization. Moreover, while both GD and SGD are close to gradient flow when the step size is small, it is shown that SGD may recover a sparse solution with larger step sizes. This can help explain the generation gap between GD and SGD with large step sizes.

**Strengths:**

This paper analyzes the implicit bias of SGD for a wide range of step sizes, which is an important contribution. In particular, it is pointed out that while larger batch sizes generally hurt the performance of GD, they may actually improve the accuracy of SGD before divergence, which is interesting.

**Weaknesses:**

My main concern is that the two-layer diagonal linear network model might be too simple, and more evidence is needed to show that observations in this setting can be transferred to more practical settings. Also the presentation is a bit technical; for example, the definition of the effective initialization is a little hard to follow. It could be better to consider a simple case to simplify the presentation while highlighting the interesting results, such as the different implicit biases of GD and SGD.

**Questions:**

N/A

---

> ### Author Rebuttal · Authors · 2023-08-09
>
> We thank the reviewer for the comments and the feedback. We would first like to point out a misunderstanding in the review (this is most certainly only a typo between batchsize and stepsize): *it is pointed out that while larger batch sizes generally hurt the performance of GD, they may actually improve the accuracy of SGD before divergence, which is interesting* should be replaced by *it is pointed out that while larger **stepizes** generally hurt the performance of GD, they may actually improve the accuracy of SGD before divergence, which is interesting*.
>
> **Limitations of DLNs.** DLNs are indeed a simplistic models of neural networks, a limitation acknowledged in our work. However, as we explain in lines 41-50, we believe that the rigorous study of complex phenomenons such as the intricate effect of noise and stepsizes has to start on such simple models. As put by Reviewer JT9J, *for now, the setting of diagonal linear networks might be the most complicated one we can expect for SGD-large-learning-rate bias to rigorously prove*.
>
> The definition of the effective initialization is indeed quite technical, but there is no simple case for which it has a simpler expression. However, its effect on the solution recovered is summarized as follows for sparse regression and large stepsizes: the effective initialization has heterogeneous coordinates that hinder the recovery of sparse vectors for GD, while for SGD its coordinates are of same order and lead to good sparse recovery.

---

> > ### Comment · Reviewer_ZVE8 · 2023-08-18
> >
> > Thanks for the response! I have no further questions.

---

### Official Review · Reviewer_JT8J · 2023-07-07

**Soundness:** 3 good
**Presentation:** 4 excellent
**Contribution:** 3 good
**Rating:** 7
**Confidence:** 4

**Summary:**

This is a theoretical work on implicit bias in diagonal linear networks. It proves the empirical observation by Pesme et al that SGD with large learning rates can recover the sparse signal while GD or small learning rates cannot. Technically it utilizes mirror descent with time-vary potentials.

**Strengths:**

1. The motivation and setting are great and non-trivial.

It is a popular and significant trend to understand the implicit bias with large learning rates, which is practically meaningful and much harder than previous works on small learning rates. For instance, the small-learning-rate version of this work is Woodworth et al for gradient flow.

Although the main empirical observation has been reported by Pesme et al last year, this work successfully proves it, completing the story. Meanwhile, for now, the setting of diagonal linear networks might be the most complicated one we can expect for SGD-large-learning-rate bias to rigorously prove.

2. The presentation and discussion are comprehensive.

The target setting is with several factors, including initialization scale, initialization shape, stochasticity, learning rates. This work does a good job to clearly its position in terms of these factors. In Section 3, it connects the main theorem 1 with previous gradient-flow result, with an emphasize on effective initialization $\alpha_{\infty}$. Then in Section 4, it carefully discusses how these four factors affect $\alpha_{\infty}$.

**Weaknesses:**

1. Is such a phenomena robust to different learning rates? In Figure 1, the x-axis region of significant drop of test loss for SGD seems to be not broad, and it looks hard to find the optimal $\gamma$ since it is too close to $\tilde{\gamma}_\max$ to explode. Meanwhile, the definition of $\tilde{\gamma}_\max$ around 257 is not much helpful to find the exact value. Moreover, my guess is that $\tilde{\gamma}_\max$ is not stable due to stochasticity in data generation and sgd sampling.

In other words, how can we choose a good lr for good sparse recovery?

2. In terms of practical computation, I am wondering how much SGD+large lr is better than GD+small initialization or SGD+label noise. It is mentioned in Section 4.1 that ''the slower the training, the larger the gain''. Hence it seems to bring more computation cost in practice for SGD to achieve good test error, while GD + small initialization may be faster in steps. And SGD+label noise might be another strong candidate.


**Questions:**

See above in Weakness.

For now, I would like to recommend an acceptance for its theoretical proof of an open problem.

---

> ### Author Rebuttal · Authors · 2023-08-09
>
> We thank the reviewer for the overall very positive feedback, all the helpful remarks and questions, that could lead to additional valuable discussions in our paper.
>
> **1.** The quantity $\tilde \gamma_{max}$ is not robust to the sampling of the random inputs $x_i$, as it is defined conditionally on them (for different samples of $(x_i)$'s, we empirically observe that $\tilde \gamma_{max}$ varies a lot, making it hard to estimate). However, it is empirically robust to SGD noise (to the noise induced by the sampling of mini-batches at each iteration): if SGD converges for some stepsize, we empirically observe that SGD converges for any smaller stepsize. Proving such a phenomenon seems however out of reach (even for GD).
>
> For sparse recovery, a good learning rate for SGD would be to choose a learning rate that produces oscillation in the initial phase, but still converges. As noticed, such a learning rate is hard to estimate, hence the interest of *learning rate schedules*, that start with a (too) large learning rate, but decreases it slowly during different phases.  These schedules are in fact also used in practice to increase generalization on more complex architectures; DLNs here offer a provable benefit for such schedules.
> We propose adding this discussion in a revised version, as we believe it will provide additional valuable insights into the effects of learning rates for SGD by discussing lr schedules.
>
> **2.** There is indeed a tradeoff between generalization power and computational efficiency here: empirically, taking into account its reduced complexity per iteration, SGD with large stepsizes performs better for large dimensions $d$, which can be explained by the fact that $\alpha$ needs to be taken too small for GD to converge quickly. Finally, SGD with large stepsizes or label noise SGD perform equally well. If the reviewer believes that illustrating this discussion with experiments could be a valuable addition, we propose to do so.

---

### Official Review · Reviewer_1MhA · 2023-07-09

**Soundness:** 3 good
**Presentation:** 4 excellent
**Contribution:** 3 good
**Rating:** 6
**Confidence:** 4

**Summary:**

This paper studies the impact of stochasticity and step size on the implicit regularization of GD and SGD, in the setting of two-layer diagonal linear network. It is show that large step size can benefit SGD, but hinder the performance of GD. Both the implicit bias and convergence are proved for (S)GD, providing a complete trajectory-dependent result. The results are further used to provide insights for explaining the "edge of stability" regime.

**Strengths:**

The writing is very clear and easy to follow, and highlighting of the important parts helps the reading a lot.

Theorem 1&2 together provides a complete set of results for trajectory-dependent analysis for GD and SGD for two-layer diagonal linear network, and the subsequent discussion on the impact of stochasticity and step size is clear and insightful.

**Weaknesses:**

It seems that the technical tools used in this paper have already been developed in previous related papers, especially [48, 50, 61] (corresponding to the reference number in the paper). Can the authors discuss the technical novelty of the current paper?

The gain vector does not have a closed-form expression, and it seems unclear what exact property does the minimization in (5) lead to. More specifically, does it give exact sparse recovery in some case?



**Questions:**

Theorem 2 gives the convergence guarantee. Is it possible to obtain an explicit convergence rate? If not, what's the difficulty here?

In Figure 1, the largest step sizes for GD and SGD are different. Why this difference doesn't appear in Theorem 2?

**Limitations:**

The authors have adequately addressed the limitations.

---

> ### Author Rebuttal · Authors · 2023-08-09
>
> We thank the reviewer for the time spent reviewing, the questions, and for the positive feedback.
>
> **Technical novelty.**
> Our technical tools are in fact very different from the cited references: our underlying process (SGD on the neurons) is **discrete** while theirs are **continuous-time** processes: gradient flow or stochastic GF.  Our analysis is thus much more challenging, and studies the real algorithms used in practice, as opposed to continuous time approximations.
> On the technical side,  [48, 50, 61] leverage a continuous mirror flow structure; our analysis is based on leveraging a (discrete) time varying mirror descent structure, indeed showing some similarity with  [48, 50, 61]. Analyzing the implicit bias in sections 4-5 is however completely new both technically speaking and in terms of insights on the effects of stepsizes and noise.
>
> In the general case, the minimization in (5) is hard to grasp; particular cases that lead to exact sparse recovery are GD or SGD with small initialization and stepsizes that are not in the EoS regime, or SGD with any initialization, but with an EoS stepsize.
>
> **Convergence rate.** From Proposition 6, we directly have $1/T \sum_{t<T} L(\beta_t)=O(1/T)$. However, for a quantitative bound on $||\beta_k-\beta_\infty||$, a more thorough analysis is required. In fact, we can show that if the loss becomes small (as can be proved with Prop. 6 in our paper), the loss becomes locally relatively strongly convex wrt the hypentropy. This property leads to a linear convergence (on the quantity $\Vert \beta_k-\beta_\infty \Vert_2$) to 0 with a rate of the form $O(\exp(-\alpha^2\mu\gamma k))$ where $\mu$ is the smallest non-null eigenvalue of $H$.
> We are open to incorporating this result into a revised version of the paper if the reviewer believes it would be beneficial. We did not include it due to space constraints and our primary focus on the implicit bias aspect of the problem.
>
> Question 2. : In Figure 1, the largest step sizes for GD and SGD are different, but this is also the case in Theorem 2, where the constant $L$ depends on the batchsize (see def in line 131).

---

> > ### Comment · Reviewer_1MhA · 2023-08-15
> > **Response to the authors**
> >
> > I thank the authors for their response, and I don't have further questions. I think it would be interesting to include the result on the convergence rate.

---

### Official Review · Reviewer_Y9wy · 2023-07-10

**Soundness:** 2 fair
**Presentation:** 3 good
**Contribution:** 2 fair
**Rating:** 5
**Confidence:** 4

**Summary:**

This paper studies the implicit bias of 2-layer diagonal linear networkss.
The authors show the convergence of GD and SGD with macroscopic stepsizes and characterise their solutions through an implicit regularization problem.
Moreover, the theoretical results reveal the difference between the generalization performances of GD and SGD for large stepsizes.

**Strengths:**

The paper is clear and well-written.
Wnderstanding what solutions do GD/SGD converges to is an important theoretical issue.
This paper achieves a clear characterization by deriving the solution found GD/SGD through an implicit regularization problem.
From a technical standpoint, this paper cleverly establishes an equivalent relationship between gradient descent and mirror descent of another problem, showcasing its potential for broader applications.
Furthermore, the authors analyze the gain in the implicit regularization problem, offering insightful explanations for the differences in generalization abilities between GD and SGD with varying step sizes.


**Weaknesses:**

This paper generalizes the existing results of the implicit regularization on GF to SGD, and the results are relatively similar, although I know this is indeed technically difficult.
Futhermore, 2-layer diagonal linear network and specific initialization restrict the influnce of this paper.

**Questions:**

1. While the authors have provided explanations of the Gain in Section 4 from various perspectives, I am still confused by its inisightful meaning. To address this, the authors can consider providing additional geometric insights or visual aids to help clarify the intuitive meaning of Gain.

2. The authors assert that the Gain of SGD is homogeneous in Equation (11) and Figure 2. However, it is important to note that Equation (11) corresponds to the expectation of the stochastic gradient rather than the realistic stochastic gradient. Previous studies, such as Zhu et al. (2018), have argued that the realistic stochastic gradient is highly anisotropic. This raises a potential contradiction, and it would be helpful if the authors could address this concern in the paper.


Zhu et al (2018). The anisotropic noise in stochastic gradient descent: Its behavior of escaping from sharp minima and regularization effects.

**Limitations:**

The results in the paper is limited to 2-layer diagonal linear network and specific initialization, but it should not be the reason for rejection.
This paper indeed offers a solid theoretical perspective on the implicit regularization of SGD.

---

> ### Author Rebuttal · Authors · 2023-08-09
>
> We thank the reviewer for the time spent reviewing, the thoughtful comments and reference.
>
> We first answer the ‘weakness’ part of the review.
> Our paper indeed generalizes existing results: references [48,50,61]  provide the implicit bias of gradient flow and stochastic gradient flow over DLNs, while we prove convergence of GD and SGD for macroscopic stepsizes, as well as determine the implicit bias for any arbitrary stepsizes that lead to convergence. Our results are thus much stronger since they directly study the algorithms which are used in practice and enables to gain insights on the effects of stepsizes and stochasticity. Similar characterisations were  not possible with previous works. We thus believe that our results are not just simple technical and incremental contributions.
>
> As we acknowledge in our paper, DLNs are indeed a very simple model of neural nets; however, we believe this model to be (currently) the only one for which a non-trivial implicit bias can be rigorously proved for large stepsize GD and SGD, as rightfully noted by Reviewer JT8J.
>
> Finally, we use a specific initialization of the neurons that lead to $\beta_0=0$, in order to have results that are clearer to understand. Our analysis can directly be extended to general initializations: non-centered initializations would only modify the term $\tilde \beta_0$ in Theorem 1; we can add a more thorough discussion about the initialization in a revised version.
>
> **Meaning of Gain.**
> The quantity Gain naturally appears in the analysis as being the key quantity which quantifies how much the solution found by (S)GD with given stepsizes deviates from that found by GF with same initialization. This quantity is insightful because it is  tractable and analyzable: the goal of sections 4 and 5 is precisely to leverage its definition in order to understand the impact of the stepsize and stochasticity on the recovered solution.
> A geometric insight could however be: for all iteration $k$, the interpolating vector that minimizes the potential $h_k$ (eq. (16)) related to the effective initialization $\alpha_k$ (line 621 in the appendix) is the solution found by GF started at $w_k$; hence the effective initializations $\alpha_k$ track the solutions found by GF starting from the iterates of (S)GD. The Gain and the effective initialization $\alpha_\infty$ are limits of these quantities for $k\to\infty$.
>
> **Anisotropic noise.**
> We thank the reviewer for this very interesting paper.
> Eq. (10) is indeed the expected squared stochastic gradient: it is thus related to both the true gradient and the noise, but the latter dominates. It is however important to note that these are the stochastic gradients of the **convex** loss $\mathcal L$, and not of the **non-convex** loss $F$ (that takes neurons as input). We thus do not believe Eq. (10) to be in contradiction with Zhu et al.: they argue that the noise of stochastic gradients of the **non-convex** loss is anisotropic, which we do not contradict. A thorough study of the SGD noise of the non-convex loss $F$ would show that for $s$-numerically sparse (i.e., almost or approximately sparse) neurons, the noise covariance and the non-convex Hessian (see Eq. (19)) are highly non-isotropic.
> We thank again the reviewer for this reference, and suggest to add it with such a discussion in a revised version.

---

> > ### Comment · Reviewer_Y9wy · 2023-08-11
> >
> > Thanks for the reviewer's response. No further questions from me.

---

### Official Review · Reviewer_uxxP · 2023-07-25

**Soundness:** 3 good
**Presentation:** 3 good
**Contribution:** 2 fair
**Rating:** 6
**Confidence:** 4

**Summary:**

The paper analyzes 2-layer diagonal linear networks and explains the different characteristics of the solutions received through GF, GD, and mini-batch SGD.

**Strengths:**

The paper analyzes the impact of the different step sizes and batch sizes and explains the differences between the solution received from training 2-layer diagonal linear networks using GF, GD, and mini-batch SGD. The paper shows that some of the differences are explainable using the linear scaling rule of \\( \frac{\text{step size}}{\text{batch size}} \\). However, the paper also shows other differences which are not explainable using the linear scaling rule.

**Weaknesses:**

Part of the theory, including theorems 2 and Proposition 1, only apply when the step size is small enough, and the edge of stability phenomenon does not occur. However, in this case, both GD and SGD behave approximately like GF. Thus any difference between the solutions received via the different optimization algorithms is minimal.

For larger step sizes, the theoretical analisis can not explain the full picture by itself. Rather, additional empirical observation are first being made, and then assumed to be true in order to complete the full analisis.

**Questions:**

1) What conditions were used to determine if the training converges, i.e., the training reached \\( w^\gamma_\infty \\)? In particular, in Figure 5, it seems like \\( \lambda_\max\left( \nabla^2F \left(  w^\gamma_\infty \right) \right) \\) is above the value of \\( 2/\gamma \\), which is impossible at convergence.

2) What is the effect of learning rate schedules? In particular, about Observation 1 and Figure 3, what would be the effect on coordinates not in \\( \text{supp}\left( \beta^*_{\text{ sparse }} \right) \\) if the step increases slowly so that the Oscillation phase is arbitrarily long?

3) In lines 290 and 295, did you perhaps mean to quote Eq.(10) and Eq.(11)?

**Limitations:**

The limitations are adequately addressed.

---

> ### Author Rebuttal · Authors · 2023-08-09
>
> We thank the reviewer for the thorough reviewing, the helpful remarks, and the noticed typos.
>
> As rightfully noticed and as acknowledged in our paper (lines 41-42), part of our results do not apply for large step sizes at the edge of stability. Nonetheless our main result (the implicit bias result in Theorem 1) holds **for any stepsize schedule such that the iterates converge**: we use this result in sections 4.2 and 5 together with empirical observations to provide informations on the behavior of GD at the edge of stability. Rigorously proving the observations made remains however a hard problem which we currently believe to be out of reach.
>
> The condition for the convergence is $\mathcal L (w_k)\leq \varepsilon$ for a small $\varepsilon$ ($\varepsilon=10^{-20}$).
> Indeed as accurately noticed there was a mild mistake in the code used to create Figure 5 (left and right): in this figure the y-axis should be rescaled by a factor 1/8 that comes from wrong normalizations (a 1/2 factor in front of the loss, and a 1/2 factor in the argument of the loss that leads to a 1/4 factor for second order derivatives, giving us this $1/8=1/2*1/4$ normalization error), we apologize for this mistake and will correct it. The trend of the figure is however unchanged (monotonicity of sharpness for GD and SGD).
>
> **Learning rate schedules.** The proposed learning rate schedule (slowly increasing the stepsize to keep oscillations) *will magnify the effects of large stepsizes*: for SGD, this will mimic label-noise SGD if we keep the magnitude of the oscillations of the same order, and lead to sparse recovery through l1 norm minimization; however, for GD, this will lead to recovery of the wrong support due to a weighted l1 norm minimization which is adversarial, as explained in the paper.
>
> In lines 290-295, we indeed meant to quote Eq. 10-11.

---

> > ### Comment · Reviewer_uxxP · 2023-08-18
> >
> > Thanks for the reviewer's response. I have no further questions.

---

### Decision · Program_Chairs · 2023-09-21

**Decision:**

Accept (poster)

**Comment:**

There is consensus among the reviewers on the validity of the problem setting and soundness of the results, but also on limitations of the theoretical characterization proposed in the paper. Nevertheless, on the balance the reviewers and I find the paper a worthy contribution, and consequently I recommend acceptance.